# NEAR-OPTIMAL ACTIVE REGRESSION OF SINGLE-INDEX MODELS

**Yi Li**
School of Physical and Mathematical Sciences and College of Computing and Data Science
Nanyang Technological University
yili@ntu.edu.sg

**Wai Ming Tai**
Independent Researcher
taiwaiming2003@gmail.com

## ABSTRACT

The active regression problem of the single-index model is to solve $\min_x \|f(Ax) - b\|_p$, where $A$ is fully accessible and $b$ can only be accessed via entry queries, with the goal of minimizing the number of queries to the entries of $b$. When $f$ is Lipschitz, previous results only obtain constant-factor approximations. This work presents the first algorithm that provides a $(1+\varepsilon)$-approximation solution by querying $\tilde{O}(d^{\frac{p}{2} \vee 1}/\varepsilon^{p \vee 2})$ entries of $b$. This query complexity is also shown to be optimal up to logarithmic factors for $p \in [1, 2]$ and the $\varepsilon$-dependence of $1/\varepsilon^p$ is shown to be optimal for $p > 2$.

## 1 INTRODUCTION

Active regression, an extension of the classical regression model, has gained increasing attention in recent years. In its most basic form, active regression aims to solve $\min_{x \in \mathbb{R}^d} \|Ax - b\|_p$ $(p \geq 1)$, where the matrix $A \in \mathbb{R}^{n \times d}$ represents $n$ data points in $\mathbb{R}^d$ and the vector $b \in \mathbb{R}^n$ represents the corresponding labels. However, since label access can be expensive, the challenge is to minimize the number of entries of $b$ that are accessed while still solving the regression problem approximately. A typical guarantee of the approximate solution is to find $\hat{x} \in \mathbb{R}^d$ such that

$$\|A\hat{x} - b\|_p \leq (1 + \varepsilon)\|Ax^* - b\|_p, \tag{1}$$

where $x^*$ is the optimal solution to the regression problem, i.e., $x^* = \arg\min_{x \in \mathbb{R}^d} \|Ax - b\|_p$.

Research on this problem often focuses on randomized algorithms with constant failure probability, i.e., the entries of $b$ are sampled randomly (but typically not uniformly) and the output $\hat{x}$ satisfies the error guarantee above with probability at least a large constant. When $p = 2$, Chen & Price (2019) showed sampling $O(d/\varepsilon)$ entries of $b$ suffices, and when $p = 1$, Parulekar et al. (2021) showed an optimal query complexity of $\Theta(d/\varepsilon^2)$. The case of general $p$ was later settled by Musco et al. (2022), who showed a query complexity of $\tilde{O}(d/\varepsilon)$ for $p \in (1, 2]$ and $\tilde{O}(d^{p/2}/\varepsilon^{p-1})$ for $p > 2$. Their proof was later refined by Yasuda (2024).

A more general form of the regression problem is the single-index model, which, in our context, asks to solve

$$\min_{x \in \mathbb{R}^d} \|f(Ax) - b\|_p,$$

where $f : \mathbb{R} \to \mathbb{R}$ is a nonlinear function and, for $u \in \mathbb{R}^n$, we abuse the notation slightly to denote $f(u) = (f(u_1), \ldots, f(u_n))$, i.e., applying $f$ entrywise. This formulation arises naturally in neural networks and has received significant recent attention (see, e.g., (Diakonikolas et al., 2020; Gajjar et al., 2023a; Huang et al., 2024) and the references therein). A neural network can be viewed as the composition of a network backbone and a linear prediction layer $x \in \mathbb{R}^d$ with an activation function $f$, where a typical choice of $f$ is the ReLU function. The network's prediction is given by $f(Ax)$,

where the matrix $A$ is the feature matrix generated by the network backbone from the dataset. The goal is to learn the linear prediction layer $x$, which corresponds to solving the regression problem.

In this paper, we consider $f$ to be a general $L$-Lipschitz function. It is tempting to expect an analogous guarantee as (1) for the canonical active regression problem, i.e. to find $\hat{x} \in \mathbb{R}^d$ such that

$$\|f(A\hat{x}) - b\|_p \le (1 + \varepsilon)\|f(Ax^*) - b\|_p$$

where $x^* = \arg\min_{x \in \mathbb{R}^d} \|f(Ax) - b\|_p$. However, Gajjar et al. (2023a) showed that achieving this guarantee with a $\mathrm{poly}(d)$ query complexity is impossible even when $f$ is a ReLU function and $\varepsilon$ is a constant. Hence, we can only expect a weaker guarantee. The single-index regression problem was studied for $p = 2$ in the same paper (Gajjar et al., 2023a), which showed that sampling $O(d^2/\varepsilon^4)$ entries suffices to find an $\hat{x} \in \mathbb{R}^d$ such that

$$\|f(A\hat{x}) - b\|_2^2 \le C(\|f(Ax^*) - b\|_2^2 + \varepsilon L^2\|Ax^*\|_2^2), \tag{2}$$

where $x^* = \arg\min_{x \in \mathbb{R}^d} \|f(Ax) - b\|_2$ is the minimizer and $C$ is an absolute constant. For general $p$, Huang et al. (2024) obtained

$$\|f(A\hat{x}) - b\|_p^p \le C(p)(\|f(Ax^*) - b\|_p^p + \varepsilon L^p\|Ax^*\|_p^p) \tag{3}$$

for some constant $C(p)$ depending only on $p$, using $O(d/\varepsilon^4)$ samples when $1 \le p \le 2$ and $O(d^{p/2}/\varepsilon^4)$ samples when $p > 2$. For $p = 2$, Gajjar et al. (2023b) also independently obtained $\tilde{O}(d/\varepsilon^4)$ samples and, with additional co-authors, further improved the query complexity to $\tilde{O}(d/\varepsilon^2)$ in (Gajjar et al., 2024).

The main drawback of the existing results for the single-index model, compared to the basic form of active regression, is that (2) and (3) only achieve constant-factor approximations, whereas (1) achieves a guarantee of $(1 + \varepsilon)$-approximation. The goal of this paper is to obtain a $(1 + \varepsilon)$-approximation for the single-index model.

Note that all existing results assume access to an oracle solver for the regression problem of the form $\arg\min_{x \in \mathbb{R}^d} \|f(A'x) - b'\|_p$ or its regularized variant, which may not be a convex programme and where the objective function may be non-differentiable due to $f$. In this work, we retain the assumption of having such an oracle solver.

## 1.1 PROBLEM DEFINITION

Now we define our problem formally. For $L \ge 0$, let $\mathsf{Lip}_L$ denote the set of $L$-Lipschitz functions $f$ such that $f(0) = 0$, i.e.

$$\mathsf{Lip}_L := \left\{ f : \mathbb{R} \to \mathbb{R} \mid f(0) = 0 \text{ and } |f(x) - f(y)| \le L \cdot |x - y| \text{ for all } x, y \in \mathbb{R} \right\}.$$

Suppose we are given a function $f \in \mathsf{Lip}_L$, a matrix $A \in \mathbb{R}^{n \times d}$ and a query access to the entries of an unknown $n$-dimensional vector $b \in \mathbb{R}^n$. Define

$$\mathsf{OPT} := \min_{x \in \mathbb{R}^d} \|f(Ax) - b\|_p^p \quad \text{and} \quad x^* := \underset{x \in \mathbb{R}^d : \|f(Ax) - b\|_p^p = \mathsf{OPT}}{\arg\min} \|Ax\|_p^p.$$

That is, if there are multiple minimizers $x^*$, we choose an arbitrary one that minimizes $\|Ax^*\|_p$. As noted in (Gajjar et al., 2024), there is no loss of generality in assuming $f(0) = 0$ because one can shift both $f(x)$ and $b$ by $f(0)$. For an error parameter $\varepsilon > 0$, our goal is to find an $\hat{x} \in \mathbb{R}^d$ such that

$$\|f(A\hat{x}) - b\|_p^p \le (1 + \varepsilon)\mathsf{OPT} + C\varepsilon\|Ax^*\|_p^p,$$

where $C$ is some constant that depends only on $L$ and $p$ while the number of queries to the entries of $b$ is minimized. Therefore, we would like to ask:

> *What is the minimum number (in terms of $d$ and $\varepsilon$) of queries to the entries of $b$ needed in order to achieve this goal?*

We will solve this problem in this paper.

## 1.2 OUR RESULTS

We first present the main result of this paper that querying $O(d/\varepsilon^2 \operatorname{poly} \log n)$ entries of $b$ is sufficient for a $(1 + \varepsilon)$-approximation when $1 \le p \le 2$ and $O(d^{p/2}/\varepsilon^p \operatorname{poly} \log n)$ entries when $p > 2$. Our result achieves the same query complexity (up to logarithmic factors) for the constant-factor approximation algorithm in (Gajjar et al., 2024) when $p = 2$.

**Theorem 1.** *There is a randomized algorithm, when given $A \in \mathbb{R}^{n \times d}$, $b \in \mathbb{R}^n$, $f \in \mathsf{Lip}_L$ and an arbitrary sufficient small $\varepsilon > 0$, with probability at least $0.9$, makes $O\big(d^{1 \vee \frac{p}{2}}/\varepsilon^{2 \vee p} \cdot \operatorname{poly} \log(d/\varepsilon)\big)$ queries to the entries of $b$ and returns an $\hat{x} \in \mathbb{R}^d$ such that*

$$\|f(A\hat{x}) - b\|_p^p \le \mathsf{OPT} + \varepsilon(\mathsf{OPT} + L^p\|Ax^*\|_p^p). \tag{4}$$

*The hidden constant in the bound on number of queries depends on $p$ only.*

Recall that in the canonical active regression problem, i.e. $f(x) = x$, the query complexity is $\tilde{\Theta}(d/\varepsilon)$ when $1 < p \le 2$ and $\tilde{\Theta}(d^{p/2}/\varepsilon^{p-1})$ when $p > 2$ (Musco et al., 2022). We can show that accommodating a general $f$ pushes up the $\varepsilon$-dependence to $1/\varepsilon^2$ and $1/\varepsilon^p$, respectively. In particular, for $1 \le p \le 2$, we show a lower bound of $\Omega(d/\varepsilon^2)$ queries, which suggests that our upper bound is tight up to logarithmic factors. The following are formal statements on our lower bound.

**Theorem 2.** *Suppose that $p \ge 1$, $\varepsilon > 0$ is sufficiently small and $n \gtrsim (d \log d)/\varepsilon^2$. Any randomized algorithm that, given $A \in \mathbb{R}^{n \times d}$, a query access to the entries of an unknown $b \in \mathbb{R}^n$ and $f \in \mathsf{Lip}_1$, outputs a $d$-dimensional vector $\hat{x}$ such that (4) holds with probability at least $4/5$ must make $\Omega(d/\varepsilon^2)$ queries to the entries of $b$.*

**Theorem 3.** *Suppose that $p > 2$, $\varepsilon > 0$ is sufficiently small and $n \gtrsim d/\varepsilon^p$. Any randomized algorithm that, given $A \in \mathbb{R}^{n \times d}$, a query access to the entries of an unknown $b \in \mathbb{R}^n$ and $f \in \mathsf{Lip}_1$, outputs a $d$-dimensional vector $\hat{x}$ such that (4) holds with probability at least $4/5$ must make $\Omega(d/\varepsilon^p)$ queries to the entries of $b$.*

## 2 PROOF OVERVIEW

In this section, we will provide a proof overview of our theorems. The full proof for the upper bound can be found in Appendix B and the full proofs for the lower bounds in Appendix C.

### 2.1 UPPER BOUND

**General Observations** We follow the usual "sample-and-solve" paradigm as in many previous active regression algorithms (Chen & Price, 2019; Musco et al., 2022; Gajjar et al., 2023a; Chen et al., 2022; Huang et al., 2024). Querying entries of $b$ can be viewed as multiplying $b$ with a sampling matrix $S$, which is a diagonal matrix with sparse diagonal entries; the nonzero entries in $Sb$ correspond to the queried entries of $b$. Hence, we would like to minimize the number of nonzero diagonal entries in $S$. The same sampling matrix $S$ is also applied to $f(Ax)$, leading to a natural attempt at solving the optimization problem $\min_{x \in \mathbb{R}^d} \|S(f(Ax) - b)\|_p^p$. However, we preview that this optimization problem is not the one we seek and we will provide more explanation on how to modify it.

The natural question is how to design the sampling matrix $S$. In all previous work, $S$ is a row-sampling matrix with respect to Lewis weights; see the statement of Lemma 6. In this paper, we adopt the same approach. This means that (i) (unbiased estimator) $\mathbb{E}\|Sv\|_p^p = \|v\|_p^p$ for each $v \in \mathbb{R}^n$ and (ii) (subspace embedding) when $S$ has sufficiently many nonzero rows, $\|SAx\|_p \approx \|Ax\|_p$ for all $x \in \mathbb{R}^d$. Note that the query complexity for active regression is usually higher than necessary for subspace embedding alone.

We will present the construction of the sampling matrix and the full algorithm in Section 3.

**Formulating the Concentration Bounds** Let $\hat{x} = \arg\min_{x \in \mathbb{R}^d} \|S(f(Ax) - b)\|_p^p$. Although $\mathbb{E}\|S(f(Ax) - b)\|_p^p = \|f(Ax) - b\|_p^p$ for *each* $x \in \mathbb{R}^d$, it is unclear what $\mathbb{E}\|S(f(A\hat{x}) - b)\|_p^p$ is since $\hat{x}$ depends on $S$. To address this, we instead argue that the sampling error

$\left| \|S(f(Ax) - b)\|_p^p - \|f(Ax) - b\|_p^p \right|$ is small for *all* $x \in T$, where $T$ is a "small" bounded domain containing $\hat{x}$. Here, the small size of $T$ is critical for a small error bound since it controls the number of points in $T$ at which the error needs to be small when applying Dudley's integral, a classical extension of the net argument. To further reduce the variance, we choose $\|S(f(Ax^*) - b)\|_p^p - \|f(Ax^*) - b\|_p^p$ as a reference point and argue that the difference

$$\mathsf{Err}(x) = |(\|S(f(Ax) - b)\|_p^p - \|f(Ax) - b\|_p^p) - (\|S(f(Ax^*) - b)\|_p^p - \|f(Ax^*) - b\|_p^p)|. \quad (5)$$

is uniformly small over $T$. Note that the reference point can be $\|S(f(A\bar{x}) - b)\|_p^p - \|f(A\bar{x}) - b\|_p^p$ for any $\bar{x} \in \mathbb{R}^d$. This approach has been employed by Yasuda (2024) for the canonical active regression (i.e. $f(x) = x$). However, for general Lipschitz functions $f$, a key challenge is to identify an appropriate domain $T$, which we shall discuss below.

**Regularized Regression** As previously noted, the optimization problem $\min_{x \in \mathbb{R}^d} \|S(f(Ax) - b)\|_p^p$ is not the one we are seeking. It turns out that there is a fundamental challenge to argue that $\hat{x} = \arg\min_{x \in \mathbb{R}^d} \|S(f(Ax) - b)\|_p^p$ satisfies the desired guarantee (4). In the canonical active regression, one can show that

$$\|A(\hat{x} - x^*)\|_p \lesssim \|Ax^* - b\|_p. \quad (6)$$

This suggests that $\hat{x} \in T$ for $T = \{ x \in \mathbb{R}^d \mid \|A(x - x^*)\|_p \lesssim \|Ax^* - b\|_p \}$, which is a "*small*" bounded region. Unfortunately, for a Lipschitz function $f$, it is not clear how to obtain an analogy to (6) and thus a bounded region $T$ containing $\hat{x}$. Hence, we still seek a bound on the norm $\|A\hat{x}\|_p^p$ to keep $T$ bounded and ideally small. For constant-factor approximations, previous work (Gajjar et al., 2023a; Huang et al., 2024) restrict the domain in the regression problem and solve $\min_{x \in T} \|S(f(Ax) - b)\|_p^p$ for some "small" $T$, but this leads to a poor $\varepsilon$-dependence of $1/\varepsilon^4$ in query complexity.

To improve the $\varepsilon$-dependence, one can consider a regularized regression problem

$$\min_{x \in \mathbb{R}^d} \|S(f(Ax) - b)\|_p^p + \tau \cdot \|Ax\|_p^p,$$

where $\tau > 0$ is a regularization parameter. This approach, as demonstrated by Gajjar et al. (2024), improves the $\varepsilon$-dependence for constant-factor approximations when $p = 2$, and will therefore be adopted in this paper. To ease the notation, assume that the Lipschitz constant $L = 1$ from now on.

An important question is how to choose the regularization parameter $\tau$. Recall that we want the sampling error $\mathsf{Err}(x)$ (defined in (5)) to be small, ideally close to 0, when $x = \hat{x}$. Then

$$\begin{aligned}
\|f(A\hat{x}) - b\|_p^p - \mathsf{OPT} &= \|f(A\hat{x}) - b\|_p^p - \|f(Ax^*) - b\|_p^p \\
&\leq \|S(f(A\hat{x}) - b)\|_p^p - \|S(f(Ax^*) - b)\|_p^p + \text{error terms} \\
&\leq \tau \cdot \|Ax^*\|_p^p + \text{error terms},
\end{aligned}$$

where the last inequality follows from the optimality of $\hat{x}$. The desired guarantee (4) has an additive error $\varepsilon\|Ax^*\|_p^p$, indicating that $\tau$ should be set smaller than $\varepsilon$. On the other hand, the main purpose of regularization is to bound $\|A\hat{x}\|_p^p$. By the optimality of $\hat{x}$ and rearranging the terms, we have

$$\|A\hat{x}\|_p^p \leq \frac{1}{\tau} \cdot \left( \|S(f(Ax^*) - b)\|_p^p - \|S(f(A\hat{x}) - b)\|_p^p \right) + \|Ax^*\|_p^p.$$

This suggests that $\tau$ should be set as large as possible for a better bound on $\|A\hat{x}\|_p^p$. Therefore, we set $\tau = \varepsilon$ and the optimization problem becomes

$$\min_{x \in \mathbb{R}^d} \|S(f(Ax) - b)\|_p^p + \varepsilon \cdot \|Ax\|_p^p. \quad (7)$$

**Bounding the Error** To avoid overloading our notation, we focus on $1 \leq p \leq 2$. A similar argument works for $p \geq 2$. Recall that our intention is to bound

$$\sup_{x \in T} \mathsf{Err}(x),$$

where $\mathsf{Err}(x)$ is the sampling error defined in (5). The first issue we need to resolve is defining the domain $T$. By the optimality of $\hat{x}$ in (7) and rearranging the terms, we have

$$\|A\hat{x}\|_p^p \le \frac{1}{\varepsilon}\left(\|S(f(Ax^*) - b)\|_p^p - \|S(f(A\hat{x}) - b)\|_p^p\right) + \|Ax^*\|_p^p \tag{8}$$

$$\le \frac{1}{\varepsilon}\|S(f(Ax^*) - b))\|_p^p + \|Ax^*\|_p^p$$

$$\lesssim \frac{1}{\varepsilon}\mathsf{OPT} + \|Ax^*\|_p^p \quad \text{by Markov inequality.} \tag{9}$$

Hence, we can set

$$R = \frac{1}{\varepsilon}\mathsf{OPT} + \|Ax^*\|_p^p \quad \text{and} \quad T = \{\, x \in \mathbb{R}^d \mid \|Ax\|_p^p \lesssim R \,\}.$$

Then $\hat{x} \in T$. Now, while we omit the details, we obtain the following concentration bound in (10) by following the standard technique of upper bounding the supremum of a stochastic process using Dudley's integral, which has been the central tool in the work on subspace embeddings (Bourgain et al., 1989; Ledoux & Talagrand, 1991; Cohen & Peng, 2015) and previous work on active regression (Musco et al., 2022; Chen et al., 2022; Huang et al., 2024). In short, when the number of queries is $m$, we can obtain that with probability at least $1 - \delta$,

$$\sup_{x \in T} \mathsf{Err}(x) \lesssim \sqrt{\frac{d\,\mathsf{poly}(\log n, \log \delta^{-1})}{m}} \cdot R. \tag{10}$$

We preview here that this concentration bound will yield a weaker result, but it serves to illustrate the key idea and will guide us in refining the analysis later.

Suppose that $m \sim \frac{d}{\varepsilon^4}\,\mathsf{poly}\log n$. By plugging $m$ and $R$ into (10), we have with constant probability,

$$\sup_{x \in T} \mathsf{Err}(x) \lesssim \varepsilon(\mathsf{OPT} + \varepsilon\|Ax^*\|_p^p).$$

Since $\hat{x} \in T$, we have

$$\|f(A\hat{x}) - b\|_p^p - \mathsf{OPT} \lesssim \|S(f(A\hat{x}) - b)\|_p^p - \|S(f(Ax^*) - b)\|_p^p + \varepsilon(\mathsf{OPT} + \varepsilon\|Ax^*\|_p^p)$$

$$\le \varepsilon\|Ax^*\|_p^p + \varepsilon(\mathsf{OPT} + \varepsilon\|Ax^*\|_p^p) \quad \text{by the optimality of } \hat{x} \text{ in (7)}$$

$$\lesssim \varepsilon(\mathsf{OPT} + \|Ax^*\|_p^p), \tag{11}$$

which achieves the desired guarantee (4) by a rescaling of $\varepsilon$.

**Attempt to Improve** The reason we previously set $m \sim \frac{d}{\varepsilon^4}\,\mathsf{poly}\log n$ is because $R = \frac{1}{\varepsilon}\mathsf{OPT} + \|Ax^*\|_p^p$ and we need the square root term in (10) to be $\varepsilon^2$ so that the overall error is at most $\varepsilon(\mathsf{OPT} + \|Ax^*\|_p^p)$. Indeed, when we compare to the canonical case of $f(x) = x$ and bound the radius of the domain in (6), this large $R$ is the main reason why extra factors of $\frac{1}{\varepsilon}$ are needed.

Notice that the term $\|S(f(Ax) - b)\|_p^p - \|S(f(Ax^*) - b)\|_p^p$ in $\mathsf{Err}(x)$ also appears in (8). This means that we can plug the bound on $\mathsf{Err}(x)$ into (8) and improve the radius $R$.

For example, let $m \sim \frac{d}{\varepsilon^3}\,\mathsf{poly}\log n$. Here, the exponent 3 can be replaced by any value *strictly* larger than 2 and we simply choose this number for demonstration purposes. By plugging $m$ and $R$ into (10), we have with constant probability

$$\sup_{x \in T} \mathsf{Err}(x) = \sup_{x \in T}|(\|S(f(Ax) - b)\|_p^p - \|S(f(Ax^*) - b)\|_p^p) - (\|f(Ax) - b\|_p^p - \|f(Ax^*) - b\|_p^p)|$$

$$\lesssim \varepsilon^{\frac{1}{2}}\mathsf{OPT} + \varepsilon^{\frac{3}{2}}\|Ax^*\|_p^p. \tag{12}$$

Since $\hat{x} \in T$, by plugging (12) into (8) and a similar calculation to that in (9), we have

$$\|A\hat{x}\|_p^p \lesssim \frac{1}{\varepsilon^{\frac{1}{2}}}\mathsf{OPT} + \|Ax^*\|_p^p \quad \text{which is smaller than } R.$$

If we set

$$R' = \frac{1}{\varepsilon^{\frac{1}{2}}}\mathsf{OPT} + \|Ax^*\|_p^p \quad \text{and} \quad T' = \{\, x \in \mathbb{R}^d \mid \|Ax\|_p^p \lesssim R' \,\},$$

then $\hat{x} \in T'$. By plugging $m$, $R'$ and $T'$ into (10), we have with constant probability,

$$\sup_{x \in T'} \mathsf{Err}(x) \lesssim \varepsilon \mathsf{OPT} + \varepsilon^{\frac{3}{2}} \|Ax^*\|_p^p.$$

This is an improved error bound compared to (12). It follows from $\hat{x} \in T'$ and a similar calculation to that in (11) that

$$\|f(A\hat{x}) - b\|_p^p - \mathsf{OPT} \lesssim \varepsilon(\mathsf{OPT} + \|Ax^*\|_p^p),$$

which achieves the guarantee (4). Therefore, we have successfully improved the query complexity from $\frac{d}{\varepsilon^4}$ poly $\log n$ to $\frac{d}{\varepsilon^3}$ poly $\log n$. Although this bootstrapping idea of reusing the error guarantee to shrink $T$ has appeared in previous work (Musco et al., 2022; Yasuda, 2024), we emphasize that there is a fundamental difference in the detailed analysis for general Lipschitz functions $f$, due to the lack of convexity of $\|f(A(\cdot)) - b\|_p^p$.

**Further Improvement**    Recall that we are targeting a query complexity of $\frac{d}{\varepsilon^2}$ poly $\log n$. One may immediately check that setting $m \sim \frac{d}{\varepsilon^2}$ poly $\log n$ in the above argument is not helpful. To address this issue, we refine the analysis of (10) and improve the bound as follows. Recall that we set $R = \frac{1}{\varepsilon}\mathsf{OPT} + \|Ax^*\|_p^p$ such that $\|A\hat{x}\|_p^p \lesssim R$. If we further restrict the domain $T$ and set it to be

$$T = \left\{ x \in \mathbb{R}^d \mid \|Ax\|_p^p \lesssim R \quad \text{and} \quad \|f(Ax) - f(Ax^*)\|_p^p \lesssim F \right\} \quad \text{for some } F \geq \mathsf{OPT}$$

then (10) can be improved to

$$\sup_{x \in T} \mathsf{Err}(x) \lesssim \sqrt{\frac{d \, \mathsf{poly}(\log n, \log \delta^{-1})}{m}} \cdot \sqrt{FR}. \tag{13}$$

Note that, by the Lipschitz condition and the fact that $\|Ax^*\|_p^p \lesssim R$, we have

$$\|f(Ax) - f(Ax^*)\|_p^p \leq \|Ax - Ax^*\|_p^p \lesssim R$$

and hence one can set $F = R$. That means (13) is always no worse than (10). To apply (13), we need to show that $\hat{x} \in T$, i.e. find a suitable $F$ such that $\|f(A\hat{x}) - f(Ax^*)\|_p^p \leq F$.

In the proof of a constant-factor approximation by Gajjar et al. (2024), a key step is

$$\|f(A\hat{x}) - f(Ax^*)\|_p^p \lesssim \mathsf{OPT} + \varepsilon \|Ax^*\|_p^p$$

when $p = 2$. A straightforward modification extends it to general $p$, which means that we can set $F = \mathsf{OPT} + \varepsilon \|Ax^*\|_p^p$.

Now, we pick $m \sim \frac{d}{\varepsilon^2}$ poly $\log n$. By plugging $m$, $R$ and $F$ into (13), we have

$$\sup_{x \in T} \mathsf{Err}(x) \lesssim \varepsilon \cdot \sqrt{(\mathsf{OPT} + \varepsilon \|Ax^*\|_p^p) \cdot (\frac{1}{\varepsilon}\mathsf{OPT} + \|Ax^*\|_p^p)} \lesssim \varepsilon^{\frac{1}{2}} \mathsf{OPT} + \varepsilon^{\frac{3}{2}} \|Ax^*\|_p^p. \tag{14}$$

Following the same argument before and plugging it into (8), we have

$$\|A\hat{x}\|_p^p \lesssim \frac{1}{\varepsilon^{\frac{1}{2}}} \mathsf{OPT} + \|Ax^*\|_p^p$$

which allows us to refine further the radius $R$ and thus the domain $T$, leading to a better bound on $\|A\hat{x}\|_p^p$. Iterate this process and apply (13) $\log \log \frac{1}{\varepsilon}$ times, we shall arrive at the bound

$$\mathsf{Err}(\hat{x}) = |(\|S(f(A\hat{x}) - b)\|_p^p - \|S(f(Ax^*) - b)\|_p^p) - (\|f(A\hat{x}) - b\|_p^p - \|f(Ax^*) - b\|_p^p)|$$
$$\lesssim \varepsilon \cdot (\mathsf{OPT} + \|Ax^*\|_p^p).$$

Finally, we follow a calculation similar to that in (11) to achieve the desired guarantee (4). The caveat here is that repeatedly applying the concentration bound (13) in the iterative process causes the failure probability to accumulate. We resolve this by setting $\delta = 1/\log \log(1/\varepsilon)$ in (13), keeping $\log(1/\delta)$ at most $\log n$. Hence, the query complexity remains $(d/\varepsilon^2)$ poly $\log n$.

**Dependence on** $n$  Although we have achieved the query complexity of $\frac{d}{\varepsilon^2} \operatorname{poly} \log n$, it may not be desirable when $n$ is large and we seek to further remove the dependence on $n$. The $\operatorname{poly} \log n$ factor arises from estimating a covering number when bounding Dudley's integral. Indeed, by using a simple net argument with a sampling matrix of $\operatorname{poly}(d/\varepsilon)$ non-zero rows, the solution guarantee can still be achieved. While the dependence on $d$ and $\varepsilon$ are both worse, the query complexity is independent of $n$. To take the advantage of this trade-off, a standard approach involves using two query matrices $S^\circ$ and $S$, where $S^\circ$ has the suboptimal number of nonzero rows, and then solving the following new regularized problem

$$\hat{x} = \underset{x \in \mathbb{R}^d}{\arg\min} \|SS^\circ(f(Ax) - b))\|_p^p + \varepsilon \|S^\circ Ax\|_p^p.$$

We need to pay close attention to the fact that we are not simply using $S^\circ A$ as the input matrix $A$ in the original statement because of the function $f$.

To bound the error, a natural attempt is to use the concentration bounds and show that

$$\begin{aligned}
\|f(A\hat{x}) &- b\|_p^p - \|f(Ax^*) - b\|_p^p \\
&\leq \|S^\circ(f(A\hat{x}) - b)\|_p^p - \|S^\circ(f(Ax^*) - b)\|_p^p + \text{error terms} && (15) \\
&\leq \|SS^\circ(f(A\hat{x}) - b)\|_p^p - \|SS^\circ(f(Ax^*) - b)\|_p^p + \text{error terms} && (16)
\end{aligned}$$

and then we use the optimality of $\hat{x}$ to complete the proof. However, we remind the readers that, to apply the concentration bounds, it is important to check that the relevant points $x^*, \hat{x}$ are in the domain of interest for the corresponding bounds. It turns out that we can obtain (15) but arguing (16) is the main obstacle because of that. While we will omit the detail, we note that we need a proxy point $x^\circ$ to circumvent this obstacle when using the concentration bound for $S$. The proxy point $x^\circ$ is defined as

$$x^\circ = \underset{x \in \mathbb{R}^d}{\arg\min} \|S^\circ(f(Ax) - b)\|_p^p + \varepsilon^2 \|Ax\|_p^p$$

and this proxy point allows us to show that $\hat{x}$ lies within the domain of interest. We can then modify the above argument by continuing from (15)

$$\begin{aligned}
\|S^\circ(f(A\hat{x}) &- b)\|_p^p - \|S^\circ(f(Ax^*) - b)\|_p^p \\
&\leq \|S^\circ(f(A\hat{x}) - b)\|_p^p - \|S^\circ(f(Ax^\circ) - b)\|_p^p + \varepsilon^2 \|Ax^*\|_p^p \\
&\leq \|SS^\circ(f(A\hat{x}) - b)\|_p^p - \|SS^\circ(f(Ax^\circ) - b)\|_p^p + \text{error terms}
\end{aligned}$$

and use the optimality of $\hat{x}$ to finish the proof.

## 2.2 Lower Bound

**General Observations**  As in the previous results (Musco et al., 2022; Yasuda, 2024), via Yao's minimax theorem, the proof of the lower bounds is reduced to distinguishing between two distributions (which are called hard instance). Specifically, we construct two "hard-to-distinguish" distributions on the vector $b$, and it requires a certain number of queries to the entries of $b$ to distinguish between these distributions with constant probability. The reduction is using an approximation solution $\hat{x}$ to determine from which distribution $b$ is drawn. We construct our hard instances for $1 \leq p \leq 2$ and $p \geq 2$ separately. These instances are inspired by those in (Musco et al., 2022; Yasuda, 2024) but are more complex, as we are showing a higher lower bound. For the purpose of exposition, we assume $d = 1$, in which case the matrix $A$ degenerates to a vector $a \in \mathbb{R}^n$. We shall then extend the result to the general $d$.

**Hard Instance for** $1 \leq p \leq 2$  We pair up the entries (say $2i - 1$ and $2i$). Let

$$u = \begin{bmatrix} 3 \\ 2 \end{bmatrix} \quad \text{and} \quad v = \begin{bmatrix} 2 \\ 3 \end{bmatrix}.$$

Let $D_0$ (resp. $D_1$) be the distribution on $b \in \mathbb{R}^{2n}$ that, for all $i = 1, \dots, n$, each pair

$$\begin{bmatrix} b_{2i-1} \\ b_{2i} \end{bmatrix} = \begin{cases} u & \text{with probability } \frac{1}{2} + \varepsilon \text{ (resp. } \frac{1}{2} - \varepsilon) \\ v & \text{with probability } \frac{1}{2} - \varepsilon \text{ (resp. } \frac{1}{2} + \varepsilon). \end{cases}$$

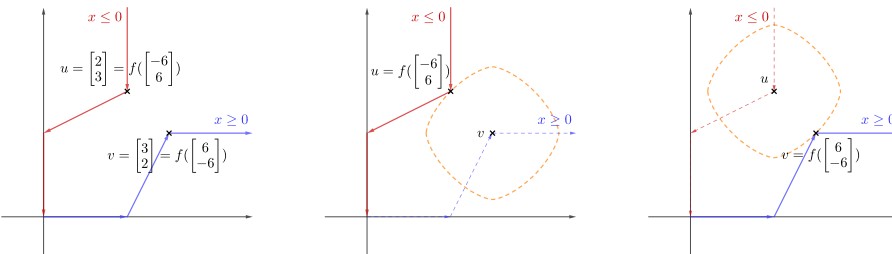

Figure 1: (left) Plot of the locus $f(\left[\begin{smallmatrix} x \\ -x \end{smallmatrix}\right])$, where the red (resp. blue) part corresponds to $x \leq 0$ (resp. $x \geq 0$); (middle) $f(\left[\begin{smallmatrix} -6 \\ 6 \end{smallmatrix}\right])$ is the point on the red part that is closest to $u$ and $v$ in the $\ell_p$-distance; (right) $f(\left[\begin{smallmatrix} 6 \\ -6 \end{smallmatrix}\right])$ is the point on the blue part that is closest to $u$ and $v$ in the $\ell_p$-distance

By the standard information-theoretic lower bounds, one needs to query $\Omega(\frac{1}{\varepsilon^2})$ entries of $b$ to distinguish $D_0$ and $D_1$.

To reduce this problem to our problem, we set

$$f(x) = \begin{cases} 2 & \text{if } x \leq -6 \\ -x - 4 & \text{if } -6 \leq x \leq -4 \\ 0 & \text{if } -4 \leq x \leq 0 \\ \frac{1}{2}x & \text{if } 0 \leq x \end{cases} \quad \text{and} \quad \begin{bmatrix} a_{2i-1} \\ a_{2i} \end{bmatrix} = \begin{bmatrix} 1 \\ -1 \end{bmatrix} \quad \text{for } i = 1, \ldots, n.$$

Let $k$ be the number of $u$'s in $b$. The objective function becomes

$$\|f(a \cdot x) - b\|_p^p = k \cdot \|f(\left[\begin{smallmatrix} x \\ -x \end{smallmatrix}\right]) - u\|_p^p + (n - k) \cdot \|f(\left[\begin{smallmatrix} x \\ -x \end{smallmatrix}\right]) - v\|_p^p.$$

The takeaway of this construction is one can view $f(\left[\begin{smallmatrix} x \\ -x \end{smallmatrix}\right])$ as a locus in $\mathbb{R}^2$ as $x$ varies, illustrated in Figure 1. Suppose $b$ is drawn from $D_0$. It implies that $k \approx \frac{n}{2} + \varepsilon n$ and hence $n - k < k$. One can view each component as the $\ell_p$ distance between the locus and $u$ or $v$. As seen in Figure 1, the locus passes through $u$ and $v$. When $x = -6$, we have $f(\left[\begin{smallmatrix} x \\ -x \end{smallmatrix}\right]) = u$ and so

$$\mathsf{OPT} \leq k \cdot 0 + (n - k) \cdot \|u - v\|_p^p = 2^p(n - k) \approx 2^{p-1}n(1 - 2\varepsilon).$$

On the other hand, Figure 1 also suggests that, when $x > 0$, we have

$$\|f(a \cdot x) - b\|_p^p \geq k \cdot \|u - v\|_p^p + (n - k) \cdot 0 \geq 2^p k \approx 2^{p-1}n(1 + 2\varepsilon).$$

Suppose we have a solution $\hat{x}$ such that

$$\begin{aligned}
\|f(a \cdot \hat{x}) - b\|_p^p &\leq (1 + c \cdot \varepsilon)\mathsf{OPT} + c \cdot \varepsilon \cdot \|a \cdot x^*\|_p^p \\
&\leq (1 + c \cdot \varepsilon)2^p(n - k) + c \cdot \varepsilon \cdot \|a \cdot x^*\|_p^p \\
&\approx (1 + c \cdot \varepsilon) \cdot 2^{p-1}n(1 - 2\varepsilon) + c \cdot \varepsilon \cdot 6^p \cdot 2n \\
&= 2^{p-1}n(1 - \Omega(\varepsilon)) \quad \text{for a sufficiently small } c > 0 \\
&< 2^{p-1}n(1 + 2\varepsilon)
\end{aligned}$$

which implies that $\hat{x} < 0$. Similarly, suppose $b$ is drawn from $D_1$, one can show the symmetric result. We can declare $b$ is drawn from $D_0$ if $\hat{x} < 0$ and $D_1$ otherwise. This concludes our reduction.

**Hard Instance for $p \geq 2$** We start with the all-one vector $b \in \mathbb{R}^{2n}$. Then, we pick a random index $i^*$ from $\{1, \ldots, 2n\}$ uniformly and update $b_{i^*} \leftarrow b_{i^*} + 1/\varepsilon$. Our question is to determine whether $i^* \leq n$ or $i^* > n$, and it follows from a straightforward probability calculation that $\Omega(n)$ queries to the entries of $b$ are required. Recall that we are targeting a query complexity of $\Omega(1/\varepsilon^p)$ and hence we set $n = 1/\varepsilon^p$.

To reduce this problem to our problem, we set

$$f(x) = \begin{cases} 0 & \text{if } x \leq 0 \\ x & \text{if } 0 \leq x \leq 1 \\ 1 & \text{if } 1 \leq x. \end{cases} \quad \text{and} \quad a_i = \begin{cases} 1 & \text{if } i = 1, \ldots, n \\ -1 & \text{if } i = n + 1, \ldots, 2n. \end{cases}$$

---

**Algorithm 1** Generating a Sampling Matrix $\texttt{GSM}(k_1, \ldots, k_n, \alpha)$

---

**Input:** $n$ integers $k_1, \ldots, k_n \geq 0$; a sampling rate $\alpha < 1$
 1: $S \leftarrow$ an $n \times n$ diagonal matrix, initialized to a zero matrix
 2: **for** $i = 1, \ldots, n$ **do**
 3:    **if** $k_i > 0$ **then**
 4:       Generate a binomial random variable $N_i \sim \text{Bin}(k_i, \alpha)$
 5:       $S_{ii} \leftarrow \left(\frac{N_i}{\alpha k_i}\right)^{\frac{1}{p}}$
 6: Return $S$

---

Suppose that $i^* \leq n$. If $x = 1$, we have the following. For $i = 1, \ldots, n$ and $i \neq i^*$, we have $f(a \cdot x)_i = f(1) = 1 = b_i$. Recall that $b_{i^*} = 1 + 1/\varepsilon$. Hence, we have $\sum_{i=1}^{n} |f(a \cdot x)_i - b_i|^p = 1/\varepsilon^p = n$. For $i = n + 1, \ldots, 2n$, we have $f(a \cdot x)_i = f(-1) = 0$ and therefore we have $\sum_{i=n+1}^{2n} |f(a \cdot x)_i - b_i|^p = \sum_{i=n+1}^{2n} 1^p = n$. Namely, we have

$$\text{OPT} \leq \|f(a \cdot x) - b\|_p^p = 2n.$$

On the other hand, it is easy to check that, when $x < 0$, we have

$$\|f(a \cdot x) - b\|_p^p \geq \sum_{i=1}^{n} |f(a \cdot x)_i - b_i|^p \geq n - 1 + \left(\frac{1}{\varepsilon} + 1\right)^p \geq 2n(1 + \varepsilon)$$

Suppose we have a solution $\hat{x}$ such that

$$\|f(a \cdot \hat{x}) - b\|_p^p \leq (1 + c \cdot \varepsilon)\text{OPT} + c \cdot \varepsilon \|a \cdot x^*\|_p^p$$
$$\leq (1 + c \cdot \varepsilon) \cdot 2n + c \cdot \varepsilon \cdot 1^p \cdot n$$
$$< 2n(1 + \varepsilon) \quad \text{for a sufficiently small } c > 0$$

which implies $\hat{x} > 0$. Similarly, suppose that $i^* \geq n + 1$, one can show the symmetric result. We can declare $i^* \leq n$ if $\hat{x} > 0$ and $i^* > n$ otherwise. This concludes our reduction.

**Extension to $d > 1$**   We consider the problem of solving multiple independent copies of hard instances of $d = 1$ and reduce this new problem to the regression. The formal construction is as follows. Let $m = \Theta(1/\varepsilon^{p \vee 2})$. We have a $dm$-dimensional vector $b$, which can be partitioned into $d$ blocks of $m$-dimensional vectors, with each block drawn from either $D_0$ or $D_1$ (the hard instances introduced earlier depending on $p$). By a straightforward probability calculation, it can be shown that $\Omega(dm)$ queries to the entries of $b$ are needed to correctly answer, with constant probability, which distribution each block of $m$-dimensional vector is drawn from, for at least $2d/3$ blocks.

To reduce it to our problem, let $A$ be a $dm$-by-$d$ block-diagonal matrix, partitioned into $d^2$ blocks of $m$-dimensional vectors. Each diagonal block is the vector $a$ which we constructed earlier. The function $f$ remains the same as before. Suppose we have a solution $\hat{x}$ satisfying (4). By the independence between blocks in $b$ and the block-diagonal structure of $A$, we can argue that (4) can be decomposed into the sum of the objective functions for each independent block and declare that each block is drawn from $D_0$ or $D_1$ based on the same criteria as in the case of $d = 1$. By the standard counting techniques, at least $2d/3$ of the $d$ answers are correct and this completes the reduction. Hence, we achieve the query complexity of $d/\varepsilon^{p \vee 2}$.

We point out that for the canonical case of $f(x) = x$ and $p \geq 2$, the previous result of Yasuda (2024) gives a stronger lower bound, in terms of $d$, of $\Omega(d^{p/2}/\varepsilon^{p-1})$. Unfortunately, it is still not clear how to apply the techniques in our setting.

## 3   ALGORITHM

To complement the proof overview in Section 2.1, we present our full algorithm in Algorithm 2 and explain the explicit implementation.

It first constructs a sampling matrix $S^\circ$ (line 1 to line 4 of Algorithm 2) and applies it to $A$, $f(A(\cdot))$ and $b$. This sampling matrix $S^\circ$ is generated using Algorithm 1. When applying $S^\circ$ to $A$, in step 5

---

**Algorithm 2** Algorithm for Active Learning without Dependence on $n$

---

**Input:** a matrix $A \in \mathbb{R}^{n \times d}$
  a query access to the entries of the vector $b \in \mathbb{R}^n$
  a function $f \in \mathsf{Lip}_L$
  an error parameter $\varepsilon$
  two sampling rates $\alpha < \alpha^\circ < 1$

1: Compute the Lewis weights of $A$, denoted by $w_1(A), \ldots, w_n(A)$
2: **for** $i = 1, \ldots, n$ **do**
3:     $k_i^\circ \leftarrow \lceil \frac{n \cdot w_i(A)}{d} \rceil$
4: $S^\circ \leftarrow \mathsf{GSM}(k_1^\circ, \ldots, k_n^\circ, \alpha^\circ)$ from Algorithm 1
5: $m \leftarrow$ number of nonzero rows in $S^\circ$
6: Compute the Lewis weights of $S^\circ A$, denoted by $w_1(S^\circ A), \ldots, w_n(S^\circ A)$
7: **for** $i = 1, \ldots, n$ **do**
8:     $k_i \leftarrow \lceil \frac{m \cdot w_i(S^\circ A)}{d} \rceil$
9: $S \leftarrow \mathsf{GSM}(k_1, \ldots, k_n, \alpha)$ from Algorithm 1
10: Solve the minimization problem $\hat{x} := \arg\min_{x \in \mathbb{R}^d} \|SS^\circ f(Ax) - SS^\circ b\|_p^p + \varepsilon \|S^\circ Ax\|_p^p$
11: Return the vector $\hat{x} \in \mathbb{R}^d$

---

of Algorithm 1, it is equivalent to splitting the rows of $A$ such that all rows have uniformly bounded Lewis weights of $O(d/n)$. To achieve this, it needs the Lewis weights of $A$ and they can be computed as in (Cohen & Peng, 2015) for $p < 4$ and as in (Fazel et al., 2022) for $p \geq 4$. Afterwards, we sample each row with the same probability $\alpha^\circ$. This row-splitting approach has been used in the proofs of Cohen & Peng (2015); Yasuda (2024) and in the algorithms in (Gajjar et al., 2023b; 2024). Details of this row-splitting technique can be found in Appendix B.1.

We set the sampling rate $\alpha^\circ = \mathsf{poly}(d/\varepsilon)/n$. This effectively reduces the dimension from $n$, the number of rows of $A$, to $m \sim \alpha^\circ n = \mathsf{poly}(d/\varepsilon)$, the number of non-zero rows of $S^\circ A$. Therefore, it removes the dependence on $n$ in our bound.

It then constructs the main sampling matrix $S$ (line 6 to line 9 of Algorithm 2) with the sampling rate $\alpha = d^{\frac{p}{2} \vee 1}/(\varepsilon^{p \vee 2} m) \, \mathsf{poly}\log(m)$, whereby avoiding dependence on $n$ as previously discussed, and applies to $S^\circ A$, $S^\circ f(A(\cdot))$ and $S^\circ b$. That means that the number of non-zero entries of $SS^\circ b$ is, with high probability, at most $2\alpha m \sim d^{\frac{p}{2} \vee 1}/(\varepsilon^{p \vee 2}) \, \mathsf{poly}\log(d/\varepsilon)$, which is the query complexity we are looking for. Note that $S$ is also generated using Algorithm 1 and hence satisfies the property of uniformly bounded Lewis weights through the previously mentioned row-splitting techniques. Finally, the algorithm outputs the optimal solution $\hat{x}$ of the regularized problem

$$\min_{x \in \mathbb{R}^d} \|SS^\circ f(Ax) - SS^\circ b\|_p^p + \varepsilon \|S^\circ Ax\|_p^p$$

and that completes our full algorithm.

## 4 CONCLUSION

In this paper, we consider the active regression problem of the single-index model, which asks to solve $\min_x \|f(Ax) - b\|_p$, with $f$ being a Lipschitz function, $A$ fully accessible and $b$ only accessible via entry queries. The goal is to minimize the number of queries to the entries of $b$ while achieving an accurate solution to the regression problem. Previous work on single-index model has only achieved constant-factor approximations (Gajjar et al., 2023a;b; Huang et al., 2024; Gajjar et al., 2024). In this paper, we achieve a $(1 + \varepsilon)$-approximation with $\tilde{O}(d^{\frac{p}{2} \vee 1}/\varepsilon^{p \vee 2})$ queries and we show that this query complexity is tight for $1 \leq p \leq 2$ up to logarithmic factors. Furthermore, we prove that the $1/\varepsilon^p$ dependence is tight for $p > 2$ and we leave the full tightness of $d^{p/2}/\varepsilon^p$ as an open problem for future work.

## ACKNOWLEDGEMENTS

Y. Li was supported in part by Singapore Ministry of Education AcRF Tier 2 grant MOE-T2EP20122-0001 and Tier 1 grant RG75/21. W. M. Tai was supported by Singapore Ministry of

Education AcRF Tier 2 grant MOE-T2EP20122-0001 when he was affiliated with Nanyang Technological University, where most part of this work was done.

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

## A    PRELIMINARIES

**Notation**    For a distribution $\mathcal{D}$, we write $X \sim \mathcal{D}$ to denote a random variable $X$ drawn from $\mathcal{D}$ and $\beta \cdot \mathcal{D}$ to denote the distribution of the scaled random variable $\beta X$, where $X \sim \mathcal{D}$. For any $0 \le p \le 1$ and positive integer $n$, we use $\mathrm{Ber}(p)$ to denote the Bernoulli distribution with expected value $p$ and $\mathrm{Bin}(n, p)$ to denote the Binomial distribution with $n$ trials and success probability $p$ for each trial. That is, if $X \sim \mathrm{Ber}(p)$ then

$$X = \begin{cases} 1 & \text{with probability } p \\ 0 & \text{with probability } 1-p \end{cases}$$

and if $X \sim \mathrm{Bin}(n, p)$ then $X$ can be expressed as $\sum_{i=1}^n X_i$ where $X_1, \ldots, X_n$ are i.i.d. $\mathrm{Ber}(p)$ variables.

For a matrix $A$, we use $A_{i,\cdot}$ to denote its $i$-th row and $A_{\cdot,i}$ its $i$-th column. For $\lambda_1, \ldots, \lambda_n \in \mathbb{R}$, we use $\mathrm{diag}\{\lambda_1, \ldots, \lambda_n\}$ to denote a diagonal matrix whose diagonal entries are $\lambda_1, \ldots, \lambda_n$.

In a normed space $(X, \|\cdot\|)$, the unit ball $B(X)$ is defined as $B(X) = \{\, x \in X \mid \|x\| \le 1 \,\}$. When $X$ is clear from the context, we may omit the space and write only $B$ for the unit ball. When $X$ is the column space of a matrix $A$, we also write the unit ball as $B(A)$. If the norm has a subscript $\|\cdot\|_\square$, we shall include the subscript of the norm and denote the associated unit ball by $B_\square$ (or $B_\square(A)$ if $X$ is the column space of $A$). In $\mathbb{R}^n$, the standard $\ell_p$-norm and the weighted $\ell_p$-norm, denoted by $\|\cdot\|_p$ and $\|\cdot\|_{w,p}$, are defined as $\|x\|_p = (\sum_{i=1}^n |x_i|^p)^{1/p}$ and $\|x\|_{w,p} = (\sum_{i=1}^n w_i |x_i|^p)^{1/p}$, respectively, where $w \in \mathbb{R}^n$ and $w_i > 0$ for $i \in [n]$.

We shall use $C$, $C_1$, $C_2$, ..., $c$, $c_1$, $c_2$, ... to denote absolute constants. We also write $\max\{a, b\}$ and $\min\{a, b\}$ as $a \vee b$ and $a \wedge b$, respectively. We use $O, \Omega, \Theta$ and $\lesssim, \gtrsim, \sim$ interchangeably.

**Lewis Weights**    We now define an important concept regarding matrices, which have played critical roles in the construction of space-efficient subspace embeddings.

**Definition 4** ($\ell_p$-Lewis weights)**.** *Let $A \in \mathbb{R}^{n \times d}$ and $p \ge 1$. For each $i \in [n]$, the $\ell_p$-Lewis weight of $A$ for the $i$-th row is defined to be $w_i$ that satisfies*

$$w_i(A) = (a_i^\top (A^\top W^{1 - \frac{2}{p}} A)^\dagger a_i)^{\frac{p}{2}}$$

*where $a_i$ is the $i$-th row of $A$ (as a column vector), $W = \mathrm{diag}\{w_1, \ldots, w_n\}$ and $\dagger$ denotes the pseudoinverse.*

When the matrix $A$ is clear in context, we will simply write $w_i(A)$ as $w_i$. Adopting that $0 \cdot \infty = 0$, we have $w_i(A) = 0$ if $a_i = 0$. The following are a few important properties of Lewis weights; see, e.g., (Wojtaszczyk, 1991) for a proof.

**Lemma 5** (Properties of Lewis weights). *Suppose that $A \in \mathbb{R}^{n \times d}$ has full column rank and Lewis weights $w_1, \ldots, w_n$. Let $W = \mathrm{diag}\{w_1, \ldots, w_n\}$. The following properties hold.*

- (a) $\sum_i w_i = d$;
- (b) *There exists a matrix $U \in \mathbb{R}^{n \times d}$ such that*
    - (i) *the column space of $U$ is the same as that of $A$;*
    - (ii) $w_i = \|U_{i,\cdot}\|_2^p$;
    - (iii) $W^{\frac{1}{2} - \frac{1}{p}} U$ *has orthonormal columns;*
- (c) *It holds for all vectors $u$ in the column space of $A$ that $\|W^{\frac{1}{2} - \frac{1}{p}} u\|_2 \leq d^{\frac{1}{2} - \frac{1}{2 \vee p}} \|u\|_p$.*
- (d) *It holds for all vectors $u$ in the column space of $A$ that $|u_i| \leq d^{\frac{1}{2} - \frac{1}{2 \vee p}} w_i^{\frac{1}{p}} \|u\|_p$.*

**Subspace Embeddings**  Suppose that $A \in \mathbb{R}^{n \times d}$ and $\varepsilon \in (0, 1)$. We say a matrix $S \in \mathbb{R}^{m \times n}$ is an $\ell_p$-subspace embedding matrix for $A$ with distortion $1 + \varepsilon$ if $(1 + \varepsilon)^{-1} \|Ax\|_p \leq \|SAx\|_p \leq (1 + \varepsilon)\|Ax\|_p$. The prevailing method to construct $\ell_p$-subspace embedding matrices is to sample the rows of $A$ according to its Lewis weights.

**Lemma 6.** *Suppose that $A \in \mathbb{R}^{n \times d}$ has Lewis weights $w_1, \ldots, w_n$. Let $p_i \in [0, 1]$ satisfy that $p_i \geq (\beta w_i) \wedge 1$ and $S \in \mathbb{R}^{n \times n}$ be a diagonal matrix with independent diagonal entries $S_{ii} \sim p_i^{-1/p} \mathrm{Ber}(p_i)$. Then with probability at least $0.99$, $S$ is an $\ell_p$-subspace embedding matrix for $A$ with distortion $1 + \varepsilon$ if*

$$
\beta \gtrsim_p \begin{cases} \frac{1}{\varepsilon^2} \log \frac{d}{\varepsilon} (\log \log \frac{d}{\varepsilon})^2, & 1 < p < 2 \\ \frac{1}{\varepsilon^2} \log \frac{d}{\varepsilon}, & p = 1, 2 \\ \frac{d^{\frac{p}{2} - 1}}{\varepsilon^2} (\log d)^2 \log \frac{d}{\varepsilon} & p > 2. \end{cases}
$$

The results for $p \in [1, 2]$ are due to Cohen & Peng (2015), based on earlier work of Talagrand (1990; 1995). The result for $p > 2$ can be found in (Yasuda, 2024; Huang et al., 2024), which improves upon the previous bound $\beta \gtrsim (d^{p/2-1}/\varepsilon^5)(\log d)\log(1/\varepsilon)$ in (Bourgain et al., 1989; Cohen & Peng, 2015).

**Covering Numbers and Dudley's Integral**  Suppose that $T$ is a pseudometric space endowed with a pseudometric $d$. The diameter of $T$, denoted by $\mathrm{Diam}(T, d)$, is defined as $\mathrm{Diam}(T, d) := \sup_{t,s \in T} \rho(t, s)$.

Given an $r > 0$, an $r$-covering of $(T, d)$ is a subset $X \subseteq T$ such that for every $t \in T$, there exists $x \in X$ such that $d(t, x) \leq r$. The covering number $\mathcal{N}(T, d, r)$ is the minimum number $K$ such that there exists an $r$-covering of cardinality $K$.

The covering numbers are intrinsically related to a subgaussian process on the space $T$ that conforms to the pseudometric $d$. This relationship is captured by the well-known Dudley's integral.

**Lemma 7** (Dudley's integral Vershynin (2018)). *Let $X_t$ be a zero-mean stochastic process that is subgaussian w.r.t. a pseudo-metric $d$ on the indexing set $T$. Then it holds that*

$$
\Pr \left\{ \sup_{t,s \in T} |X_t - X_s| > C \left( \int_0^\infty \sqrt{\log N(T, d, \varepsilon)} d\varepsilon + u \cdot \mathrm{Diam}(T) \right) \right\} \leq 2 \exp(-u^2),
$$

*where $C$ is an absolute constant. As a consequence,*

$$
\mathbb{E} \left( \sup_{t,s \in T} |X_t - X_s| \right)^\ell \leq C' \cdot C^\ell \left[ \left( \int_0^\infty \sqrt{\log N(T, d, \varepsilon)} d\varepsilon \right)^\ell + (\sqrt{\ell} \, \mathrm{Diam}(T))^\ell \right],
$$

*where $C$ and $C'$ are absolute constants.*

Note that when $r > \mathrm{Diam}(T, d)$, the covering number $\mathcal{N}(T, d, r) = 1$ and thus the integrand becomes $0$. Hence, Dudley's integral is in fact taken over the finite interval $[0, \mathrm{Diam}(T, d)]$.

The following covering numbers related to $\|\cdot\|_{w,p}$ will be useful our analysis. These are not novel results though we include a proof in Appendix E for completeness.

**Lemma 8.** *Suppose that $A \in \mathbb{R}^{n \times d}$ has full column rank and $W$ is a diagonal matrix whose diagonal entries are the Lewis weights of A. It holds that*

$$\log \mathcal{N}(B_{w,p}(W^{-1/p}A), \|\cdot\|_{w,q}, t) \lesssim \begin{cases} d \log \frac{1}{t} & q = p \geq 1 \\ t^{-p}q\sqrt{\log d} & 1 \leq p \leq 2 \text{ and } q > 2 \\ t^{-2}qd^{1-\frac{2}{p}+\frac{2}{q}} & p, q \geq 2. \end{cases}$$

**Lower Bound** The following two lemmata, Lemma 9 and Lemma 10, are needed in the proof of our lower bounds for $p \leq 2$ and $p \geq 2$, respectively. Lemma 9 is a classical result, whose proof can be found, for example, in (Kiltz & Vaikuntanathan, 2022, p711).

**Lemma 9.** *Let $m$ be a positive integer. Suppose we have an $m$-dimensional vector whose entries are i.i.d. samples all drawn from either $\text{Ber}(\frac{1}{2} + \frac{1}{\sqrt{m}})$ or $\text{Ber}(\frac{1}{2} - \frac{1}{\sqrt{m}})$. It requires $\Omega(m)$ queries to distinguish $\text{Ber}(\frac{1}{2} + \frac{1}{\sqrt{m}})$ and $\text{Ber}(\frac{1}{2} - \frac{1}{\sqrt{m}})$ with probability at least $3/5$.*

**Lemma 10.** *Let $m$ be a positive integer. Suppose that $x \in \mathbb{R}^{2m}$ is a random vector in which all but one of the entries are the same and the distinct entry $x_{i^*}$ is located at a uniformly random position $i^* \in [2m]$. Any deterministic algorithm that determines with probability at least $3/5$ whether $i^*$ lies within $\{1, \ldots, m\}$ or $\{m+1, \ldots, 2m\}$ must read $\Omega(m)$ entries of $x$.*

*Proof.* Let $Q$ be the set of indices the algorithm reads and $\mathcal{A}(Q, i^*)$ be the output of the algorithm. Note that, if $i^* \notin Q$, then $\mathcal{A}(Q, i^*)$ does not depend on $i^*$ and we write it $\mathcal{A}(Q)$.

Now, let $\mathcal{E}$ be the event of $\mathcal{A}(Q, i^*)$ being the correct set and $\mathcal{I}$ be the event of $i^*$ being chosen among these $q$ queries. Then, we have

$$\Pr(\mathcal{E}) = \Pr(\mathcal{E} \mid \mathcal{I})\Pr(\mathcal{I}) + \Pr(\mathcal{E} \mid \overline{\mathcal{I}})\Pr(\overline{\mathcal{I}}).$$

Note that

$$\Pr(\mathcal{E} \mid \mathcal{I}) \leq 1, \quad \Pr(\mathcal{I}) = \frac{q}{2m} \quad \text{and} \quad \Pr(\overline{\mathcal{I}}) = 1 - \frac{q}{2m}.$$

where $q = |Q|$.

Now, we evaluate $\Pr(\mathcal{E} \mid \overline{\mathcal{I}})$. Let $q_1$ (resp. $q_2$) be the size of the set $Q \cap \{1, \ldots, m\}$ (resp. $Q \cap \{m+1, \ldots, 2m\}$), so $q_1 + q_2 = q$. Given the event $\overline{\mathcal{I}}$, recall that we have $\mathcal{A}(Q, i^*) = \mathcal{A}(Q)$. If $\mathcal{A}(Q) = \{1, \ldots, m\}$, the event of $\mathcal{E} \mid \overline{\mathcal{I}}$ is equivalent to the event that $i^*$ belongs to $\{1, \ldots, m\}$ but is not queried, thus we have

$$\Pr(\mathcal{E} \mid \overline{\mathcal{I}}) = 1 - \frac{m - q_1}{2m - q} = \frac{m - q_2}{2m - q}.$$

Similarly, if $\mathcal{A}(Q) = \{m+1, \ldots, 2m\}$, we have

$$\Pr(\mathcal{E} \mid \overline{\mathcal{I}}) = 1 - \frac{m - q_2}{2m - q} = \frac{m - q_1}{2m - q}.$$

Hence, we have

$$\Pr(\mathcal{E} \mid \overline{\mathcal{I}}) \leq \max\{\frac{m - q_2}{2m - q}, \frac{m - q_1}{2m - q}\}.$$

Namely, we have

$$\begin{aligned} \Pr(\mathcal{E}) &\leq \frac{q}{2m} + \max\{\frac{m - q_2}{2m - q}, \frac{m - q_1}{2m - q}\}(1 - \frac{q}{2m}) \\ &= \frac{q}{2m} + \max\{\frac{1}{2} - \frac{q_1}{2m}, \frac{1}{2} - \frac{q_2}{2m}\} \\ &= \frac{1}{2} + \max\{\frac{q_1}{2m}, \frac{q_2}{2m}\}. \end{aligned}$$

If $q \leq \frac{m}{5}$, it implies $\max\{q_1, q_2\} \leq \frac{m}{5}$ and hence $\Pr(\mathcal{E}) \leq \frac{3}{5}$. $\qquad\square$

---

**Algorithm 3** Algorithm for Active Learning

---

**Input:** a matrix $A \in \mathbb{R}^{n \times d}$
      a query access to the entries of the vector $b \in \mathbb{R}^n$
      a function $f \in \mathsf{Lip}_L$
      an error parameter $\varepsilon$
      a sampling rate $\alpha < 1$
1: Compute the Lewis weights $w_1, \ldots, w_n$ of $A$
2: **for** $i = 1, \ldots, n$ **do**
3:    $k_i \leftarrow \lceil \frac{n \cdot w_i}{d} \rceil$
4: $S \leftarrow \mathsf{GSM}(k_1, \ldots, k_n, \alpha)$ from Algorithm 1
5: Solve the minimization problem $\hat{x} := \arg\min_{x \in \mathbb{R}^d} \|Sf(Ax) - Sb\|_p^p + \varepsilon \|Ax\|_p^p$
6: Return the vector $\hat{x} \in \mathbb{R}^d$

---

The next lemma extends the previous two lemmata to multiple instances of the problem considered therein.

**Lemma 11.** *Let $d$ and $m$ be positive integers. Suppose that $D_0$ and $D_1$ are two distributions in $\mathbb{R}^m$ and distinguishing whether a vector is drawn from $D_0$ or $D_1$ with probability at least $3/5$ requires querying $\beta m$ entries of the vector for some constant $\beta > 0$. Consider a $dm$-dimensional random vector consisting of $d$ blocks, each of which is an $m$-dimensional vector drawn from either $D_0$ or $D_1$. Every deterministic algorithm that correctly distinguishes, with probability at least $2/3$, the distributions in $2d/3$ instances requires $\Omega(dm)$ entry queries to this $dm$-dimensional random vector.*

*Proof.* Suppose that an algorithm makes fewer than $\frac{\beta}{10} dm$ queries in total. Then there exist $\frac{9}{10} d$ blocks, each of which makes fewer than $\beta m$ queries. Therefore, each of these blocks will make an error in distinguishing the distributions with probability at least $2/5$. By a Chernoff bound, with probability at least $1/3$, at least $d' = \frac{2}{5} \cdot \frac{9}{10} \cdot d - \Theta(\sqrt{d})$ instances make an error. It means that $d' > d/3$ and we arrive at a contradiction against the assumption on the correctness of the algorithm. $\qquad\square$

## B   UPPER BOUND

In this section, we first obtain a query complexity with $\mathrm{poly} \log n$ factors with the algorithm presented in Algorithm 3 (see Theorem 12) and then remove the dependence on $n$ in Appendix B.4.

### B.1   EQUIVALENT STATEMENT

We shall first reduce the problem to the case where $A$ has uniformly bounded Lewis weights, before proving in the next section that the output of Algorithm 3 with a suitable $\alpha$ satisfies (20) with probability 0.99.

We start with the following observation. Let $k_i$ be $\lceil \frac{n \cdot w_i}{d} \rceil$ for $i = 1, \ldots, n$ which is the same term also defined in Algorithm 3. Hence, we rewrite

$$\|f(Ax) - b\|_p^p = \sum_{i=1}^{n} |(f(Ax) - b)_i|^p = \sum_{i=1}^{n} k_i \cdot \frac{1}{k_i} |(f(Ax) - b)_i|^p = \sum_{i=1}^{n} \sum_{j=1}^{k_i} \frac{1}{k_i} |(f(Ax) - b)_i|^p.$$

Now, suppose that we duplicate the $i$-th term, $|(f(Ax) - b)_i|^p$, $k_i$ times and assign a weight of $1/k_i^{\frac{1}{p}}$ to each duplicate term. Formally, let

- $n' = \sum_{i=1}^{n} k_i$,
- $A'$ be an $n'$-by-$d$ matrix in which $A'_{j,\cdot} = A_{i,\cdot}$ if $\sum_{a=1}^{i-1} k_a < j \leq \sum_{a=1}^{i} k_a$,
- $b'$ be an $n'$-dimensional vector in which $b'_j = b_i$ if $\sum_{a=1}^{i-1} k_a < j \leq \sum_{a=1}^{i} k_a$,
- $\Lambda$ be an $n'$-by-$n'$ diagonal matrix in which $\Lambda_{jj} = k_i^{-\frac{1}{p}}$ if $\sum_{a=1}^{i-1} k_a < j \leq \sum_{a=1}^{i} k_a$,

In other words, we have

$$\|f(Ax) - b\|_p^p = \sum_{j=1}^{n'} \Lambda_{jj}^p |(f(A'x) - b')_j|^p = \|\Lambda f(A'x) - \Lambda b'\|_p^p$$

Note that we still have

$$\mathsf{OPT} = \min_{x \in \mathbb{R}^d} \|\Lambda f(A'x) - \Lambda b'\|_p^p \quad \text{and} \quad x^* = \underset{\substack{x \in \mathbb{R}^d \\ \|\Lambda f(A'x) - \Lambda b'\|_p^p = \mathsf{OPT}}}{\arg\min} \|\Lambda A'x\|_p^p. \tag{17}$$

On the other hand, in Algorithm 3, recall that $N_i \sim \mathrm{Bin}(k_i, \alpha)$ for $i = 1, \dots, n$, it can be rewritten as

$$N_i = \sum_{j=1}^{k_i} N_{i,j},$$

where $N_{i,1}, \dots, N_{i,k_i}$ are i.i.d. $\mathrm{Ber}(\alpha)$ variables. In other words, we have

$$\|Sf(Ax) - Sb\|_p^p = \sum_{i=1}^{n} S_{ii}^p |(f(Ax) - b)_i|^p = \sum_{i=1}^{n} \sum_{j=1}^{k_i} \frac{N_{i,j}}{\alpha} \frac{1}{k_i} |(f(Ax) - b)_i|^p.$$

Let $S'$ be an $n'$-by-$n'$ diagonal matrix in which $S'_{jj} = \left(\frac{N_{i,j'}}{\alpha}\right)^{\frac{1}{p}}$ if $j = \sum_{a=1}^{i-1} k_a + j'$ for $j' = 1, \dots, k_i$. Then

$$\|Sf(Ax) - Sb\|_p^p = \sum_{j=1}^{n'} S_j {j'}^p \Lambda_{jj}^p |(f(A'x) - b')_j|^p = \|S'\Lambda f(A'x) - S'\Lambda b'\|_p^p.$$

Also, it is easy to check that

$$\|Ax\|_p^p = \|\Lambda A'x\|_p^p.$$

We still have

$$\hat{x} = \underset{x \in \mathbb{R}^d}{\arg\min} \|S'\Lambda f(A'x) - S'\Lambda b'\|_p^p + \varepsilon \|\Lambda A'x\|_p^p. \tag{18}$$

The advantage of introducing the diagonal matrix $\Lambda$ is to bound the Lewis weights. Formally, we have the following observation. By the definition of Lewis weights, the $j$-th Lewis weight of $\Lambda A'$ is $\frac{w_i}{k_i}$ if $j = \sum_{a=1}^{i-1} k_a + 1, \dots, \sum_{a=1}^{i} k_a$ for $j = 1, \dots, n'$. Recall that $k_i = \lceil \frac{n \cdot w_i}{d} \rceil$ and we have

$$\frac{w_i}{k_i} = \frac{w_i}{\lceil \frac{n \cdot w_i}{d} \rceil} \leq \frac{d}{n} \quad \text{and} \quad n' = \sum_{i=1}^{n} k_i \leq \sum_{i=1}^{n} \left(\frac{n \cdot w_i}{d} + 1\right) = 2n.$$

Therefore, we generalize our statement to be the following. Let $A'$ be an $n'$-by-$d$ matrix, $f \in \mathsf{Lip}_L$, $b'$ be an $n'$-dimensional vector, $\Lambda$ be an arbitrary $n'$-by-$n'$ positive diagonal matrix such that the Lewis weights of $\Lambda A'$ is at most $\frac{2d}{n'}$. Define $\mathsf{OPT}$ and $x^*$ as in (17). Furthermore, let $S'$ be an $n'$-by-$n'$ diagonal random matrix in which the diagonal entries are i.i.d. $\alpha^{-\frac{1}{p}} \cdot \mathrm{Ber}(\alpha)$ variables, i.e.

$$S'_{ii} = \begin{cases} \alpha^{-\frac{1}{p}} & \text{with probability } \alpha \\ 0 & \text{with probability } 1 - \alpha \end{cases}$$

and define $\hat{x}$ as in (18). Our goal is to show, for a suitable $\alpha$, we have in correspondence to (4)

$$\|\Lambda f(A'\hat{x}) - \Lambda b'\|_p^p \leq (1 + \varepsilon)\mathsf{OPT} + L^p \varepsilon \|\Lambda A'x^*\|_p^p.$$

### B.2 CORRECTNESS

We would like to prove that the output of Algorithm 3 satisfies (4) with probability 0.99. In view of Appendix B.1, we can replace $A$ with $\Lambda A$, where the Lewis weights of $\Lambda A$ are uniformly bounded by $2d/n$. The desired error guarantee is therefore

$$\|\Lambda f(A\hat{x}) - \Lambda b\|_p^p \le (1+\varepsilon)\mathsf{OPT} + L^p \varepsilon \|\Lambda A x^*\|_p^p. \tag{19}$$

By replacing $f(x)$ with $f(x)/L$ and $b$ with $b/L$, we can henceforce assume that $L = 1$ and the error guarantee (4) becomes

$$\|\Lambda f(A\hat{x}) - \Lambda b\|_p^p \le (1+\varepsilon)\mathsf{OPT} + \varepsilon \|\Lambda A x^*\|_p^p. \tag{20}$$

We shall first prove a weaker version of Theorem 1 with query complexity containing $\log n$ factors and then show how to remove the $\log n$ factors in Appendix B.4. The weaker version of Theorem 1 is stated formally below.

**Theorem 12.** *Let $A \in \mathbb{R}^{n \times d}$, $\bar{x} \in \mathbb{R}^d$, $b \in \mathbb{R}^n$, $f \in \mathsf{Lip}_1$, $\varepsilon \in (0,1)$ be sufficiently small and $\Lambda$ be an $n \times n$ diagonal matrix satisfying that $\Lambda_{ii} > 0$ and $w_i(\Lambda A) \lesssim d/n$ for all $i$. There is a randomized algorithm which, with probability at least $0.9$, makes $O\left(d^{1 \vee \frac{p}{2}}/\varepsilon^{2 \vee p} \cdot \mathsf{poly} \log n\right)$ queries to the entries of $b$ and returns an $\hat{x} \in \mathbb{R}^d$ satisfying*

$$\|\Lambda(f(A\hat{x}) - b)\|_p^p \le (1+\varepsilon)\|\Lambda(f(A\bar{x} - b)\|_p^p + \varepsilon\|\Lambda A\bar{x}\|_p^p.$$

*The hidden constant in the bound on number of queries depends on $p$ only.*

Note that we introduce a vector $\bar{x} \in \mathbb{R}^d$. If we take $\bar{x} = x^*$, Theorem 12 becomes Theorem 1 except that the query complexity contains $\log n$ factors. The reason we introduce $\bar{x}$ is because when we remove the $\log n$ factors in Appendix B.4 we can reuse the theorem using a different $\bar{x}$. Now, to prove Theorem 12, we first provide a concentration bound in Lemma 13.

**Lemma 13.** *Let $A \in \mathbb{R}^{n \times d}$, $f \in \mathsf{Lip}_1$, $\varepsilon \in (0,1)$ be sufficiently small and $\Lambda$ be an $n \times n$ diagonal matrix satisfying that $\Lambda_{ii} > 0$ and $w_i(\Lambda A) \lesssim d/n$ for all $i \in [n]$. Also, let $S$ be an $n$-by-$n$ random diagonal matrix in which the diagonal entries are i.i.d. $\alpha^{-\frac{1}{p}} \cdot \mathrm{Ber}(\alpha)$ variables where $\alpha \gtrsim \frac{d^{\frac{p}{2} \vee 1}}{n\varepsilon^{p \vee 2}} \cdot \mathsf{poly} \log n$. If $\hat{x}, \bar{x} \in \mathbb{R}^d$ satisfy*

$$\hat{x} = \underset{x \in \mathbb{R}^d}{\arg\min} \|S\Lambda(f(Ax) - b)\|_p^p + \varepsilon\|\Lambda Ax\|_p^p$$

*and*

$$\|\Lambda(f(A\bar{x}) - b)\|_p^p - \|\Lambda(f(A\hat{x}) - b)\|_p^p \lesssim \varepsilon(\|\Lambda(f(A\bar{x}) - b)\|_p^p + \varepsilon\|\Lambda A\bar{x}\|^p)$$

*then, with probability at least $0.9$,*

$$\left|(\|S\Lambda(f(A\hat{x}) - b)\|_p^p - \|S\Lambda(f(A\bar{x}) - b)\|_p^p) - (\|\Lambda(f(A\hat{x}) - b)\|_p^p - \|\Lambda(f(A\bar{x}) - b)\|_p^p)\right|$$
$$\le \varepsilon \cdot (\|\Lambda(f(A\bar{x}) - b)\|_p^p + \|\Lambda A\bar{x}\|_p^p).$$

We now show how Lemma 13 can be used to prove Theorem 12. The proof of Lemma 13 will be presented in Appendix B.3.

*Proof of Theorem 12.* We shall apply Lemma 13 with $\bar{x} = x^*$ to prove Theorem 12. First, we verify the conditions in Lemma 13. Clearly, the output $\hat{x}$ of Algorithm 3 satisfies

$$\hat{x} = \underset{x \in \mathbb{R}^d}{\arg\min} \|S\Lambda(f(Ax) - b)\|_p^p + \varepsilon\|\Lambda Ax\|_p^p$$

and, by the optimality of $x^*$, we also have

$$\|\Lambda(f(Ax^*) - b)\|_p^p - \|\Lambda(f(A\hat{x}) - b)\|_p^p \le 0.$$

Recall that $\|\Lambda(f(Ax^*) - b)\|_p^p = \mathsf{OPT}$. By Lemma 13, with probability at least $0.9$, we have

$$\left|(\|S\Lambda(f(A\hat{x}) - b)\|_p^p - \|S\Lambda(f(Ax^*) - b)\|_p^p) - (\|\Lambda(f(A\hat{x}) - b)\|_p^p - \mathsf{OPT})\right|$$
$$\le \varepsilon \cdot (\mathsf{OPT} + \|\Lambda Ax^*\|_p^p),$$

which implies that

$$\|\Lambda(f(A\hat{x}) - b)\|_p^p - \mathsf{OPT}$$
$$\leq \|S\Lambda(f(A\hat{x}) - b)\|_p^p - \|S\Lambda(f(Ax^*) - b)\|_p^p + \varepsilon \cdot (\mathsf{OPT} + \|\Lambda Ax^*\|_p^p)$$
$$\leq \varepsilon \cdot \|\Lambda Ax^*\|_p^p + \varepsilon \cdot (\mathsf{OPT} + \|\Lambda Ax^*\|_p^p) \quad \text{by the optimality of } \hat{x}$$
$$\lesssim \varepsilon \cdot (\mathsf{OPT} + \|\Lambda Ax^*\|_p^p).$$

This completes the proof of Theorem 12. $\qquad\square$

### B.3 CONCENTRATION BOUNDS

#### B.3.1 PROOF OF LEMMA 13

To prove Lemma 13, we rely on the following concentration bound provided in Lemma 14 and provide the proof in Appendix B.3.2.

**Lemma 14.** *Let $A \in \mathbb{R}^{n \times d}$, $f \in \mathsf{Lip}_1$, $\varepsilon \in (0, 1)$ be sufficiently small and $\Lambda$ be an $n \times n$ diagonal matrix satisfying that $\Lambda_{ii} > 0$ and $w_i(\Lambda A) \lesssim d/n$ for all $i \in [n]$. Additionally, suppose that $\bar{x} \in \mathbb{R}^d$ and $v \in \mathbb{R}^n$ are fixed vectors, $0 \leq \alpha \leq 1$, $R$ is any value that $R \geq \|\Lambda A\bar{x}\|_p^p$, $F$ is any value that $F \geq V := \|\Lambda(f(A\bar{x}) - v)\|_p^p$ and $T$ is any subset of $\mathbb{R}^d$ that $\{\bar{x}\} \subseteq T \subseteq \{x \in \mathbb{R}^d \mid \|\Lambda Ax\|_p^p \leq R\}$.*

*Let $S$ be an $n$-by-$n$ random diagonal matrix in which the diagonal entries are i.i.d. $\alpha^{-\frac{1}{p}} \cdot \mathrm{Ber}(\alpha)$ variables. When conditioned on the event that*

$$\|S\Lambda(f(A\bar{x}) - v)\|_p^p \lesssim V \quad \text{and} \quad \sup_{x \in T} \|S\Lambda(f(Ax) - f(A\bar{x}))\|_p^p \lesssim F,$$

*it holds with probability at least $1 - \delta$ that*

$$\sup_{x \in T} \left|\left(\|S\Lambda(f(Ax) - v)\|_p^p - \|S\Lambda(f(A\bar{x}) - v)\|_p^p\right) - \left(\|\Lambda(f(Ax) - v)\|_p^p - \|\Lambda(f(A\bar{x}) - v)\|_p^p\right)\right|$$

$$\leq C \cdot \left(\varepsilon V + \frac{d^{1 \vee \frac{p}{2}}}{\alpha n} R + \Gamma \cdot \left(\log^{\frac{5}{4}} d \sqrt{\log \frac{n}{\varepsilon d}} + \sqrt{\log \frac{1}{\delta}}\right)\right),$$

*where $C$ is an absolute constant and*

$$\Gamma = \begin{cases} (d/(\alpha n))^{\frac{1}{2}} F^{\frac{1}{2}} R^{\frac{1}{2}} & \text{when } 1 \leq p \leq 2 \\ (d^{\frac{p}{2}}/(\alpha n))^{\frac{1}{p}} F^{1 - \frac{1}{p}} R^{\frac{1}{p}} & \text{when } p > 2. \end{cases} \tag{21}$$

With Lemma 14, we immediately have the following two corollaries.

**Corollary 15.** *Let $A \in \mathbb{R}^{n \times d}$, $f \in \mathsf{Lip}_1$, $\Lambda \in \mathbb{R}^{n \times n}$, $\bar{x} \in \mathbb{R}^d$, $\alpha \in (0, 1)$, $R \in \mathbb{R}^d$, $T \subseteq \mathbb{R}^d$ and $S \in \mathbb{R}^{n \times n}$ be as defined in Lemma 14 and satisfy the same constraints. Additionally, suppose that $\alpha \gtrsim d^{\frac{p}{2} \vee 1}/n$. When conditioned on the event that $\|S\Lambda Ax\|_p^p \lesssim \|\Lambda Ax\|_p^p$ for all $x \in \mathbb{R}^d$, it holds with probability at least $1 - \delta$ that*

$$\sup_{x \in T} \left|\|S\Lambda(f(Ax) - f(A\bar{x}))\|_p^p - \|\Lambda(f(Ax) - f(A\bar{x}))\|_p^p\right|$$

$$\leq C \cdot \frac{d^{\frac{1}{2}}}{(\alpha n)^{\frac{1}{2 \vee p}}} R \cdot \left(\log^{\frac{5}{4}} d \sqrt{\log \frac{n}{d}} + \sqrt{\log \frac{1}{\delta}}\right),$$

*where $C$ is an absolute constant.*

*Proof.* In Lemma 14, we take $v = f(A\bar{x})$ and $\varepsilon$ to be a constant. Note that $V = \|S\Lambda(f(A\bar{x}) - v)\|_p^p = 0$. For any $x \in T$, if we take $F = 2^p R$ then

$$\|S\Lambda(f(Ax) - f(A\bar{x})))\|_p^p \leq \|S\Lambda(Ax - A\bar{x})\|_p^p \qquad \text{by the Lipschitz condition}$$
$$\lesssim \|\Lambda(Ax - A\bar{x})\|_p^p \qquad \text{by the assumption of } \|S\Lambda Ax\|_p^p \lesssim \|\Lambda Ax\|_p^p$$
$$\leq 2^p R \qquad \text{by } x, \bar{x} \in T$$

and hence the result follows by Lemma 14. $\qquad\square$

**Corollary 16.** *Let $A \in \mathbb{R}^{n \times d}$, $f \in \mathsf{Lip}_1$, $\varepsilon \in (0, 1)$, $\Lambda \in \mathbb{R}^{n \times n}$, $\bar{x} \in \mathbb{R}^d$, $\alpha \in (0, 1)$, $R \in \mathbb{R}$, $F \in \mathbb{R}$, $T \subseteq \mathbb{R}^d$ and $S \in \mathbb{R}^{n \times n}$ be as defined in Lemma 14 and satisfy the same constraints. Additionally, suppose that $\alpha \gtrsim \frac{d^{\frac{p}{2} \vee 1}}{n \varepsilon}$ and $F \gtrsim \varepsilon R$. When conditioned on the event that*

$$\|S\Lambda(f(A\bar{x}) - b)\|_p^p \lesssim \|\Lambda(f(A\bar{x}) - b)\|_p^p \quad and \quad \sup_{x \in T} \|S\Lambda(f(Ax) - f(A\bar{x}))\|_p^p \lesssim F,$$

*it holds with probability at least $1 - \delta$ that*

$$\sup_{x \in T} \left| (\|S\Lambda(f(Ax) - b)\|_p^p - \|S\Lambda(f(A\bar{x}) - b)\|_p^p) - (\|\Lambda(f(Ax) - b)\|_p^p - \|\Lambda(f(A\bar{x}) - b)\|_p^p) \right|$$

$$\leq C\,\Gamma \cdot \left( \log^{\frac{5}{4}} d \sqrt{\log \frac{n}{\varepsilon d}} + \sqrt{\log \frac{1}{\delta}} \right),$$

*where $C$ is an absolute constant and $\Gamma$ is as defined in (21).*

*Proof.* In Lemma 14, we take $v = b$. Note that we have $V = \|\Lambda(f(A\bar{x}) - b)\|_p^p$ and hence the result follows, noticing that the last term in the error bound of Lemma 14 is the dominating term. $\square$

Now, we are ready to complete the proof of Lemma 13.

*Proof of Lemma 13.* Without loss of generality, we can assume that $n \gtrsim d^{\frac{p}{2} \vee 1} / \varepsilon^{p \vee 2}$. We rely on Corollary 16 in this proof. To apply the corollary, we need to pick a suitable subset $T$ so that the output $\hat{x} \in T$. The set $T$ will be defined through suitable bounds for $R$ and $F$ and the main part of the proof will focus on obtaining these bounds.

Before doing so, we present some useful inequalities. First, by Markov inequality, with probability at least 0.99, we have

$$\|S\Lambda(f(A\bar{x}) - b)\|_p^p \leq 100 \|\Lambda(f(A\bar{x}) - b)\|_p^p. \tag{22}$$

We condition on this event in the remainder of the proof. By the optimality of $\hat{x}$, we have

$$\|S\Lambda(f(A\hat{x}) - b)\|_p^p + \varepsilon \|\Lambda A\hat{x}\|_p^p \leq \|S\Lambda(f(A\bar{x}) - b)\|_p^p + \varepsilon \|\Lambda A\bar{x}\|_p^p. \tag{23}$$

It implies that, by (22) and (23),

$$\|\Lambda A\hat{x}\|_p^p \leq \frac{1}{\varepsilon} \|S\Lambda(f(A\bar{x}) - b)\|_p^p + \|\Lambda A\bar{x}\|_p^p \leq \underbrace{\frac{100}{\varepsilon} \|\Lambda(f(A\bar{x}) - b)\|_p^p + \|\Lambda A\bar{x}\|_p^p}_{:=R_0} \tag{24}$$

Throughout the remainder of the proof, we assume that

$$\alpha \gtrsim \frac{d^{1 \vee \frac{p}{2}}}{n \varepsilon^{p \vee 2}} \operatorname{poly} \log n \quad \text{and} \quad \delta \sim \frac{1}{\log \log(1/\varepsilon)}$$

so that the error term in Corollary 16 can be upper bounded as

$$\Gamma \cdot \left( \operatorname{poly} \log n + \sqrt{\log(1/\delta)} \right) \lesssim \varepsilon F^\theta R^\beta,$$

where $\beta = \frac{1}{2} \wedge \frac{1}{p}$ and $\theta = (1 - \frac{1}{p}) \vee \frac{1}{2}$. Note that $\beta + \theta = 1$.

**Bounding $F$ in Corollary 16** We would like to first use Corollary 15 and let

$$T_{-1} = \{ x \in \mathbb{R}^d \mid \|\Lambda Ax\|_p^p \leq R_0 \}.$$

Now, we check the conditions. Our choice of $\alpha$ satisfies that $\alpha \gtrsim \frac{d^{1 \vee \frac{p}{2}}}{n} \operatorname{poly} \log n$, thus, by Lemma 6, $S$ is a constant-distortion subspace embedding for $\Lambda A$ with probability at least 0.99, i.e. $\|S\Lambda Ax\|_p \leq 2\|\Lambda Ax\|_p$ for all $x \in \mathbb{R}^d$. Recall that

$$R_0 = \frac{100}{\varepsilon} \|\Lambda(f(A\bar{x}) - b)\|_p^p + \|\Lambda A\bar{x}\|_p^p.$$

Hence, by Corollary 15 with our choice of $\alpha$ and $R = R_0$, it holds with probability 0.99 that

$$\sup_{x \in T_{-1}} \left| \|S\Lambda(f(Ax) - f(A\bar{x}))\|_p^p - \|\Lambda(f(Ax) - f(A\bar{x}))\|_p^p \right| \leq C_1 \varepsilon R_0, \tag{25}$$

where $C_1$ is a constant that depends only on $p$. Below we shall use $C_2, C_3, \ldots$ to denote constants that depend only on $p$. Conditioning on the event in (25), it follows that

$$\|\Lambda(f(A\hat{x}) - f(A\bar{x}))\|_p^p$$
$$\leq \|S\Lambda(f(A\hat{x}) - f(A\bar{x}))\|_p^p + C_1 \varepsilon R_0$$
$$\leq 2^p \left( \|S\Lambda(f(A\hat{x}) - b)\|_p^p + \|S\Lambda(f(A\bar{x}) - b)\|_p^p \right) + C_1 \varepsilon R_0$$
$$\overset{(A)}{\leq} 2^p \left( \|S\Lambda(f(A\bar{x}) - b)\|_p^p + \varepsilon\|\Lambda A\bar{x}\|_p^p + \|S\Lambda(f(A\bar{x}) - b)\|_p^p \right) + C_1 \varepsilon R_0$$
$$= 2^p \left( 2\|S\Lambda(f(A\bar{x}) - b)\|_p^p + \varepsilon\|\Lambda A\bar{x}\|_p^p \right) + C_1 \varepsilon R_0$$
$$\overset{(B)}{\leq} 2^p \left( 2 \cdot 100\|\Lambda(f(A\bar{x}) - b)\|_p^p + \varepsilon\|\Lambda A\bar{x}\|_p^p \right) + C_1 \varepsilon \left( \frac{100}{\varepsilon}\|\Lambda(f(A\bar{x}) - b)\|_p^p + \|\Lambda A\bar{x}\|_p^p \right)$$
$$\leq C_2 (\|\Lambda(f(A\bar{x}) - b)\|_p^p + \varepsilon\|\Lambda A\bar{x}\|_p^p) \quad \text{for some large constant } C_2, \tag{26}$$

where (A) is due to (23), the optimality of $\hat{x}$, and (B) to (22), the Markov inequality for $\|\Lambda(f(A\bar{x}) - b)\|_p^p$, and the definition of $R_0$.

Define $F_0$ to be the RHS of (26), i.e.

$$F_0 := C_2 (\|\Lambda(f(A\bar{x}) - b)\|_p^p + \varepsilon\|\Lambda A\bar{x}\|_p^p).$$

We preview that the set $T$ we use in Corollary 16 contains the element $x$ satisfying the inequality

$$\|\Lambda(f(Ax) - f(A\bar{x}))\|_p^p \leq F_0$$

Hence, (26) suggests that $\hat{x}$ is in the domain of interest and hence $F_0$ is the bound we will use in Corollary 16.

**Bounding $R$ in Corollary 16** Now, we would like to use Corollary 16. Recall that

$$R_0 = \frac{100}{\varepsilon}\|\Lambda(f(A\bar{x}) - b)\|_p^p + \|\Lambda A\bar{x}\|_p^p.$$

We can apply Corollary 16 with $R_0$ but it will give a weaker result. However, we shall still use this weaker result and improve the bounds iteratively. Specifically, we shall define $R_i$ based on $R_{i-1}$, ensuring that $R_i \leq R_0$ and that each $R_i$ has the form of $X_i\|\Lambda(f(A\bar{x}) - b)\|_p^p + Y_i\|\Lambda A\bar{x}\|_p^p$ for some $X_i, Y_i \geq 1$ (for example, $X_0 = \frac{100}{\varepsilon}$ and $Y_0 = 1$). Furthermore, let

$$T_i = \{ x \in \mathbb{R}^d \mid \|\Lambda Ax\|_p^p \leq R_i \text{ and } \|\Lambda(f(Ax) - f(A\bar{x}))\|_p^p \leq F_0 \},$$

so that $T_i \subseteq T_0$. More specifically, we shall use $T_i$ to estimate an upper bound of $\|\Lambda A\hat{x}\|_p^p$ and define $R_{i+1}$ based on the upper bound, ensuring that $\bar{x} \in T_i$. This guarantees that $T_i$ satisfies the subset condition in Corollary 16. We shall also verify other conditions of Corollary 16.

It is clear that $R_i \leq R_0 \lesssim \frac{1}{\varepsilon}F_0$. By (22), we have $\|S\Lambda(f(A\bar{x}) - b)\|_p^p \lesssim \|\Lambda(f(A\bar{x}) - b)\|_p^p$ and, by (25) and the fact that $T_i \subseteq T_{-1}$, we have

$$\sup_{x \in T_i} \|S\Lambda(f(Ax) - f(A\bar{x}))\|_p^p \leq \sup_{x \in T_i} \|\Lambda(f(Ax) - f(A\bar{x}))\|_p^p + C_1 \varepsilon R_0 \lesssim F_0.$$

We invoke Corollary 16 with our choice of $\alpha$, $R = R_i$ and $F = F_0$. Hence, with probability $1 - \delta$,

$$\sup_{x \in T_i} \left| (\|S\Lambda(f(Ax) - b)\|_p^p - \|S\Lambda(f(A\bar{x}) - b)\|_p^p) - (\|\Lambda(f(Ax) - b)\|_p^p - \|\Lambda(f(A\bar{x}) - b)\|_p^p) \right|$$
$$\leq C_3 \cdot \varepsilon R_i^\beta F_0^\theta \quad \text{for some constant } C_3. \tag{27}$$

To use (27), we would like to argue that the solution $\hat{x} \in T_i$. For $T_0$, we have

$$\|\Lambda A\hat{x}\|_p^p \leq R_0 \quad \text{by (24) and} \quad \|\Lambda(f(A\hat{x}) - f(A\bar{x}))\|_p^p \quad \text{by (26)}$$

and hence $\hat{x} \in T_0$. From now on, suppose that $\hat{x} \in T_i$ and we will argue that $\hat{x} \in T_{i+1}$.

We continue to bound (27). Assume that $KY_i/X_i \geq \varepsilon$ for some $K \geq 1$, then we can upper bound $R_i^\beta F_0^\theta$ as follows.

$$R_i^\beta F_0^\theta = \left(X_i\|\Lambda(f(A\bar{x}) - b)\|_p^p + Y_i\|\Lambda A\bar{x}\|_p^p\right)^\beta \cdot C_2^\theta \left(\|\Lambda(f(A\bar{x}) - b)\|_p^p + \varepsilon\|\Lambda A\bar{x}\|_p^p\right)^\theta$$

$$\leq C_2^\theta \left(X_i\|\Lambda(f(A\bar{x}) - b)\|_p^p + Y_i\|\Lambda A\bar{x}\|_p^p\right)^\beta \left(\|\Lambda(f(A\bar{x}) - b)\|_p^p + \frac{KY_i}{X_i}\|\Lambda A\bar{x}\|_p^p\right)^\theta$$

$$\leq \left(\frac{C_2}{X_i}\right)^\theta \left(X_i\|\Lambda(f(A\bar{x}) - b)\|_p^p + KY_i\|\Lambda A\bar{x}\|_p^p\right) \quad \text{note that } \beta + \theta = 1. \tag{28}$$

Thus,

$$\|\Lambda A\hat{x}\|_p^p \leq \frac{1}{\varepsilon} \left(\|S\Lambda(f(A\bar{x}) - b)\|_p^p - \|S\Lambda(f(A\hat{x}) - b)\|_p^p\right) + \|\Lambda A\bar{x}\|_p^p$$

$$\leq \frac{1}{\varepsilon} \cdot \left(\|\Lambda(f(A\bar{x}) - b)\|_p^p - \|\Lambda(f(A\hat{x}) - b)\|_p^p + C_3 \cdot \varepsilon R_i^\beta F_0^\theta\right) + \|\Lambda A\bar{x}\|_p^p \quad \text{by (27)}$$

From our assumption, we have

$$\|\Lambda(f(A\bar{x}) - b)\|_p^p - \|\Lambda(f(A\hat{x}) - b)\|_p^p$$

$$\lesssim \varepsilon(\|\Lambda(f(A\bar{x}) - b)\|_p^p + \varepsilon\|\Lambda A\bar{x}\|^p)$$

$$\lesssim \varepsilon R_i^\beta F_0^\theta \quad \text{by the definition of } F_0 \text{ and } F_0 \lesssim R_i.$$

In other words, we have

$$\|\Lambda A\hat{x}\|_p^p \leq C_4 \cdot (R_i^\beta F_0^\theta) + \|\Lambda A\bar{x}\|_p^p \quad \text{for some constant } C_4$$

$$\leq C_4 \left(\frac{C_2}{X_i}\right)^\theta \left(X_i\|\Lambda(f(A\bar{x}) - b)\|_p^p + KY_i\|\Lambda A\bar{x}\|_p^p\right) + \|\Lambda A\bar{x}\|_p^p$$

$$\leq X_{i+1}\|\Lambda(f(A\bar{x}) - b)\|_p^p + Y_{i+1}\|\Lambda A\bar{x}\|_p^p, \tag{29}$$

where

$$X_{i+1} = C_4 C_2^\theta X_i^{1-\theta} \quad \text{and} \quad Y_{i+1} = 1 + \frac{C_4 C_2^\theta KY_i}{X_i^\theta}.$$

Define $R_{i+1}$ to be the minimum of $R_0$ and the expression in (29), i.e.

$$R_{i+1} := R_0 \wedge (X_{i+1}\|\Lambda(f(A\bar{x}) - b)\|_p^p + Y_{i+1}\|\Lambda A\bar{x}\|_p^p).$$

We immediately have $\hat{x} \in T_{i+1}$, which is needed to iterate the argument.

Let $X_0 = 100/\varepsilon$ and $Y_0 = 1$. By induction, one can show that

$$X_i = C_5^{\frac{1-(1-\theta)^i}{\theta}} \left(\frac{100}{\varepsilon}\right)^{(1-\theta)^i}$$

for $C_5 = C_4 C_2^\theta$. Then $C_6 \leq X_i \leq 100 C_5^{1/\theta}/\varepsilon$ for some constant $C_6$ for all $i \leq r$, thus $Y_{i+1} \leq 1 + C_7 Y_i \leq C_8 Y_i$ for some constants $C_7$ and $C_8$. When $r \sim_p \log\log(1/\varepsilon)$, we have

$$X_r \leq C_9 \quad \text{and} \quad Y_r \leq C_4(C_8)^{r-1} = \text{poly}\log\frac{1}{\varepsilon}.$$

We shall also verify that $KY_i/X_i \geq \varepsilon$ for some $\varepsilon$. Indeed, $Y_i/X_i \geq 1/(100 C_5^{1/\theta}/\varepsilon) \geq \varepsilon/K$ for $K = 100 C_5^{1/\theta}$.

Iterating the argument above $r$ times. The total failure probability is at most $\delta r + 0.03 = 0.1$ since $\delta \sim 1/r$. It then follows from (27) with $i = r - 1$ that

$$\|\Lambda(f(A\hat{x}) - b)\|_p^p - \|\Lambda(f(A\bar{x}) - b)\|_p^p$$

$$\leq \|S\Lambda(f(A\hat{x}) - b)\|_p^p - \|S\Lambda(f(A\bar{x}) - b)\|_p^p + C_3 \cdot \varepsilon R_{r-1}^\beta F_0^\gamma$$

$$\leq \varepsilon\|\Lambda A\bar{x}\|_p^p + C_3 \cdot \varepsilon R_{r-1}^\beta F_0^\gamma \quad \text{by the optimality of } \hat{x}$$

$$\lesssim \varepsilon\|\Lambda A\bar{x}\|_p^p + \varepsilon(X_r\|\Lambda(f(A\bar{x}) - b)\|_p^p + Y_r\|\Lambda A\bar{x}\|_p^p) \quad \text{by (28)}$$

$$\lesssim \varepsilon\|\Lambda(f(A\bar{x}) - b)\|_p^p + \left(\varepsilon\,\text{poly}\log\frac{1}{\varepsilon}\right)\|\Lambda A\bar{x}\|_p^p.$$

Rescaling $\varepsilon\,\text{poly}\log\frac{1}{\varepsilon}$ to $\varepsilon$ proves the claimed result of the theorem. $\qquad\square$

### B.3.2 PROOF OF LEMMA 14

In this section, we will prove Lemma 14. Recall that $\bar{x}$ is an arbitrary fixed vector in $\mathbb{R}^d$, $v$ is an arbitrary fixed vector in $\mathbb{R}^n$, $V = \|\Lambda(f(A\bar{x}) - v)\|_p^p$, $R \geq \|\Lambda A\bar{x}\|_p^p$, $F \geq V$ and $\{\bar{x}\} \subseteq T \subseteq \{x \in \mathbb{R}^d \mid \|\Lambda Ax\|_p^p \leq R\}$. Also, we would like to bound the following expression

$$\sup_{x \in T} \left| (\|S\Lambda(f(Ax) - v)\|_p^p - \|S\Lambda(f(A\bar{x}) - v)\|_p^p) - (\|\Lambda(f(Ax) - v)\|_p^p - \|\Lambda(f(A\bar{x}) - v)\|_p^p) \right|,$$

which can be written as

$$\sup_{x \in T} \left| \sum_{i=1}^n (S_{ii}^p - 1)\Lambda_{ii}^p (|(f(Ax) - v)_i|^p - |(f(A\bar{x}) - v)_i|^p) \right|. \tag{30}$$

We shall bound (30) from above by, up to a constant factor,

$$\varepsilon V + \frac{d^{1 \vee \frac{p}{2}}}{\alpha n} R + \Gamma \cdot \left( \text{poly} \log n + \sqrt{\log \frac{1}{\delta}} \right) \tag{31}$$

with probability $1 - \delta$. Recall that, as defined in (21),

$$\Gamma = \begin{cases} (d/(\alpha n))^{\frac{1}{2}} F^{\frac{1}{2}} R^{\frac{1}{2}} & \text{when } 1 \leq p \leq 2 \\ (d^{\frac{p}{2}}/(\alpha n))^{\frac{1}{p}} F^{1 - \frac{1}{p}} R^{\frac{1}{p}} & \text{when } p > 2. \end{cases}$$

We preview that the first term $\varepsilon V$ comes from Lemma 18, the second term $\frac{d^{1 \vee \frac{p}{2}} \varepsilon^p}{\alpha n} R$ from Lemma 19 and the third term $\Gamma \cdot (\text{poly} \log n + \sqrt{\log \frac{1}{\delta}})$ from Dudley's integral (Lemma 7). We first present a useful lemma.

**Lemma 17.** *For any $R \geq \|\Lambda A\bar{x}\|_p^p$, let $T$ be a set that $\{\bar{x}\} \subseteq T \subseteq \{x \in \mathbb{R}^d \mid \|\Lambda Ax\|_p^p \leq R\}$. Also, let $w_1, \ldots, w_n$ be the Lewis weights of $\Lambda A$. It holds for all $x \in T$ and $i \in [n]$ that*

$$|\Lambda_{ii}(f(Ax) - f(A\bar{x}))_i| \leq 2d^{\frac{1}{2} - \frac{1}{2 \vee p}} w_i^{\frac{1}{p}} R^{\frac{1}{p}}.$$

*Proof.* Note that

$$|\Lambda_{ii}(f(Ax) - f(A\bar{x}))_i| \leq |\Lambda_{ii}(Ax - A\bar{x})_i| \qquad \text{by the Lipschitz condition}$$
$$\leq d^{\frac{1}{2} - \frac{1}{2 \vee p}} w_i^{\frac{1}{p}} \|\Lambda Ax - \Lambda A\bar{x}\|_p \qquad \text{by Lemma 5(d)}$$

Since $x, \bar{x}$ are both in $T$, we have

$$\|\Lambda Ax - \Lambda A\bar{x}\|_p \leq \|\Lambda Ax\|_p + \|\Lambda A\bar{x}\|_p \leq 2R^{\frac{1}{p}}.$$

The desired result follows. $\qquad\square$

Define the set $G$ of "good" indices to be

$$G := \left\{ i \in [n] \; \middle| \; |\Lambda_{ii}(f(A\bar{x}) - v)_i| \leq \frac{d^{\frac{1}{2} - \frac{1}{2 \vee p}} w_i^{\frac{1}{p}} R^{\frac{1}{p}}}{\varepsilon} \right\}. \tag{32}$$

We shall first take care of the terms with "bad" indices in (30), i.e. the indices *not* in $G$, and hence obtain the first term $\varepsilon V$ in (31). We highlight that only the property of the nonnegativity of the diagonal entries of $S$ is used in Lemma 18.

**Lemma 18.** *For any $R \geq \|\Lambda A\bar{x}\|_p^p$ and $\varepsilon > 0$, let $T$ be a set that $\{\bar{x}\} \subseteq T \subseteq \{x \in \mathbb{R}^d \mid \|\Lambda Ax\|_p^p \leq R\}$ and $G$ be the set defined in (32). Suppose that $S$ is an $n$-by-$n$ diagonal matrix with nonnegative diagonal entries and*

$$\|S\Lambda(f(A\bar{x}) - v)\|_p^p \lesssim V,$$

*where $V = \|\Lambda(f(A\bar{x}) - v)\|_p^p$. Then, we have*

$$\sup_{x \in T} \left| \sum_{i \notin G} (S_{ii}^p - 1)\Lambda_{ii}^p \big(|(f(Ax) - v)_i|^p - |(f(A\bar{x}) - v)_i|^p\big) \right| \lesssim \varepsilon V.$$

*Proof.* To ease the notations, let

$$u_x := f(Ax) \quad \text{for all } x \in \mathbb{R}^d \quad \text{and} \quad \lambda_i := \Lambda_{ii} \quad \text{for } i \in [n].$$

Note that by the triangle inequality,

$$\left| \sum_{i \notin G} (S_{ii}^p - 1)\lambda_i^p \big(|(u_x - v)_i|^p - |(u_{\bar{x}} - v)_i|^p\big) \right| \le \sum_{i \notin G} (S_{ii}^p + 1)\lambda_i^p \big||(u_x - v)_i|^p - |(u_{\bar{x}} - v)_i|^p\big|$$

Furthermore, by the inequality $\big||a|^p - |b|^p\big| \le p|a - b| \big||a|^{p-1} + |b|^{p-1}\big|$, we have

$$\lambda_i^p \big||(u_x - v)_i|^p - |(u_{\bar{x}} - v)_i|^p\big| \le p|\lambda_i(u_x - u_{\bar{x}})_i| \cdot \big||\lambda_i(u_x - v)_i|^{p-1} + |\lambda_i(u_{\bar{x}} - v)_i|^{p-1}\big|.$$

For any $i \notin G$ and $x \in T$, by Lemma 17 and the definition of $G$, we have

$$|\lambda_i(u_x - u_{\bar{x}})_i| \le 2\varepsilon|\lambda_i(u_{\bar{x}} - v)_i|,$$
$$|\lambda_i(u_x - v)_i| \le |\lambda_i(u_x - u_{\bar{x}})_i| + |\lambda_i(u_{\bar{x}} - v)_i| \le (1 + 2\varepsilon)|\lambda_i(u_{\bar{x}} - v)_i|.$$

It follows that

$$\lambda_i^p \big||(u_x - v)_i|^p - |(u_{\bar{x}} - v)_i|^p\big| \lesssim \varepsilon|\lambda_i(u_{\bar{x}} - v)_i|^p.$$

which implies that

$$\sup_{x \in T}\left| \sum_{i \notin G} (S_{ii}^p - 1)\lambda_i^p\big(|(u_x - v)_i|^p - |(u_{\bar{x}} - v)_i|^p\big) \right|$$

$$\lesssim \varepsilon \sum_{i \notin G} (S_{ii}^p + 1)|\lambda_i(u_{\bar{x}} - v)_i|^p \le \varepsilon \sum_{i=1}^n (S_{ii}^p + 1)|\lambda_i(u_{\bar{x}} - v)_i|^p \lesssim \varepsilon \cdot V,$$

where we used the assumption of the lemma in the last step.

$\square$

Now, we also define the set of indices whose term has a high Lewis weight within $G$. Let

$$J := \left\{ i \in G \ \middle| \ w_i > \frac{\varepsilon^p d}{n^2} \right\}. \tag{33}$$

We now take care of the terms with low Lewis weights in (30), i.e. the indices $i \notin J$, and hence obtain the second term $\frac{d^{1 \vee \frac{p}{2}}\varepsilon^p}{\alpha n}R$ in (31). We highlight that only the property of the diagonal entries of $S$ being in $[0, \frac{1}{\alpha}]$ is used in Lemma 19.

**Lemma 19.** *For any $R \ge \|\Lambda A\bar{x}\|_p^p$ and $\varepsilon > 0$, let $T$ be a set that $\{\bar{x}\} \subseteq T \subseteq \{ x \in \mathbb{R}^d \mid \|\Lambda Ax\|_p^p \le R \}$ and $J$ be the set defined in (33). Suppose $S$ is an $n$-by-$n$ diagonal matrix whose entries satisfy $0 \le S_{ii}^p \le \frac{1}{\alpha}$ for any $\alpha > 0$. Then, we have*

$$\sup_{x \in T}\left| \sum_{i \in G \setminus J} (S_{ii}^p - 1)\Lambda_{ii}^p\big(|(f(Ax) - v)_i|^p - |(f(A\bar{x}) - v)_i|^p\big) \right| \lesssim \frac{d^{1 \vee \frac{p}{2}}}{\alpha n}R.$$

*Proof.* To ease the notations, let

$$u_x := f(Ax) \quad \text{for all } x \in \mathbb{R}^d \quad \text{and} \quad \lambda_i := \Lambda_{ii} \quad \text{for } i \in [n].$$

Note that by the triangle inequality,

$$\left| \sum_{i \in G \setminus J} (S_{ii}^p - 1)\lambda_i^p\big(|(u_x - v)_i|^p - |(u_{\bar{x}} - v)_i|^p\big) \right| \le \sum_{i \in G \setminus J} (S_{ii}^p + 1)\lambda_i^p\big||(u_x - v)_i|^p - |(u_{\bar{x}} - v)_i|^p\big|.$$

Since $i \in G$, by Lemma 17, we have

$$|\lambda_i(u_x - v)_i| \le |\lambda_i(u_x - u_{\bar{x}})_i| + |\lambda_i(u_{\bar{x}} - v)_i| \le (2 + \frac{1}{\varepsilon})d^{\frac{1}{2} - \frac{1}{2 \vee p}} w_i^{\frac{1}{p}} R^{\frac{1}{p}},$$

which, together with $i \notin J$, implies that

$$|\lambda_i(u_x - v)_i|^p \lesssim \frac{d^{(\frac{p}{2} \vee 1) - 1}}{\varepsilon^p} w_i R \lesssim \frac{d^{1 \vee \frac{p}{2}}}{n^2} R.$$

Recall that we assume $S_{ii}^p \le \frac{1}{\alpha}$. Therefore,

$$\sup_{x \in T} \left| \sum_{i \in G \setminus J} (S_{ii}^p - 1)\lambda_i^p \big( |(u_x - v)_i|^p - |(u_{\bar{x}} - v)_i|^p \big) \right| \lesssim \sum_{i \in G \setminus J} (\frac{1}{\alpha} + 1) \cdot \frac{d^{1 \vee \frac{p}{2}}}{n^2} R \lesssim \frac{d^{1 \vee \frac{p}{2}}}{\alpha n} R. \quad \square$$

With Lemma 18 and Lemma 19, we only need to take care of the indices in $J$. Namely the set of indices $i$ such that

$$|(f(A\bar{x}) - v)_i| \le \frac{d^{\frac{1}{2} - \frac{1}{2 \vee p}} w_i^{\frac{1}{p}} R^{\frac{1}{p}}}{\varepsilon} \quad \text{and} \quad w_i > \frac{\varepsilon^p d}{n^2}.$$

Now, we would like to bound the following expression

$$\sup_{x \in T} \left| \sum_{i \in J} (S_{ii}^p - 1)\Lambda_{ii}^p \big( |(f(Ax) - v)_i|^p - |(f(A\bar{x}) - v)_i|^p \big) \right|.$$

We consider bounding its $\ell$-th moment and then apply Markov's inequality for some $\ell$ to be determined later. To that end, consider

$$\Theta_S := \sup_{x \in T} \left| \sum_{i \in J} (S_{ii}^p - 1)\Lambda_{ii}^p \big( |(f(Ax) - v)_i|^p - |(f(A\bar{x}) - v)_i|^p \big) \right|$$

By the standard symmerization trick, we have

$$\mathbb{E}_S \Theta_S^\ell \le 2^\ell \mathbb{E}_{\xi, S} \left( \sup_{x \in T} \left| \sum_{i \in J} \xi_i \cdot S_{ii}^p \Lambda_{ii}^p \big( |(f(Ax) - v)_i|^p - |(f(A\bar{x}) - v)_i|^p \big) \right| \right)^\ell, \quad (34)$$

where $\xi$ is a $|J|$-dimensional vector whose entries are independent Rademacher random variable, i.e. each $\xi_i$ is uniform on $\{-1, 1\}$.

Next, we condition on $S$. Recall that $S_{ii}^p$ is either $\frac{1}{\alpha}$ or $0$, let $I \subseteq J$ be the set of indices $i$ such that $S_{ii}^p = \frac{1}{\alpha}$. For any $x \in \mathbb{R}^d$, we define $z_x$ to be the $n$-dimensional vector whose $i$-th entry is

$$(z_x)_i := |\Lambda_{ii}(f(Ax) - v)_i|^p - |\Lambda_{ii}(f(A\bar{x}) - v)_i|^p \quad (35)$$

Also, we define a pseudometric $\rho$ to be

$$\rho(x, x') := \|(z_x)_I - (z_{x'})_I\|_2 \quad \text{for any } x, x' \in \mathbb{R}^d$$

$$= \left( \sum_{i \in I} \big( |\Lambda_{ii}(f(Ax) - v)_i|^p - |\Lambda_{ii}(f(Ax') - v)_i|^p \big)^2 \right)^{1/2}$$

Recall that $(\cdot)_I$ means we shrink the vector by only retaining the entries whose index is in $I$. Now, in order to upper bound the right-hand side of (34), we seek to upper bound

$$\mathbb{E}_\xi \left( \sup_{x \in T} |\langle \xi_I, (z_x)_I \rangle| \right)^\ell.$$

Since $\bar{x} \in T$, the $\ell$-moment of the supremum can be upper bounded using Dudley's integral (Lemma 7) as

$$\mathbb{E}_\xi \left( \sup_{x \in T} |\langle \xi_I, (z_x)_I \rangle| \right)^\ell \lesssim C^\ell \left[ \left( \int_0^{\text{Diam}(T, \rho)} \sqrt{\log \mathcal{N}(T, \rho, r)} dr \right)^\ell + (\sqrt{\ell} \, \text{Diam}(T, \rho))^\ell \right].$$

(36)

Recall that $\mathcal{N}(T, \rho, r)$ is the covering number of $T$ w.r.t. $\rho$ and $r$. We will prove in Appendix B.3.4 that

$$\int_0^{\text{Diam}(T,\rho)} \sqrt{\log \mathcal{N}(T, \rho, r)} dr \lesssim \alpha \cdot \Gamma \log^{\frac{5}{4}} d \sqrt{\log \frac{n}{\varepsilon d}} \quad \text{and} \quad \text{Diam}(T, \rho) \lesssim \alpha \cdot \Gamma \qquad (37)$$

where

$$\Gamma = \left( \frac{d^{\frac{p}{2} \vee 1} R F^{(p-1) \vee 1}}{\alpha n} \right)^{\frac{1}{p} \wedge \frac{1}{2}}.$$

Taking expectation over $S$, it follows from (34), (36) and (37) that

$$\mathbb{E}_S \Theta_S^\ell \lesssim (C'\Gamma)^\ell \left( \left( \log^{\frac{5}{4}} d \sqrt{\log \frac{n}{\varepsilon d}} \right)^\ell + \sqrt{\ell}^\ell \right).$$

Take $\ell = \log(1/\delta)$. By Markov inequality, it holds with probability $1 - \delta$ that

$$\Theta_S = \sup_{x \in T} \left| \sum_{i \in J} \xi_i \cdot S_{ii}^p \Lambda_{ii}^p \left( |(u_x - v)_i|^p - |(u_{\bar{x}} - v)_i|^p \right) \right| \lesssim \Gamma \cdot \left( \log^{\frac{5}{4}} d \sqrt{\log \frac{n}{\varepsilon d}} + \sqrt{\log \frac{1}{\delta}} \right).$$

Note that it is the third term in (31). Combining Lemmas 18 and 19 proves Lemma 14.

### B.3.3 DIAMETER ESTIMATES

In order to bound Dudley's integral in (37), we need to bound the covering number $\mathcal{N}(T, \rho, r)$. To this end, we shall bound the metric, $\rho$, and the diameter $\text{Diam}(T, \rho)$. The proof imitates the proofs in earlier works, e.g., Ledoux & Talagrand (1991); Huang et al. (2024); Gajjar et al. (2024), on subspace embeddings and active regression problems.

**Lemma 20.** *Let $A \in \mathbb{R}^{n \times d}$, $\bar{x} \in \mathbb{R}^d$, $v \in \mathbb{R}^n$, $f \in \mathsf{Lip}_1$, $\Lambda \in \mathbb{R}^{n \times n}$, $\alpha \in [0, 1]$ and $S \in \mathbb{R}^{n \times n}$ be as defined in Lemma 14 and satisfy the same constraints. Suppose that $0 \le \alpha \le 1$, $R \ge \|\Lambda A \bar{x}\|_p^p$ and $F \ge \|\Lambda (f(A\bar{x}) - v)\|_p^p$. Let $T$ be a set that $\{\bar{x}\} \subseteq T \subseteq \{ x \in \mathbb{R}^d \mid \|\Lambda A x\|_p^p \le R \}$. If $I$ is a subset of $J$ such that*

$$\|(\Lambda (f(A\bar{x}) - v))_I\|_p^p \lesssim \alpha \cdot \|\Lambda (f(A\bar{x}) - v)\|_p^p \quad \text{and} \quad \sup_{x \in T} \|(\Lambda (f(Ax) - f(A\bar{x})))_I\|_p^p \le \alpha \cdot F$$

*then, for any $x, x' \in T$ and $q = \log(\frac{n}{\varepsilon d})$, we have*

$$\rho(x, x') \lesssim K \cdot \left( \left\| W^{-\frac{1}{p}} \Lambda A (x - x') \right\|_{w,q}^{\frac{p}{2} \wedge 1} \wedge d^{\frac{1}{2} - \frac{1}{2 \vee p}} \left\| W^{-\frac{1}{p}} \Lambda A (x - x') \right\|_{w,p}^{\frac{p}{2} \wedge 1} \right)$$

*and*

$$\text{Diam}(T, \rho) \lesssim d^{\frac{1}{2} - \frac{1}{2 \vee p}} K R^{\frac{1}{2} \wedge \frac{1}{p}},$$

*where $W = \text{diag}\{w_1, \ldots, w_n\}$ and*

$$K = \begin{cases} \sqrt{\alpha d F / n} & \text{for } 1 \le p \le 2; \\ (\alpha^{p-1} d F^{p-1} / n)^{1/p} & \text{for } p > 2. \end{cases}$$

*Proof.* As in the proof of Lemma 18, we let $u_x = f(Ax)$ and $\lambda_i = \Lambda_{ii}$ to simplify the notation. We further define semi-norms $\|u\|_{I,\infty} := \max_{i \in I} |u_i|$ and $\|u\|_{I,p} = (\sum_{i \in I} |u_i|^p)^{1/p}$.

For $i \in I$ and $x, x' \in T$, we have

$$|(z_x)_i - (z_{x'})_i| \le ||\lambda_i (u_x - v)_i|^p - |\lambda_i (u_{x'} - v)_i|^p|$$
$$\le p |\lambda_i (u_x - u_{x'})_i| \cdot \left( |\lambda_i (u_x - v)_i|^{p-1} + |\lambda_i (u_{x'} - v)_i|^{p-1} \right)$$

where the first inequality is due to the definition in (35) and the second to the fact that $||a|^p - |b|^p| \le p|a - b| \cdot ||a|^{p-1} + |b|^{p-1}|$. It follows that

$$\rho(x, x')^2 \le \sum_{i \in I} p^2 |\lambda_i (u_x - u_{x'})_i|^2 \left( |\lambda_i (u_x - v)_i|^{p-1} + |\lambda_i (u_{x'} - v)_i|^{p-1} \right)^2$$

$$\lesssim \sum_{i \in I} |\lambda_i (u_x - u_{x'})_i|^2 \left( |\lambda_i (u_x - v)_i|^{2p-2} + |\lambda_i (u_{x'} - v)_i|^{2p-2} \right). \qquad (38)$$

When $1 \leq p \leq 2$, we further bound (38) by

$$\sum_{i \in I} |\lambda_i (u_x - u_{x'})_i|^2 \left( |\lambda_i (u_x - v)_i|^{2p-2} + |\lambda_i (u_{x'} - v)_i|^{2p-2} \right)$$

$$\leq \max_{i \in I} \{ |\lambda_i (u_x - u_{x'})_i|^p \} \cdot \sum_{i \in I} |\lambda_i (u_x - u_{x'})_i|^{2-p} \left( |\lambda_i (u_x - v)_i|^{2p-2} + |\lambda_i (u_{x'} - v)_i|^{2p-2} \right).$$

We can then proceed as

$$\sum_{i \in I} |\lambda_i (u_x - u_{x'})_i|^{2-p} \left( |\lambda_i (u_x - v)_i|^{2p-2} + |\lambda_i (u_{x'} - v)_i|^{2p-2} \right)$$

$$\lesssim \sum_{i \in I} \lambda_i^p |(u_x - u_{x'})_i|^{2-p} \max\{ |(u_x - v)_i|^{2p-2}, |(u_{x'} - v)_i|^{2p-2} \}$$

$$\lesssim \left( \sum_{i \in I} \lambda_i^p |(u_x - u_{x'})_i|^p \right)^{\frac{2-p}{p}} \left( \sum_{i \in I} \lambda_i^p \max\{ |(u_x - v)_i|^p, |(u_{x'} - v)_i|^p \} \right)^{\frac{2p-2}{p}}$$

$$\leq \|\Lambda(u_x - u_{x'})\|_{I,p}^{2-p} \left( \|\Lambda(u_x - v)\|_{I,p}^p + \|\Lambda(u_{x'} - v)\|_{I,p}^p \right)^{\frac{2p-2}{p}}.$$

For the $\ell_p$-norms in the preceding line, we remind the readers that they have been restricted to the indices in $I$ and do not refer to the $\ell_p$-norm of the entire vector.

Since $u_x - v = (u_{\bar{x}} - v) + (u_x - u_{\bar{x}})$, by our assumptions,

$$\|\Lambda(u_x - v)\|_{I,p}^p \leq 2^{p-1} (\|\Lambda(u_{\bar{x}} - v)\|_{I,p}^p + \|\Lambda(u_x - u_{\bar{x}})\|_{I,p}^p)$$
$$\lesssim \alpha \cdot \|\Lambda(u_{\bar{x}} - v)\|_p^p + \alpha \cdot F$$
$$\lesssim \alpha \cdot F. \tag{39}$$

Similarly,

$$\|\Lambda(u_x - u_{x'})\|_{I,p} \leq \|\Lambda(u_x - v)\|_{I,p} + \|\Lambda(u_{x'} - v)\|_{I,p} \lesssim (\alpha F)^{\frac{1}{p}}.$$

It follows that

$$\rho(x, x')^2 \lesssim \|\Lambda(u_x - u_{x'})\|_{I,\infty}^p (\alpha F)^{\frac{2-p}{p}} (\alpha F)^{\frac{2p-2}{p}} \leq \alpha F \cdot \|\Lambda(u_x - u_{x'})\|_{I,\infty}^p. \tag{40}$$

When $p > 2$, we use the fact that $|z_i|^{2p-2} \leq |z_i|^p \|z\|_\infty^{p-2}$ for a vector $z$ and so we can proceed from (38) as

$$\sum_{i \in I} |\lambda_i (u_x - u_{x'})_i|^2 \left( |\lambda_i (u_x - v)_i|^{2p-2} + |\lambda_i (u_{x'} - v)_i|^{2p-2} \right)$$

$$\leq \sum_{i \in I} \|\Lambda(u_x - u_{x'})\|_{I,\infty}^2 (|\lambda_i (u_x - v)_i|^p \|\Lambda(u_x - v)\|_{I,\infty}^{p-2} + |\lambda_i (u_{x'} - v)_i|^p \|\Lambda(u_{x'} - v)\|_{I,\infty}^{p-2})$$

$$= \|\Lambda(u_x - u_{x'})\|_{I,\infty}^2 \left( \|\Lambda(u_x - v)\|_{I,p}^p \|\Lambda(u_x - v)\|_{I,\infty}^{p-2} + \|\Lambda(u_{x'} - v)\|_{I,p}^p \|\Lambda(u_{x'} - v)\|_{I,\infty}^{p-2} \right)$$

$$\lesssim \alpha F \cdot \|\Lambda(u_x - u_{x'})\|_{I,\infty}^2 \left( \|\Lambda(u_x - v)\|_{I,\infty}^{p-2} + \|\Lambda(u_{x'} - v)\|_{I,\infty}^{p-2} \right) \quad \text{by (39)}$$

$$\leq \alpha F \cdot \|\Lambda(u_x - u_{x'})\|_{I,\infty}^2 \left( \|\Lambda(u_x - v)\|_{I,p}^{p-2} + \|\Lambda(u_{x'} - v)\|_{I,p}^{p-2} \right).$$

Since

$$\|\Lambda(u_x - v)\|_{I,p} \leq \|\Lambda(u_x - u_{\bar{x}})\|_{I,p} + \|\Lambda(u_{\bar{x}} - v)\|_{I,p} \lesssim (\alpha F)^{\frac{1}{p}}$$

we have

$$\rho(x, x')^2 \lesssim (\alpha F)^{2 - \frac{2}{p}} \cdot \|\Lambda(u_x - u_{x'})\|_{I,\infty}^2. \tag{41}$$

Combining (40) and (41) yields

$$\rho(x, x') \lesssim (\alpha F)^{(1 - \frac{1}{p}) \vee \frac{1}{2}} \|\Lambda(u_x - u_{x'})\|_{I,\infty}^{\frac{p}{2} \wedge 1} \tag{42}$$

and our next task to upper bound $\|\Lambda(u_x - u_{x'})\|_{I,\infty}$.

Recall that $w_i \le 2d/n$, we have by the Lipschitz condition and Lemma 5(d),

$$|\lambda_i(u_x - u_{x'})_i| \le |(\Lambda(Ax - Ax'))_i| \le w_i^{\frac{1}{p}} d^{\frac{1}{2} - \frac{1}{2 \vee p}} \|\Lambda A(x - x')\|_p$$
$$\lesssim \left(\frac{d}{n}\right)^{\frac{1}{p}} d^{\frac{1}{2} - \frac{1}{2 \vee p}} \|\Lambda A(x - x')\|_p. \tag{43}$$

Plugging this result into (42) immediately leads to

$$\rho(x, x') \lesssim K \cdot d^{\frac{1}{2} - \frac{1}{2 \vee p}} \|\Lambda A(x - x')\|_p^{\frac{p}{2} \wedge 1}. \tag{44}$$

Alternatively,

$$|\lambda_i(u_x - u_{x'})_i| \lesssim \left(\frac{d}{n}\right)^{\frac{1}{p}} \cdot \frac{|\Lambda_{ii}(Ax - Ax')_i|}{w_i^{\frac{1}{p}}}$$
$$\lesssim \left(\frac{d}{n}\right)^{\frac{1}{p}} \left(w_i \left(\frac{|\Lambda_{ii}(Ax - Ax')_i|}{w_i^{\frac{1}{p}}}\right)^q\right)^{\frac{1}{q}} \quad \text{since } w_i > \frac{\varepsilon^p d}{n^2}$$
$$= \left(\frac{d}{n}\right)^{\frac{1}{p}} \left\|W^{-\frac{1}{p}} \Lambda A(x - x')\right\|_{w,q} \quad \text{recall that } \|x\|_{w,p} = (\sum_{i=1}^n w_i |x_i|^q)^{1/q}$$

and thus

$$\rho(x, x') \lesssim K \cdot \left\|W^{-\frac{1}{p}} \Lambda A(x - x')\right\|_{w,q}^{\frac{p}{2} \wedge 1}. \tag{45}$$

as claimed.

Finally, we bound $\mathrm{Diam}(T, \rho)$. By the definition of $T$ and (44), we have that $\|\Lambda A(x - x')\|_p^p \lesssim R$. The claimed upper bound on $\mathrm{Diam}(T, \rho)$ follows immediately. $\qquad \square$

### B.3.4 Bounding Dudley's Integral

In this section, we will prove (37), i.e.

$$\int_0^{\mathrm{Diam}(T, \rho)} \sqrt{\log \mathcal{N}(T, \rho, r)} dr \lesssim \alpha \cdot \Gamma \cdot \mathsf{poly} \log n,$$

where $\Gamma$ is as defined in (21). To further simplify the notation, let

$$\phi = \frac{p}{2} \wedge 1, \quad \beta = \frac{1}{p} \wedge \frac{1}{2}, \quad \gamma = \frac{1}{2} - \frac{1}{p \vee 2}, \quad \theta = (1 - \frac{1}{p}) \vee \frac{1}{2}.$$

Note that

$$\log \mathcal{N}(T, \rho, r) \le \log \mathcal{N}(R^{\frac{1}{p}} \cdot B_p(\Lambda A), K d^\gamma \|W^{-\frac{1}{p}} \Lambda A(\cdot)\|_{w,p}^\phi, r) \quad \text{by Lemma 20}$$
$$= \log \mathcal{N}(B_p(\Lambda A), \|W^{-\frac{1}{p}} \Lambda A(\cdot)\|_{w,p}, \frac{1}{R^{\frac{1}{p}}} (\frac{r}{K d^\gamma})^{\frac{1}{\phi}})$$
$$= \log \mathcal{N}(B_{w,p}(E), \|\cdot\|_{w,p}, \frac{1}{R^{\frac{1}{p}} d^\gamma} (\frac{r}{K})^{\frac{1}{\phi}}) \quad \text{since } \gamma/\phi = \gamma$$

where $E = \mathrm{colspace}(W^{-\frac{1}{p}} \Lambda A)$ and is endowed with norms $\|\cdot\|_{w,p}$ for $p \ge 1$.

Now, we shall split the integral domain into two parts $[0, \lambda]$ and $[\lambda, \mathrm{Diam}(T, \rho)]$ for some $\lambda \le \frac{1}{2} K R^\beta d^\gamma$ to be determined later (note that $\phi = p\beta$). Note that when $r \le \lambda$, we have $(r/K)^{1/\phi}/(R^{\frac{1}{p}} d^\gamma) \le (\frac{1}{2})^\phi < 1$.

By Lemma 8 Case 1, we have (letting $\lambda' = \lambda/(KR^\beta d^\gamma)$)

$$
\begin{aligned}
\int_0^\lambda \sqrt{\log\mathcal{N}(T,\rho,r)}dr &\lesssim \int_0^\lambda \sqrt{\log\mathcal{N}(B_{w,p}(E), \|\cdot\|_{w,p}, \frac{1}{R^{\frac{1}{p}}d^\gamma}(\frac{r}{K})^{\frac{1}{\phi}})}dr \\
&= \int_0^{\lambda'} \sqrt{\log\mathcal{N}(B_{w,p}(E), \|\cdot\|_{w,p}, s^{\frac{1}{\phi}})}Kd^\gamma R^\beta ds \\
&\lesssim \int_0^{\lambda'} \sqrt{d\log\frac{1}{s}}Kd^\gamma R^\beta ds \\
&\lesssim \sqrt{d}\cdot Kd^\gamma R^\beta \int_0^{\lambda'} \log\frac{1}{s}ds \\
&\lesssim \sqrt{d}\cdot Kd^\gamma R^\beta \cdot \lambda'\log\frac{1}{\lambda'} \\
&\lesssim \lambda\sqrt{d}\log\frac{d^\gamma R^\beta K}{\lambda}.
\end{aligned}
\tag{46}
$$

To handle the integral over $[\lambda, \mathrm{Diam}(T,\rho)]$, we bound

$$
\begin{aligned}
\log\mathcal{N}(T,\rho,r) &\leq \log\mathcal{N}(R^{\frac{1}{p}}\cdot B_p(\Lambda A), K\|W^{-\frac{1}{p}}\Lambda A(\cdot)\|_{w,q}^\phi, r) && \text{by Lemma 20} \\
&= \log\mathcal{N}(B_p(\Lambda A), \|W^{-\frac{1}{p}}\Lambda A(\cdot)\|_{w,q}, \frac{1}{R^{\frac{1}{p}}}(\frac{r}{K})^{\frac{1}{\phi}}) \\
&= \log\mathcal{N}(B_{w,p}(E), \|\cdot\|_{w,q}, \frac{1}{R^{\frac{1}{p}}}(\frac{r}{K})^{\frac{1}{\phi}})
\end{aligned}
$$

where $E = \mathrm{colspace}(W^{-\frac{1}{p}}\Lambda A)$ and is endowed with norms $\|\cdot\|_{w,q}$ for $q \geq 1$. We further divide the estimates into two cases. For $p \in [1,2]$, we invoke Lemma 8 Case 2 and obtain that

$$
\begin{aligned}
\int_\lambda^{\mathrm{Diam}(T,\rho)} \sqrt{\log\mathcal{N}(T,\rho,r)}dr &\lesssim \int_\lambda^{\mathrm{Diam}(T,\rho)} \sqrt{\log\mathcal{N}(B_{w,p}(E), \|\cdot\|_{w,q}, \frac{1}{R^{\frac{1}{p}}}(\frac{r}{K})^{\frac{2}{p}})}dr \\
&\lesssim R^{\frac{1}{2}}K\sqrt{\log(\frac{n}{\varepsilon d})\sqrt{\log d}}\int_\lambda^{\mathrm{Diam}(T,\rho)} \frac{1}{r}dr \\
&\lesssim R^{\frac{1}{2}}K\sqrt{\log\frac{n}{\varepsilon d}\sqrt{\log d}}\log\frac{\mathrm{Diam}(T,\rho)}{\lambda}.
\end{aligned}
\tag{47}
$$

For $p > 2$, we invoke Lemma 8 Case 3 and obtain that

$$
\begin{aligned}
\int_\lambda^{\mathrm{Diam}(T,\rho)} \sqrt{\log\mathcal{N}(T,\rho,r)}dr &\lesssim \int_\lambda^{\mathrm{Diam}(T,\rho)} \sqrt{\log\mathcal{N}(B_{w,p}(E), \|\cdot\|_{w,q}, \frac{r}{R^{\frac{1}{p}}\cdot K})}dr \\
&\lesssim d^{\frac{1}{2}-\frac{1}{p}}\sqrt{\log\frac{n}{\varepsilon d}}R^{\frac{1}{p}}K\int_\lambda^{\mathrm{Diam}(T,\rho)} \frac{1}{r}dr \\
&\lesssim d^{\frac{1}{2}-\frac{1}{p}}\sqrt{\log\frac{n}{\varepsilon d}}R^{\frac{1}{p}}K\log\frac{\mathrm{Diam}(T,\rho)}{\lambda}.
\end{aligned}
\tag{48}
$$

Combining (47) and (48) yields

$$
\int_\lambda^{\mathrm{Diam}(T,\rho)} \sqrt{\log\mathcal{N}(T,\rho,r)}dr \lesssim d^\gamma R^\beta K\sqrt{\log\frac{n}{\varepsilon d}\sqrt{\log d}}\log\frac{\mathrm{Diam}(T,\rho)}{\lambda}.
\tag{49}
$$

Recall that by Lemma 20,

$$
\mathrm{Diam}(T,\rho) \lesssim d^\gamma KR^\beta, \quad \text{where} \quad K = (\alpha F)^\theta \left(\frac{d}{n}\right)^\beta.
$$

Combining (46) and (49) and taking $\lambda = (d^{\gamma - \frac{1}{2}} R^\beta K) \wedge (\frac{1}{2} d^\gamma R^\beta K)$, we have

$$\int_0^{\text{Diam}(T,\rho)} \sqrt{\log \mathcal{N}(T, \rho, r)} dr \lesssim \lambda \sqrt{d} \log \frac{d^\gamma R^\beta K}{\lambda} + d^\gamma R^\beta K \sqrt{\log \frac{n}{\varepsilon d}} \sqrt{\log d} \log \frac{d^\gamma K R^\beta}{\lambda}$$

$$\lesssim d^\gamma R^\beta K \log^{\frac{5}{4}} d \sqrt{\log \frac{n}{\varepsilon d}}$$

$$= \alpha \cdot \Gamma \cdot \log^{\frac{5}{4}} d \sqrt{\log \frac{n}{\varepsilon d}}.$$

## B.4 REMOVING THE DEPENDENCE ON $n$

In this section, we reduce the $\log n$ factors in Theorem 12 to $\log(d/\varepsilon)$ factors, thereby proving Theorem 1. This is achieved by first applying a sampling matrix $S^\circ$ to reduce the dimension of the regression problem from $n$ to $\text{poly}(d/\varepsilon)$ before invoking Algorithm 3; see Algorithm 2 for the full algorithm. The sampling matrix $S^\circ$ uses a larger sampling rate $\alpha^\circ$, which allows for controlling the error in Lemma 14 via Bernstein's inequality with a simple net argument instead of the chaining argument or Dudley's integral and thus avoiding the $\log n$ factor from entropy estimates.

Recall that, in Appendix B.1, we introduce the matrix $\Lambda$ to ensure that the Lewis weights are bounded uniformly. We will include the matrix $\Lambda$ in our proof and abuse the notations by dropping the prime mark as indicated in Appendix B.1.

The following is a weaker version of Lemma 14 for reducing $n$ to $\text{poly}(d/\varepsilon)$.

**Lemma 21.** *Let $A \in \mathbb{R}^{n \times d}$, $v \in \mathbb{R}^n$, $f \in \text{Lip}_1$, $\varepsilon > 0$, $\Lambda \in \mathbb{R}^{n \times n}$, $\alpha \in [0, 1]$, $S \in \mathbb{R}^{n \times n}$ and $R \in \mathbb{R}$ be as defined in Lemma 14 and satisfy the same constraints. Suppose that $0 \leq \alpha \leq 1$ such that $\alpha n \gtrsim (d^{(\frac{p}{2} \vee 1)+1}/\varepsilon^{p+2}) \log(1/\varepsilon)$ and $R \geq \|\Lambda A x^*\|_p^p \vee \|\Lambda(f(Ax^*) - v)\|_p^p$. Let $T = \left\{ x \in \mathbb{R}^d \mid \|\Lambda A x\|_p^p \leq R \right\}$. When conditioned on*

$$\|S\Lambda(f(Ax^*) - v)\|_p^p \lesssim R,$$

*it holds with probability at least $0.99$ that*

$$\sup_{x \in T} \left| (\|S\Lambda(f(Ax) - v)\|_p^p - \|S\Lambda(f(Ax^*) - v)\|_p^p) - (\|\Lambda(f(Ax) - v)\|_p^p - \|\Lambda(f(Ax^*) - v)\|_p^p) \right|$$

$$\leq C\varepsilon R, \tag{50}$$

*where $C$ is an absolute constant.*

*Proof.* Recall that the error bound in Lemma 14 consists of three terms. By the same proofs as in Lemmas 18 and 19, the first two terms remain the same, both of which are now bounded by $C\varepsilon R^p$ under our assumptions. The rest of the proof is devoted to deriving a similar bound for the third term. Recall that we need to upper bound

$$\sup_{x \in T} \left| \sum_{i \in J} S_{ii}^p \xi_i (z_x)_i \right|,$$

where $z_x$ is as defined in (35) and $\{\xi_i\}$ are independent Rademacher variables. We shall use a net argument here.

Fix an $x \in T$. Let $W_i = S_{ii}^p \xi_i (z_x)_i$, then $\mathbb{E} W_i = 0$ and

$$|W_i| \leq \frac{1}{\alpha} |(z_x)_i|$$

$$= \frac{1}{\alpha} |(\Lambda(f(Ax) - v)_i^p - (\Lambda(f(Ax^*) - v)_i^p|$$

$$\leq \frac{1}{\alpha} \cdot 2^{p-1} (|\Lambda_{ii}(f(Ax) - f(Ax^*))_i|^p + |\Lambda_{ii}(f(Ax^*) - v)_i|^p) + |\Lambda_{ii}(f(Ax^*) - v)_i|^p$$

$$\lesssim \frac{1}{\alpha} (|\Lambda_{ii}(Ax - Ax^*)_i|^p + |\Lambda_{ii}(f(Ax^*) - v)_i|^p).$$

Now we use (43) for the first term in the brackets and the definition of $G$ in (32) for the second term. We proceed as

$$|W_i| \lesssim \frac{1}{\alpha} \left( \frac{d}{n} \cdot d^{(\frac{p}{2}-1)\vee 0} \|\Lambda A(x - x^*)\|_p^p + \frac{1}{\varepsilon^p} d^{(\frac{p}{2}-1)\vee 0} \cdot \frac{d}{n} \cdot R \right) \lesssim \underbrace{\frac{1}{\alpha n \varepsilon^p} \cdot d^{\frac{p}{2}\vee 1} R}_{:=\Delta}.$$

Next we bound $\mathbb{E}(\sum_{i \in J} W_i^2)$.

$$\mathbb{E} \sum_{i \in J} W_i^2 = \sum_{i \in J} \mathbb{E} S_{ii}^{2p}(z_x)_i^2 \leq \sum_i \frac{1}{\alpha} \cdot \max_i |(z_x)_i| \cdot |(z_x)_i| \lesssim \Delta \cdot \sum_{i \in J} |(z_x)_i|.$$

Note that

$$\sum_{i \in J} |(z_x)_i| \leq \|\Lambda(f(Ax) - v)\|_p^p + \|\Lambda(f(Ax^*) - v)\|_p^p$$

$$\leq 2^{p-1}(\|\Lambda(f(Ax) - f(Ax^*))\|_p^p + \|\Lambda(f(Ax^*) - v)\|_p^p) + \|\Lambda(f(Ax^*) - v)\|_p^p$$

$$\leq 2^{p-1} \|\Lambda A(x - x^*)\|_p^p + (2^{p-1} + 1) \|\Lambda(f(Ax^*) - v)\|_p^p$$

$$\lesssim \|\Lambda Ax\|_p^p + \|\Lambda Ax^*\|_p^p + \|\Lambda(f(Ax^*) - v)\|_p^p$$

$$\lesssim R.$$

Hence

$$\mathbb{E} \sum_{i \in J} W_i^2 \lesssim \Delta \cdot R.$$

It follows from Bernstein's inequality that

$$\Pr\left\{ \left| \sum_{i \in J} W_i \right| \geq \eta R \right\} \leq 2 \exp\left( -c \frac{\eta^2 R^2}{\Delta \cdot R + \Delta \cdot \eta R} \right)$$

$$\leq 2 \exp\left( -c' \frac{\eta^2 R}{\Delta} \right)$$

$$\leq \exp\left( -Cd \log \frac{1}{\eta} \right),$$

provided that

$$\eta^2 R \gtrsim d \cdot \Delta = \begin{cases} d^{\frac{p}{2}+1} \log \frac{1}{\eta} \cdot R/(\alpha n \varepsilon^p) & p > 2 \\ d^2 R \log \frac{1}{\eta} /(\alpha n \varepsilon^p) & 1 \leq p < 2 \end{cases}$$

or

$$\alpha n \gtrsim \frac{d^{1+(\frac{p}{2}\vee 1)}}{\eta^2 \varepsilon^p} \log \frac{1}{\eta}. \tag{51}$$

To summarize, we have shown that when (51), for each fixed $x \in T$ that with probability at least $1 - \delta$ (where $\delta = \exp(-Cd \log(1/\eta))$), it holds

$$\left| \sum_{i \in J} S_{ii}^p \xi_i (z_x)_i \right| \leq \eta R.$$

To obtain an upper bound for the supremum over $x \in T$, we employ a standard net argument. Let $N \subseteq T$ be an $(\eta R^{\frac{1}{p}})$-net of $T$ such that $|N| \leq (3/\eta)^d$. By a union bound, we have with probability at least $1 - |N|\delta \geq 0.99$ that

$$\sup_{x \in N} \left| \sum_{i \in J} S_{ii}^p \xi_i (z_x)_i \right| \leq \eta R.$$

For an $\Lambda Ax \in T$, there exists $\Lambda Ay \in N$ such that $\|\Lambda A(x - y)\|_p \leq \eta R^{\frac{1}{p}}$. Thus

$$\sup_{x \in T} \left| \sum_{i \in J} S_{ii}^p \xi_i (z_x)_i \right| \leq \sup_{y \in N} \left| \sum_{i \in J} S_{ii}^p \xi_i (z_y)_i \right| + \sup_{x \in T} \left| \sum_{i \in J} S_{ii}^p \xi_i (z_x - z_y)_i \right| \leq \eta R + \sup_{x \in T} \left| \sum_{i \in J} S_{ii}^p \xi_i (z_x - z_y)_i \right|.$$

We bound the error term by Hölder's inequality as

$$
\left| \sum_{i \in J} S_{ii}^p \xi_i (z_x - z_y)_i \right|
$$

$$
= \sum_{i \in J} S_{ii}^p |(z_x - z_y)_i|
$$

$$
= \sum_{i \in J} S_{ii}^p \left| |\Lambda_{ii}(u_x - v)_i|^p - |\Lambda_{ii}(u_y - v)_i|^p \right|
$$

$$
\lesssim \sum_{i \in J} S_{ii}^p |\Lambda_{ii}(u_x - u_y)_i| \left( |\Lambda_{ii}(u_x - v)_i|^{p-1} + |\Lambda_{ii}(u_y - v)_i|^{p-1} \right)
$$

$$
\lesssim \left( \sum_{i \in J} S_{ii}^p |\Lambda_{ii}(Ax - Ay)_i|^p \right)^{\frac{1}{p}} \left( \left( \sum_{i \in J} S_{ii}^p |\Lambda_{ii}(u_x - v)_i|^p \right)^{1-\frac{1}{p}} + \left( \sum_{i \in J} S_{ii}^p |\Lambda_{ii}(u_y - v)_i|^p \right)^{1-\frac{1}{p}} \right)
$$

$$
\leq \|S\Lambda A(x - y)\|_p (\|S\Lambda(f(Ax) - v)\|_p^{p-1} + \|S\Lambda(f(Ay) - v)\|_p^{p-1})
$$

Using the the subspace embedding property of $S$,

$$
\|S\Lambda A(x - y)\|_p \leq 2\|\Lambda A(x - y)\|_p \leq 2\eta R^{\frac{1}{p}}
$$

and

$$
\begin{aligned}
\|S\Lambda(f(Ax) - v)\|_p^{p-1} &\lesssim \|S\Lambda(f(Ax) - f(Ax^*))\|_p^{p-1} + \|S\Lambda(f(Ax^*) - v)\|_p^{p-1} \\
&\leq \|S\Lambda A(x - x^*)\|_p^{p-1} + \|S\Lambda(f(Ax^*) - v)\|_p^{p-1} \\
&\lesssim \|\Lambda A(x - x^*)\|_p^{p-1} + \|S\Lambda(f(Ax^*) - v)\|_p^{p-1} \\
&\lesssim R^{1-\frac{1}{p}}.
\end{aligned}
$$

Hence,

$$
\sup_{x \in T} \left| \sum_{i \in J} S_{ii}^p \xi_i (z_x - z_y)_i \right| \lesssim \eta R.
$$

Therefore,

$$
\left| \sum_{i \in J} S_{ii}^p \xi_i (z_x)_i \right| \lesssim \eta R
$$

and the claimed result follows from setting $\eta \sim \varepsilon$. $\qquad \square$

To prove that the output $\hat{x}$ of Algorithm 2 satisfies (20), let $x^\circ \in \mathbb{R}^d$ be

$$
x^\circ := \arg\min_{x \in \mathbb{R}^d} \|S^\circ \Lambda(f(Ax) - b)\|_p^p + \varepsilon^2 \|\Lambda Ax\|_p^p
$$

where $\Lambda$ is the matrix ensuring the Lewis weights of $\Lambda A$ are uniformly bounded. Note that we set the regularized parameter to be $\varepsilon^2$ instead of $\varepsilon$. We highlight that this $x^\circ$ is for the purpose of analysis and we do not actually compute it in the algorithm. From now on, we set

$$
\alpha \sim \frac{d^{\frac{p}{2} \vee 1}}{m \varepsilon^{p \vee 2}} \cdot \text{poly} \log m \quad \text{and} \quad \alpha^\circ \sim \frac{d^{(\frac{p}{2} \vee 1)+1}}{n(\varepsilon^4)^{p+2}} \log(1/\varepsilon)
$$

where $m$ is the number of nonzero rows in $S^\circ A$ which is the same as the $m$ defined in Algorithm 2. Note that our choice of $\alpha^\circ$ implies that $m \sim \frac{d^{(\frac{p}{2} \vee 1)+1}}{\varepsilon^{4p+8}} \log(1/\varepsilon) = \text{poly}(d, \frac{1}{\varepsilon})$. We preview that when we use Lemma 21 we aim for the error of $C\varepsilon^2 R$ in (50). Combining with the regularized parameter $\varepsilon^2$, we set $\varepsilon^4$ in $\alpha^\circ$. Recall that our goal is to simply reduce $n$ to $\text{poly}(\frac{d}{\varepsilon})$ and thus these choices of the exponents of $\varepsilon$ may not be optimized. Nonetheless, they are sufficient to achieve our objective.

We begin with using Lemma 13 with $\bar{x} = x^\circ$ and $S^\circ \Lambda$ as the matrix ensuring the Lewis weights of $S^\circ \Lambda A$ are uniformly bounded. We now verify the conditions in Lemma 13. By Appendix D, the

Lewis weights of $S^\circ \Lambda A$ are uniformly bounded by $O(d/m)$ with probability at least 0.99. Clearly, the output of Algorithm 2 satisfies

$$\hat{x} = \arg\min_{x \in \mathbb{R}^d} \|SS^\circ \Lambda(f(Ax) - b)\|_p^p + \varepsilon\|S^\circ \Lambda Ax\|_p^p$$

and, by the optimality of $x^\circ$, we have

$$\|S^\circ \Lambda(f(Ax^\circ) - b)\|_p^p - \|S^\circ \Lambda(f(A\hat{x}) - b)\|_p^p \leq \varepsilon^2 \|\Lambda A\hat{x}\|_p^p \tag{52}$$

We need to upper bound $\|\Lambda A\hat{x}\|_p^p$. By the optimality of $\hat{x}$, we have

$$\|SS^\circ \Lambda(f(A\hat{x}) - b)\|_p^p + \varepsilon\|S^\circ \Lambda A\hat{x}\|_p^p \leq \|SS^\circ \Lambda(f(Ax^\circ) - b)\|_p^p + \varepsilon\|S^\circ \Lambda Ax^\circ\|_p^p$$

which implies

$$\|S^\circ \Lambda A\hat{x}\|_p^p \leq \frac{1}{\varepsilon}\|SS^\circ \Lambda(f(Ax^\circ) - b)\|_p^p + \|S^\circ \Lambda Ax^\circ\|_p^p. \tag{53}$$

Since $S^\circ$ is an $\ell_p$-subspace embedding matrix for $\Lambda A$ with constant distortion with probability 0.99 because of our choice of $\alpha^\circ$ and we condition on it from now on, we have

$$\frac{1}{2}\|\Lambda A\hat{x}\|_p^p \leq \|S^\circ \Lambda A\hat{x}\|_p^p. \tag{54}$$

Also, by Markov inequality, with probability at least 0.99, we have

$$\|SS^\circ \Lambda(f(Ax^\circ) - b)\|_p^p \leq 100\|S^\circ \Lambda(f(Ax^\circ) - b)\|_p^p. \tag{55}$$

Plugging (54) and (55) into (53), we have

$$\|\Lambda A\hat{x}\|_p^p \lesssim \frac{1}{\varepsilon}\|S^\circ \Lambda(f(Ax^\circ) - b)\|_p^p + \|S^\circ \Lambda Ax^\circ\|_p^p.$$

and when we further plug this into (52) we have

$$\|S^\circ \Lambda(f(Ax^\circ) - b)\|_p^p - \|S^\circ \Lambda(f(A\hat{x}) - b)\|_p^p$$
$$\lesssim \varepsilon^2 \cdot (\frac{1}{\varepsilon}\|S^\circ \Lambda(f(Ax^\circ) - b)\|_p^p + \|S^\circ \Lambda Ax^\circ\|_p^p)$$
$$= \varepsilon \cdot (\|S^\circ \Lambda(f(Ax^\circ) - b)\|_p^p + \varepsilon\|S^\circ \Lambda Ax^\circ\|_p^p)$$

which completes the condition verification for Lemma 13.

By Lemma 13, with probability 0.99, we have

$$|(\|SS^\circ \Lambda(f(A\hat{x}) - b)\|_p^p - \|SS^\circ \Lambda(f(Ax^\circ) - b)\|_p^p) - (\|S^\circ \Lambda(f(A\hat{x}) - b)\|_p^p - \|S^\circ \Lambda(f(Ax^\circ) - b)\|_p^p)|$$
$$\leq \varepsilon \cdot (\|S^\circ \Lambda(f(Ax^\circ) - b)\|_p^p + \|S^\circ \Lambda Ax^\circ\|^p).$$

By rearranging the terms, we have

$$\|S^\circ \Lambda(f(A\hat{x}) - b)\|_p^p - \|S^\circ \Lambda(f(Ax^\circ) - b)\|_p^p$$
$$\leq \|SS^\circ \Lambda(f(A\hat{x}) - b)\|_p^p - \|SS^\circ \Lambda(f(Ax^\circ) - b)\|_p^p + \varepsilon \cdot (\|S^\circ \Lambda(f(Ax^\circ) - b)\|_p^p + \|S^\circ \Lambda Ax^\circ\|^p)$$
$$\leq \varepsilon \cdot \|S^\circ \Lambda Ax^\circ\|_p^p + \varepsilon \cdot (\|S^\circ \Lambda(f(Ax^\circ) - b)\|_p^p + \|S^\circ \Lambda Ax^\circ\|^p) \quad \text{by the optimality of } \hat{x}$$
$$\lesssim \varepsilon \cdot (\|S^\circ \Lambda(f(Ax^\circ) - b)\|_p^p + \|S^\circ \Lambda Ax^\circ\|^p) \tag{56}$$

Now, we would like to use Lemma 21 with $R \sim \frac{1}{\varepsilon}\mathsf{OPT} + \|\Lambda Ax^*\|_p^p$ for $x = \hat{x}$ and hence we need to verify $\hat{x} \in T$. By the optimality of $\hat{x}$, we have

$$\|SS^\circ \Lambda(f(A\hat{x}) - b)\|_p^p + \varepsilon\|S^\circ \Lambda A\hat{x}\|_p^p \leq \|SS^\circ \Lambda(f(Ax^*) - b)\|_p^p + \varepsilon\|S^\circ \Lambda Ax^*\|_p^p$$

which implies

$$\|S^\circ \Lambda A\hat{x}\|_p^p \leq \frac{1}{\varepsilon}\|SS^\circ \Lambda(f(Ax^*) - b)\|_p^p + \|S^\circ \Lambda Ax^*\|_p^p. \tag{57}$$

Recall that we condition that $S^\circ$ is an $\ell_p$-subspace embedding matrix for $\Lambda A$ with constant distortion. We have

$$\frac{1}{2}\|\Lambda A\hat{x}\|_p^p \leq \|S^\circ \Lambda A\hat{x}\|_p^p \quad \text{and} \quad \|S^\circ \Lambda Ax^*\|_p^p \leq 2\|\Lambda Ax^*\|_p^p. \tag{58}$$

Also, by Markov inequality, with probability 0.99, we have
$$\|SS^\circ \Lambda(f(Ax^*) - b)\|_p^p \le 100\|\Lambda(f(Ax^*) - b)\|_p^p = 100\mathsf{OPT}. \tag{59}$$
Plugging (58) and (59) into (58), we have
$$\|\Lambda A\hat{x}\|_p^p \lesssim \frac{1}{\varepsilon}\mathsf{OPT} + \|\Lambda Ax^*\|_p^p \quad \text{which implies } \hat{x} \in T.$$
By Lemma 21 with our choice of $\alpha^\circ$, with probability 0.99, we have
$$\left|(\|S^\circ \Lambda(f(A\hat{x}) - b)\|_p^p - \|S^\circ \Lambda(f(Ax^*) - b)\|_p^p) - (\|\Lambda(f(A\hat{x}) - b)\|_p^p - \|\Lambda(f(Ax^*) - b)\|_p^p)\right|$$
$$\lesssim \varepsilon^4 \cdot (\frac{1}{\varepsilon}\mathsf{OPT} + \|\Lambda Ax^*\|_p^p)$$
$$\lesssim \varepsilon \cdot (\mathsf{OPT} + \|\Lambda Ax^*\|_p^p). \tag{60}$$
By the optimality of $x^\circ$, we have
$$-\|S^\circ \Lambda(f(Ax^*) - b)\|_p^p \le -\|S^\circ \Lambda(f(Ax^\circ) - b)\|_p^p + \varepsilon^2\|\Lambda Ax^*\|_p^p$$
which implies
$$\|S^\circ \Lambda(f(A\hat{x}) - b)\|_p^p - \|S^\circ \Lambda(f(Ax^*) - b)\|_p^p$$
$$\le \|S^\circ \Lambda(f(A\hat{x}) - b)\|_p^p - \|S^\circ \Lambda(f(Ax^\circ) - b)\|_p^p + \varepsilon^2\|\Lambda Ax^*\|_p^p$$
$$\lesssim \varepsilon \cdot (\|S^\circ \Lambda(f(Ax^\circ) - b)\|_p^p + \|S^\circ \Lambda Ax^\circ\|^p) + \varepsilon^2\|\Lambda Ax^*\|_p^p \quad \text{by (56)}.$$

By rearranging the terms in (60), we have
$$\|\Lambda(f(A\hat{x}) - b)\|_p^p - \|\Lambda(f(Ax^*) - b)\|_p^p$$
$$\lesssim \varepsilon \cdot (\|S^\circ \Lambda(f(Ax^\circ) - b)\|_p^p + \|S^\circ \Lambda Ax^\circ\|^p + \mathsf{OPT} + \|\Lambda Ax^*\|^p) \tag{61}$$
It means that we need to upper bound the terms $\|S^\circ \Lambda(f(Ax^\circ) - b)\|_p^p$ and $\|S^\circ \Lambda Ax^\circ\|^p$. For $\|S^\circ \Lambda(f(Ax^\circ) - b)\|_p^p$, we have
$$\|S^\circ \Lambda(f(Ax^\circ) - b)\|_p^p \le \|S^\circ \Lambda(f(Ax^*) - b)\|_p^p + \varepsilon^2\|\Lambda Ax^*\|_p^p \quad \text{by the optimality of } x^\circ$$
$$\le 100\mathsf{OPT} + \varepsilon^2\|\Lambda Ax^*\|_p^p, \tag{62}$$
where the last inequality holds with probability 0.99 by Markov inequality. For $\|S^\circ \Lambda Ax^\circ\|^p$, we have
$$\|S^\circ \Lambda Ax^\circ\|^p \lesssim \|\Lambda Ax^\circ\|^p \quad \text{recall that } S^\circ \text{ is an } \ell_p \text{ subspace embedding}.$$
To further bound the term $\|\Lambda Ax^\circ\|^p$, we have
$$\|\Lambda Ax^\circ\|_p^p \le \frac{1}{\varepsilon^2}\|S^\circ \Lambda(f(Ax^*) - b)\|_p^p + \|\Lambda Ax^*\|_p^p \quad \text{by the optimality of } x^\circ$$
$$\le \frac{100}{\varepsilon^2}\mathsf{OPT} + \|\Lambda Ax^*\|_p^p \quad \text{by Markov inequality with probability 0.99}.$$
Note that this bound is not enough to finish our proof. However, it implies that $x^\circ \in T$ where $T$ is the set defined in Lemma 21 with $R = \frac{100}{\varepsilon^2}\mathsf{OPT} + \|\Lambda Ax^*\|_p^p$. By Lemma 21 with our choice of $\alpha^\circ$, with probability 0.99, we have
$$\left|(\|S^\circ \Lambda(f(Ax^\circ) - v)\|_p^p - \|S^\circ \Lambda(f(Ax^*) - v)\|_p^p) - (\|\Lambda(f(Ax^\circ) - v)\|_p^p - \|\Lambda(f(Ax^*) - v)\|_p^p)\right|$$
$$\lesssim \varepsilon^4 \cdot (\frac{1000}{\varepsilon^2}\mathsf{OPT} + \|\Lambda Ax^*\|_p^p)$$
$$\lesssim \varepsilon^2 \cdot (\mathsf{OPT} + \|\Lambda Ax^*\|_p^p).$$
Then, we have
$$\|\Lambda Ax^\circ\|_p^p \le \frac{1}{\varepsilon^2}\left(\|S^\circ \Lambda(f(Ax^*) - b)\|_p^p - \|S^\circ \Lambda(f(Ax^\circ) - b)\|_p^p\right) + \|\Lambda Ax^*\|_p^p \quad \text{by the optimality of } x^\circ$$
$$\lesssim \frac{1}{\varepsilon^2}\left(\|\Lambda(f(Ax^\circ) - v)\|_p^p - \|\Lambda(f(Ax^*) - v)\|_p^p + \varepsilon^2 \cdot (\mathsf{OPT} + \|\Lambda Ax^*\|_p^p)\right) + \|\Lambda Ax^*\|_p^p$$
$$\lesssim \mathsf{OPT} + \|\Lambda Ax^*\|_p^p \quad \text{by the optimality of } x^*. \tag{63}$$
Plugging (62) and (63) into (61), we have
$$\|\Lambda(f(A\hat{x}) - b)\|_p^p - \mathsf{OPT} \lesssim \varepsilon \cdot (\mathsf{OPT} + \|\Lambda Ax^*\|^p).$$
This completes the proof for the query complexity without dependence on $n$. The overall failure probability is at most 0.1 in the above argument.

## C  LOWER BOUND

### C.1  CASE OF $p \in [1, 2]$

By Yao's minimax theorem, it suffices to show the following theorem.

**Theorem 22.** *Suppose that $p \geq 1$, $\varepsilon > 0$ is sufficiently small and $n \gtrsim_p (d \log d)/\varepsilon^2$. There exist a deterministic function $f \in \mathsf{Lip}_1$, a deterministic matrix $A \in \mathbb{R}^{n \times d}$ and a distribution over $b \in \mathbb{R}^n$ such that the following holds: every deterministic algorithm that outputs $\hat{x} \in \mathbb{R}^d$ which with probability at least $4/5$ over the randomness of $b$ satisfies (20) must make $\Omega(d/\varepsilon^2)$ queries to the entries of $b$.*

We remark that the lower bound holds for all $p \geq 1$, and is tight up to logarithmic factors for $p \in [1, 2]$. To prove Theorem 22, we reduce Problem 23 below to our problem.

**Problem 23.** *Suppose that $0 < \varepsilon < 1$, $d$ is a positive integer, $m \sim (\log d)/\varepsilon^2$ and $n = 2md$. Let*

$$u = \begin{bmatrix} 3 \\ 2 \end{bmatrix} \quad and \quad v = \begin{bmatrix} 2 \\ 3 \end{bmatrix}.$$

*Let $D_0$ be the distribution on the $(2m)$-dimensional vector $b'$ such that, for $i = 1, \ldots, m$,*

$$\begin{bmatrix} b'_{2i-1} \\ b'_{2i} \end{bmatrix} = \begin{cases} u & \text{with probability } \frac{1}{2} + \varepsilon \\ v & \text{with probability } \frac{1}{2} - \varepsilon \end{cases}$$

*and $D_1$ be the distribution on the $(2m)$-dimensional vector $b'$ such that, for $i = 1, \ldots, m$,*

$$\begin{bmatrix} b'_{2i-1} \\ b'_{2i} \end{bmatrix} = \begin{cases} u & \text{with probability } \frac{1}{2} - \varepsilon \\ v & \text{with probability } \frac{1}{2} + \varepsilon. \end{cases}$$

*Let $b$ be the $n$-dimensional random vector formed by concatenating $d$ i.i.d. random vectors $b^{(1)}, \ldots, b^{(d)}$, where each $b^{(i)}$ is drawn from $D_0$ with probability $1/2$ and from $D_1$ with probability $1/2$.*

*Given a query access to the entries of $b$, we would like to, with probability at least $2/3$, correctly identify whether $b^{(i)}$ is drawn from $D_0$ or $D_1$ for at least $2d/3$ indices $i$.*

By Lemma 9 and Lemma 11, any deterministic algorithm that solves this problem requires $\Omega(d/\varepsilon^2)$ queries to the entries of $b$.

Now, we construct the reduction. Let $f$ be the function

$$f(x) = \begin{cases} 2 & \text{if } x \leq -6 \\ -x - 4 & \text{if } -6 \leq x \leq -4 \\ 0 & \text{if } -4 \leq x \leq 0 \\ \frac{1}{2}x & \text{if } 0 \leq x \end{cases}$$

Let $a$ be the $(2m)$-dimensional vector such that

$$\begin{bmatrix} a_{2i-1} \\ a_{2i} \end{bmatrix} = \begin{bmatrix} 1 \\ -1 \end{bmatrix} \quad \text{for } i = 1, \ldots, m$$

and $A$ be the $n \times d$ block-diagonal matrix whose diagonal blocks are the same $a \in \mathbb{R}^{2m \times 1}$.

Given a deterministic algorithm $\mathcal{A}$ that takes $f, A, \varepsilon$ and a query access to the entries of $b$ as inputs and returns $\hat{x} \in \mathbb{R}^d$ satisfying (20), we claim that $\hat{x}$ can be used to solve Problem 23. This claim is proved in Lemma 24.

**Lemma 24.** *Let $f, A, b$ be as specified above. There exists $K = K(p)$, a constant only depending on $p$ such that given an $\hat{x} \in \mathbb{R}^d$ satisfying*

$$\|f(A\hat{x}) - b\|_p^p \leq (1 + \frac{\varepsilon}{K})\mathsf{OPT} + \frac{\varepsilon}{K}\|Ax^*\|_p^p,$$

*we can, with probability at least $99/100$ (over the randomness of $b$), identify whether $b^{(i)}$ is drawn from $D_0$ or $D_1$ for at least $2d/3$ indices $i$.*

We need the next lemma, whose proof is postponed to Appendix C.3, to prove Lemma 24.

**Lemma 25.** *Let $b'$ be a $(2m)$-dimensional vector in which*

$$\begin{bmatrix} b'_{2i-1} \\ b'_{2i} \end{bmatrix} = u \text{ or } v, \quad i = 1, \dots, m.$$

*Then*

 (a) *it holds for $x \leq 0$ that $\|f(ax) - b'\|_p^p \geq \|f(-6a) - b'\|_p^p$, and*

 (b) *it holds for $x \geq 0$ that $\|f(ax) - b'\|_p^p \geq \|f(6a) - b'\|_p^p$.*

Now we are ready to prove Lemma 24.

*Proof of Lemma 24.* To prove the statement, we first give a bound for OPT. We have

$$\text{OPT} = \min_{x \in \mathbb{R}^d} \|f(Ax) - b\|_p^p = \sum_{i=1}^d \min_{x_i \in \mathbb{R}} \|f(ax_i) - b^{(i)}\|_p^p \quad \text{by the structure of } A \qquad (64)$$

and hence we can look at each term $\min_{x_i \in \mathbb{R}} \|f(ax_i) - b\|_p^p$ individually. By Lemma 25, we have

$$\min_{x_i \in \mathbb{R}} \|f(ax_i) - b^{(i)}\|_p^p = \min\{\|f(-6a) - b^{(i)}\|_p^p, \|f(6a) - b^{(i)}\|_p^p\}. \qquad (65)$$

For $i = 1, \dots, d$, let $k_i$ be the number of occurrences of $u = \begin{bmatrix} 3 \\ 2 \end{bmatrix}$ in $b^{(i)}$. Recall that $m \sim (\log d)/\varepsilon^2$. By choosing the hidden constant to be large enough, we have by a Chernoff bound that for every $i = 1, \dots, d$, with probability at least $1 - \frac{1}{100d}$,

$$k_i \in \begin{cases} [\frac{m}{2} + (1-\beta)\varepsilon m, \frac{m}{2} + (1+\beta)\varepsilon m] & \text{if } b^{(i)} \text{ is drawn from } D_0 \\ [\frac{m}{2} - (1+\beta)\varepsilon m, \frac{m}{2} - (1-\beta)\varepsilon m] & \text{if } b^{(i)} \text{ is drawn from } D_1 \end{cases}, \qquad (66)$$

where $\beta > 0$ is a constant to be determined. Taking a union bound, with probability at least $99/100$, every $k_i$ satisfies this condition. We condition on this event below.

Note that

$$\|f(6a) - b^{(i)}\|_p^p = 2(m - k_i) \quad \text{and} \quad \|f(-6a) - b^{(i)}\|_p^p = 2k_i$$

By (66), if $b^{(i)}$ is drawn from $D_0$, we have

$$m - 2(1+\beta)\varepsilon m \leq \|f(6a) - b^{(i)}\|_p^p \leq m - 2(1-\beta)\varepsilon m$$
$$\|f(-6a) - b^{(i)}\|_p^p \geq m + 2(1-\beta)\varepsilon m.$$

Similarly, if $b^{(i)}$ is drawn from $D_1$, we have

$$\|f(6a) - b^{(i)}\|_p^p \geq m + 2(1-\beta)\varepsilon m$$
$$m - 2(1+\beta)\varepsilon m \leq \|f(-6a) - b^{(i)}\|_p^p \leq m - 2(1-\beta)\varepsilon m.$$

By plugging them into (65) and (64), we have

$$\text{OPT} \leq d\left(m - 2(1-\beta)\varepsilon m\right).$$

Now, suppose that a solution $\hat{x}$ satisfies

$$\begin{aligned}
\|f(A\hat{x}) - b\|_p^p &\leq (1 + \frac{\varepsilon}{K})\text{OPT} + \frac{\varepsilon}{K}\|Ax^*\|_p^p \\
&\leq \left(1 + \frac{\varepsilon}{K}\right) d\left(m - 2(1-\beta)\varepsilon m\right) + \frac{\varepsilon}{K} \cdot 2md \cdot 6^p \\
&\leq md - \left(2(1-\beta) - \frac{C_p}{K}\right)\varepsilon md \\
&\leq md - (2 - 3\beta)\varepsilon md,
\end{aligned}$$

provided that $K \geq C_p/\beta$. Here $C_p$ is a constant that depends only on $p$.

We declare $b^{(i)}$ is drawn from $D_0$ if $\hat{x}_i > 0$ and from $D_1$ otherwise. Suppose that our declaration is wrong on $\ell$ indices, then by Lemma 25,

$$\|f(A\hat{x}) - b\|_p^p \geq \ell \cdot (m + 2(1 - \beta)\varepsilon m) + (d - \ell) \cdot (m - 2(1 + \beta)\varepsilon m)$$
$$= md - 2(1 + \beta)\varepsilon md + 4\varepsilon \ell m.$$

Therefore,

$$md - 2(1 + \beta)\varepsilon md + 4\varepsilon \ell m \leq md - (2 - 3\beta)\varepsilon md$$

which implies that

$$\ell \leq \frac{5}{4}\beta d.$$

We can conclude that we have used an approximate solution $\hat{x}$ to deduce the distribution of $b^{(i)}$ for at least $(1 - 5\beta/4)d$ indices of $i = 1, \ldots, d$. Choosing $\beta = 4/15$ and $K = 4C_p$ completes the proof of Lemma 24. $\qquad \square$

To finish the proof of Theorem 22, by Lemma 24, with probability $1 - 1/100 - 1/5 > 2/3$, we can correctly identify whether $b^{(i)}$ is drawn from $D_0$ or $D_1$ for at least $2d/3$ indices $i$, i.e. we solve Problem 23. Hence, we conclude that $\mathcal{A}$ must make $\Omega(d/\varepsilon^2)$ queries to the entries of $b$.

## C.2 CASE OF $p \geq 2$

By Yao's minimax theorem, it suffices to show the following theorem.

**Theorem 26.** *Suppose that $p \geq 1$, $\varepsilon > 0$ is sufficiently small and $n \gtrsim_p d/\varepsilon^p$. There exist a deterministic function $f \in \mathsf{Lip}_1$, a deterministic matrix $A \in \mathbb{R}^{n \times d}$ and a distribution over $b \in \mathbb{R}^n$ such that the following holds: every deterministic algorithm that outputs $\hat{x} \in \mathbb{R}^d$ which with probability at least $2/3$ over the randomness of $b$ satisfies (20) must make $\Omega(d/\varepsilon^p)$ queries to the entries of $b$.*

We remark that the lower bound holds for all $p \geq 1$. To prove Theorem 26, we reduce Problem 27 below to our problem.

**Problem 27.** *Suppose that $0 < \varepsilon < 1$, $d$ is a positive integer, $m = 1/\varepsilon^p$ and $n = 2md$. Let $v$ be the $2m$-dimensional vector whose entries are all 1, i.e. $v = [1, \ldots, 1]^\top$, $D_0$ be the uniform distribution on $\{v + (1/\varepsilon) \cdot e_1, \ldots, v + (1/\varepsilon) \cdot e_m\}$ and $D_1$ be the uniform distribution on $\{v + (1/\varepsilon) \cdot e_{m+1}, \ldots, v + (1/\varepsilon) \cdot e_{2m}\}$ where $e_1, \ldots, e_{2m}$ are the canonical basis vectors in $\mathbb{R}^{2m}$. Let $b$ be the $n$-dimensional random vector formed by concatenating $d$ i.i.d. random vectors $b^{(1)}, \ldots, b^{(d)}$, where each $b^{(i)}$ is drawn from $D_0$ with probability $1/2$ and from $D_1$ with probability $1/2$, i.e. each $b^{(i)}$ is an all one vector with a value $1/\varepsilon$ planted at a uniformly random entry.*

*Given a query access to the entries of $b$, we would like to, with probability at least $2/3$, correctly identify whether $b^{(i)}$ is drawn from $D_0$ or $D_1$ for at least $2d/3$ indices $i$.*

By Lemma 10 and Lemma 11, any deterministic algorithm that solves this problem requires $\Omega(d/\varepsilon^p)$ queries to the entries of $b$.

Now, we construct the reduction. Let $f$ be the function

$$f(x) = \begin{cases} 0 & \text{if } x \leq 0 \\ x & \text{if } 0 \leq x \leq 1 \\ 1 & \text{if } 1 \leq x. \end{cases}$$

Let $a$ be the $2m$-dimensional vector such that

$$a_i = \begin{cases} 1 & \text{for } i = 1, \ldots, m \\ -1 & \text{for } i = m+1, \ldots, 2m \end{cases}$$

and $A$ be the $n \times d$ block-diagonal matrix whose diagonal blocks are the same $a \in \mathbb{R}^{2m \times 1}$.

Given a deterministic algorithm $\mathcal{A}$ that takes $f, A, \varepsilon$ and a query access to the entries of $b$ as inputs and returns $\hat{x} \in \mathbb{R}^d$ satisfying (20), we claim that $\hat{x}$ can be used to solve Problem 27. This claim is proved in Lemma 28.

**Lemma 28.** *Let $f$, $A$, $b$ be as specified above. There exists $K$, a constant, such that given an $\hat{x} \in \mathbb{R}^d$ satisfying*

$$\|f(A\hat{x}) - b\|_p^p \leq (1 + \frac{\varepsilon}{K})\mathsf{OPT} + \frac{\varepsilon}{K}\|Ax^*\|_p^p,$$

*we can identify whether $b^{(i)}$ is drawn from $D_0$ or $D_1$ for at least $2d/3$ indices $i$.*

We need the next lemma, whose proof is postponed to Appendix C.3.

**Lemma 29.** *Let $b'$ be a $2m$-dimensional vector in which all entries are $1$ except one of them is $1 + \frac{1}{\varepsilon}$. Then*

(a) *it holds for $x \leq 0$ that $\|f(ax) - b'\|_p^p \geq \|f(-a) - b'\|_p^p$, and*

(b) *it holds for $x \geq 0$ that $\|f(ax) - b'\|_p^p \geq \|f(a) - b'\|_p^p$.*

Now we are ready to prove Lemma 28.

*Proof of Lemma 28.* To prove the statement, we first give a bound for OPT. We have

$$\mathsf{OPT} = \min_{x \in \mathbb{R}^d}\|f(Ax) - b\|_p^p = \sum_{i=1}^d \min_{x_i \in \mathbb{R}}\|f(ax_i) - b^{(i)}\|_p^p \tag{67}$$

and hence we can look at each term $\min_{x_i \in \mathbb{R}}\|f(ax_i) - b\|_p^p$ individually. By Lemma 29, we have

$$\min_{x_i \in \mathbb{R}}\|f(ax_i) - b^{(i)}\|_p^p = \min\{\|f(-a) - b^{(i)}\|_p^p, \|f(a) - b^{(i)}\|_p^p\}. \tag{68}$$

Recall that $m = \frac{1}{\varepsilon^p}$. For $i = 1, \ldots, d$, if $b^{(i)}$ is drawn from $D_0$, we have

$$\|f(a) - b^{(i)}\|_p^p = m + \frac{1}{\varepsilon^p} = 2m \quad \text{and} \quad \|f(-a) - b^{(i)}\|_p^p = (1 + \frac{1}{\varepsilon})^p + m - 1 \geq (2 + \varepsilon)m$$

and, if $b^{(i)}$ is drawn from $D_1$, we have

$$\|f(-a) - b^{(i)}\|_p^p = m + \frac{1}{\varepsilon^p} = 2m \quad \text{and} \quad \|f(a) - b^{(i)}\|_p^p = (1 + \frac{1}{\varepsilon})^p + m - 1 \geq (2 + \varepsilon)m.$$

By plugging them into (68) and (67), we have

$$\mathsf{OPT} \leq 2dm.$$

Now, suppose that a solution $\hat{x}$ satisfies

$$\begin{aligned}\|f(A\hat{x}) - b\|_p^p &\leq (1 + \frac{\varepsilon}{K})\mathsf{OPT} + \frac{\varepsilon}{K}\|Ax^*\|_p^p \\ &\leq (1 + \frac{\varepsilon}{K}) \cdot (2dm) + \frac{\varepsilon}{K} \cdot 2dm \cdot 1^p \\ &\leq 2dm + \frac{4\varepsilon dm}{K}.\end{aligned}$$

We declare that $b^{(i)}$ is drawn from $D_0$ if $\hat{x}_i > 0$ and from $D_1$ otherwise. Suppose that our declaration is wrong on $\ell$ indices, then by Lemma 28,

$$\|f(A\hat{x}) - b\|_p^p \leq \ell(2 + \varepsilon)m + (d - \ell)(2m) = 2dm + \varepsilon\ell m.$$

Therefore,

$$2dm + \varepsilon\ell m \leq 2dm + \frac{4\varepsilon dm}{K},$$

which implies, when $K = 12$, that

$$\ell \leq \frac{d}{3},$$

completing the proof of Lemma 28. $\qquad\square$

To finish the proof of Theorem 26, by Lemma 28, with probability $2/3$, we can correctly identify whether $b^{(i)}$ is drawn from $D_0$ or $D_1$ for at least $2d/3$ indices $i$, i.e. we solve Problem 27. Hence, we conclude that $\mathcal{A}$ must make $\Omega(d/\varepsilon^p)$ queries to the entries of $b$.

## C.3 OMITTED PROOFS

*Proof of Lemma 25.* Suppose that $b'$ contains $k$ occurrences of $u = \begin{bmatrix} 3 \\ 2 \end{bmatrix}$ and $\theta = k/m$. Then

$$\frac{1}{m}\|f(ax) - b'\|_p^p = \theta(g(x))^p + (1-\theta)(h(x))^p,$$

where

$$g(x) = \left\| \begin{bmatrix} f(x) \\ f(-x) \end{bmatrix} - \begin{bmatrix} 3 \\ 2 \end{bmatrix} \right\|_p \quad \text{and} \quad h(x) = \left\| \begin{bmatrix} f(x) \\ f(-x) \end{bmatrix} - \begin{bmatrix} 2 \\ 3 \end{bmatrix} \right\|_p.$$

It suffices to show that both $h(x)$ and $g(x)$ attain a local minimum at $x = -6$ when $x \leq 0$ and at $x = 6$ when $x \geq 0$.

Now, we view the 2-dimensional vectors as points in $\mathbb{R}^2$. For any $x \in \mathbb{R}$, let $\zeta(x)$ be the point $(f(x), f(-x))$. Also, let $\gamma \subset \mathbb{R}^2$ be the locus of $\zeta(x) = (f(x), f(-x))$, i.e. $\gamma := \{\zeta(x) \mid x \in \mathbb{R}\}$. It has a positive branch $\gamma^+ := \{\zeta(x) \mid x \geq 0\}$ and a negative branch $\gamma^- := \{\zeta(x) \mid x \leq 0\}$.

We first consider $x \leq 0$. Note that $\zeta(-6) = (2,3)$ is on $\gamma^-$ and $x = -6$ is the only value such that $\zeta(x) = (2,3)$. Hence, we immediately have that $h(x) = 0$ attains a local minimum at $x = -6$. For $g(x)$, consider the smallest $\ell_p$-ball centered at $(3,2)$ that touches $(2,3)$ on its boundary. It is not difficult to verify that this $\ell_p$-ball does not intersect $\gamma^-$ at any other point. (Figure 2 provides a geometric intuition.)

Since $\gamma^+$ and $\gamma^-$ are symmetric about $y = x$, we can show the symmetric result for $x \geq 0$.

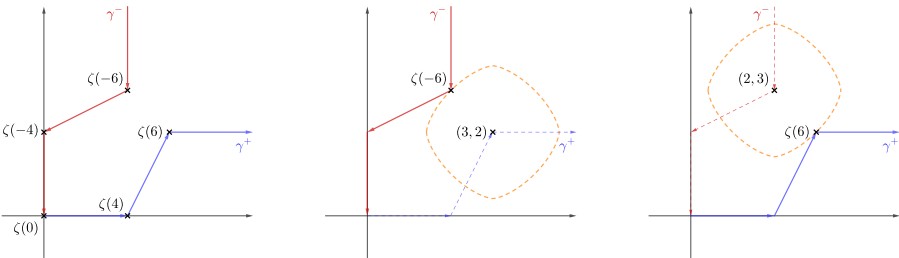

Figure 2: Illustration of the locus $\gamma$ (left), the minimizers when $x \leq 0$ (middle) and the minimizer when $x \geq 0$ (right)

$\square$

*Proof of Lemma 29.* We would like to show that the function $\|f(ax) - b'\|_p^p$ attains a local minimum at $x = -1$ when $x \leq 0$ and at $x = 1$ when $x \geq 0$. Note that, by the construction of $f$, $f(x) = f(1) = 1$ for all $x \geq 1$ and $f(x) = f(-1) = 0$ for all $x \leq -1$. Therefore, we now restrict our domain to be $x \in [-1, 1]$.

Suppose that the index of the entry whose value is $1 + \frac{1}{\varepsilon}$ is in $\{1, \ldots, m\}$. Then

$$\|f(ax) - b'\|_p^p = (1 + \frac{1}{\varepsilon} - f(x))^p + (m-1)(1 - f(x))^p + m(1 - f(-x))^p.$$

Note that we drop the absolute value sign because $f(x) \leq 1$ on $\mathbb{R}$. When $-1 \leq x \leq 0$, we have

$$\|f(ax) - b'\|_p^p = (1 + \frac{1}{\varepsilon})^p + (m-1) + m(1+x)^p$$

$$\geq (1 + \frac{1}{\varepsilon})^p + (m-1).$$

where the equality holds if $x = -1$. When $0 \leq x \leq 1$, we have

$$\|f(ax) - b'\|_p^p = (1 + \frac{1}{\varepsilon} - x)^p + (m-1)(1-x)^p + m$$

$$\geq \frac{1}{\varepsilon^p} + m$$

where the equality holds if $x = 1$.

Similarly, we can prove the same result when the index of the entry whose value is $1 + \frac{1}{\varepsilon}$ is in $\{m + 1, \ldots, 2m\}$. $\qquad\square$

## D  LEWIS WEIGHTS OF ROW-SAMPLED MATRIX

In this section, we shall show the following theorem.

**Theorem 30.** *Suppose that $A \in \mathbb{R}^{n \times d}$ has uniformly bounded Lewis weights $w_i(A) \lesssim d/n$. Let $S$ be an $n \times n$ diagonal matrix in which the diagonal elements are i.i.d. $\alpha^{-1/p} \operatorname{Ber}(\alpha)$ for some sampling rate $\alpha \in (0, 1)$. When $\alpha n \gtrsim_p d \log d$ when $p < 4$ or $\alpha n \gtrsim_p d^2 \log d$ when $p \geq 4$, it holds with probability at least $0.99$ that $w_i(SA) \lesssim d/m$, where $m$ is the number of nonzero rows of $S$.*

This theorem is similar to (Chen et al., 2022, Lemma A.3), where the sampling rates are proportional to Lewis weights and no assumptions on the bounds of $w_i(A)$ were made. Our proof is also similar.

*Proof.* Let $w_1, \ldots, w_n$ denote the Lewis weights of $A$ and suppose that $S = \operatorname{diag}\{\sigma_1, \ldots, \sigma_n\}$. We first show that with probability at least $0.995$,

$$\frac{1}{2} \sum_i w_i^{1-\frac{2}{p}} a_i a_i^\top \preceq \sum_i \left(\frac{w_i}{\alpha}\right)^{1-\frac{2}{p}} (\sigma_i a_i)(\sigma_i a_i)^\top \preceq \frac{3}{2} \sum_i w_i^{1-\frac{2}{p}} a_i a_i^\top.$$

Here, the $\preceq$ sign denotes semi-positive definiteness. We prove this claim by the matrix Bernstein inequality. Notice that

$$\sum_i \left(\frac{w_i}{\alpha}\right)^{1-\frac{2}{p}} (\sigma_i a_i)(\sigma_i a_i)^\top = \sum_i \frac{\xi_i}{\alpha} w_i^{1-\frac{2}{p}} a_i a_i^\top,$$

where $\xi_i$ are i.i.d. $\operatorname{Ber}(\alpha)$ variables. Let $M = \sum_i w_i^{1-\frac{2}{p}} a_i a_i^\top$, $a_i' = M^{-\frac{1}{2}} a_i$, and $X_i = \frac{\xi_i}{\alpha} w_i^{1-\frac{2}{p}} a_i'(a_i')^\top - w_i^{1-\frac{2}{p}} a_i'(a_i')^\top$, then $\mathbb{E} X_i = 0$ and, by the definition of Lewis weights, $\|a_i'\|^2 = w_i^{2/p}$. We bound

$$\|X_i\|_2 \leq \frac{1}{\alpha} w_i^{1-\frac{2}{p}} \|a_i'\|_2^2 = \frac{1}{\alpha} w_i \lesssim \frac{d}{\alpha n}.$$

Also,

$$\left\|\sum_i \mathbb{E} X_i X_i^\top\right\|_2 \lesssim \frac{1}{\alpha} \left\|\sum_i w_i^{2(1-\frac{2}{p})} \|a_i\|_2^2 a_i'(a_i')^\top\right\|_2 = \frac{1}{\alpha} \left\|\sum_i w_i \cdot w_i^{1-\frac{2}{p}} a_i'(a_i')^\top\right\|_2$$

$$\lesssim \frac{d}{\alpha n} \left\|\sum_i w_i^{1-\frac{2}{p}} a_i'(a_i')^\top\right\|_2$$

$$= \frac{d}{\alpha n} \|I\|_2$$

$$= \frac{d}{\alpha n}.$$

It follows from matrix Bernstein inequality that

$$\Pr\left\{\left\|\sum_i X_i\right\|_2 \geq \eta\right\} \leq 2d \exp\left(-c\frac{\alpha n \eta^2}{d}\right) \leq 0.005,$$

provided that $\alpha n \gtrsim \eta^{-2} d \log d$. This shows that

$$(1 - \eta)I \preceq \sum_i \frac{\xi_i}{\alpha} w_i^{1-\frac{2}{p}} a_i'(a_i')^\top \preceq (1 + \eta)I,$$

which is equivalent to our claim.

When $\sigma_i > 0$,

$$(\sigma_i a_i)^\top \left( \sum_j \left( \frac{w_j}{\alpha} \right)^{1-\frac{2}{p}} (\sigma_j a_j)(\sigma_j a_j)^\top \right)^{-1} (\sigma_i a_i)$$

$$\leq \frac{1}{1-\eta} \sigma_i^2 a_i^\top \left( \sum_j w_j^{1-\frac{2}{p}} a_j a_j^\top \right)^{-1} a_i$$

$$= \frac{1}{1-\eta} \sigma_i^2 w_i^{\frac{2}{p}}$$

$$= \frac{1}{1-\eta} \left( \frac{w_i}{\alpha} \right)^{\frac{2}{p}}.$$

and, similarly,

$$(\sigma_i a_i)^\top \left( \sum_j (\frac{w_j}{\alpha})^{1-\frac{2}{p}} (\sigma_j a_j)(\sigma_j a_j)^\top \right)^{-1} (\sigma_i a_i) \geq \frac{1}{1+\eta} \left( \frac{w_i}{\alpha} \right)^{\frac{2}{p}}.$$

We take $\eta$ to be a constant depending on $p$ for $p < 4$ and $\eta = 1/(C_p \sqrt{d})$ for $p \geq 4$. It then follows from (Cohen & Peng, 2015, Lemmas 5.3 and 5.4) that $w_i(SA) \lesssim w_{i'}(A)/\alpha \lesssim d/(\alpha n)$, where $i'$ is the index of the corresponding row in $A$.

By a Chernoff bound, with probability at least $0.995$, $m \lesssim \alpha n$. The result then follows. □

## E    Entropy Estimates

In this section we provide a proof of Lemma 8 for completeness. The proof is decomposed into the following three lemmata, Lemma 31, Lemma 32 and Lemma 33.

**Lemma 31.** *Suppose that $A \in \mathbb{R}^{n \times d}$ has full column rank and $W$ is a diagonal matrix whose diagonal entries are the Lewis weights of $A$. It holds for $p \geq 1$ that*

$$\log \mathcal{N}(B_{w,p}(W^{-1/p}A), \|\cdot\|_{w,p}, t) \lesssim d \log \frac{1}{t}.$$

*Proof.* This is a standard result following from a standard volume argument, which we reproduce below for completeness. Suppose that $E$ is the column space of $W^{-1/p}A$ and is endowed with norms $\|\cdot\|_{w,p}$. Using the notation simplification defined in Appendix A, we denote by $B_{w,p}$ the unit ball of $E$ w.r.t. $\|\cdot\|_{w,q}$, i.e. $B_{w,p} = B_{w,p}(W^{-1/p}A)$. Consider a maximal $t$-separation set $N \subseteq B_{w,p}$, then the balls $x + \frac{t}{2}B_{w,p}$ ($x \in N$) are contained in $(1 + \frac{t}{2})B_{w,p}$ and are nearly disjoint (intersection has zero volume). Hence $\sum_{x \in N} \text{vol}(x + \frac{t}{2}B_{w,p}) \leq \text{vol}((1 + \frac{t}{2})B_{w,p})$, that is, $|N| \cdot \text{vol}(\frac{t}{2}B_{w,p}) \leq \text{vol}((1 + \frac{t}{2})B_{w,p})$, leading to $|N| \leq (1 + 2/t)^d$. It is easy to check that $N$ is a $t$-covering of $(B_{w,p}, \|\cdot\|_{w,p})$ and it implies $\mathcal{N}(B_{w,p}(W^{-1/p}A), \|\cdot\|_{w,p}, t) \leq |N| \leq (1 + 2/t)^d$.

□

**Lemma 32.** *Suppose that $A \in \mathbb{R}^{n \times d}$ has full column rank and $W$ is a diagonal matrix whose diagonal entries are the Lewis weights of $A$. When $1 \leq p \leq 2$ and $q > 2$, it holds that*

$$\log \mathcal{N}(B_{w,p}(W^{-1/p}A), \|\cdot\|_{w,q}, t) \lesssim \frac{q\sqrt{\log d}}{t^p}$$

*Proof.* Suppose that $E$ is the column space of $W^{-1/p}A$ and is endowed with norms $\|\cdot\|_{w,p}$ and an inner product $\langle \cdot, \cdot \rangle_w$. We first have

$$\log \mathcal{N}(B_{w,p}, \|\cdot\|_{w,q}, t) \leq \log \mathcal{N}(B_{w,p}, \|\cdot\|_{w,2}, \lambda) + \log \mathcal{N}\left( B_{w,2}, \|\cdot\|_{w,q}, \frac{t}{\lambda} \right).$$

Recall that $B_{w,p} = B_{w,p}(W^{-1/p}A)$ and $B_{w,2} = B_{w,2}(W^{-1/p}A)$. For the second term, we can apply Lemma 33 directly and obtain that

$$\log \mathcal{N}\left(B_{w,2}, \|\cdot\|_{w,q}, \frac{t}{\lambda}\right) \lesssim \frac{\lambda^2}{t^2} d^{2/q} q.$$

Next we deal with the first term.

We first consider the case $1 < p \leq 2$. Let $p'$ be the conjugate index of $p$ and $r \geq p'$ to be determined. Define $\theta \in [0, 1]$ by

$$\frac{1}{p'} = \frac{1-\theta}{2} + \frac{\theta}{r}.$$

For $x, y \in B_{w,2}$, by Hölder's inequality,

$$\|x - y\|_{w,p'} \leq \|x - y\|_{w,2}^{1-\theta} \|x - y\|_{w,r}^{\theta} \leq 2^{1-\theta} \|x - y\|_{w,r}^{\theta}.$$

This implies that

$$\log \mathcal{N}\left(B_{w,2}, \|\cdot\|_{w,p'}, \lambda\right) \leq \log \mathcal{N}\left(B_{w,2}, \|\cdot\|_{w,r}, 2\left(\frac{\lambda}{2}\right)^{1/\theta}\right) \lesssim \left(\frac{2}{\lambda}\right)^{2/\theta} r d^{2/r},$$

where the last inequality follows from Lemma 33. Since

$$\frac{1}{\theta} = \frac{p'}{p'-2}\left(1 - \frac{2}{r}\right) = \frac{p}{2-p}\left(1 - \frac{2}{r}\right),$$

it follows that

$$\log \mathcal{N}\left(B_{w,2}, \|\cdot\|_{w,p'}, \lambda\right) \lesssim \left(\frac{2}{\lambda}\right)^{\frac{2p}{2-p}} r d^{2/r}.$$

Choose $r = p' \vee (\log d)$,

$$\log \mathcal{N}\left(B_{w,2}, \|\cdot\|_{w,p'}, \lambda\right) \lesssim \left(\frac{2}{\lambda}\right)^{\frac{2p}{2-p}} r.$$

By duality (Artstein et al., 2004),

$$\log \mathcal{N}(B_{w,p}, \|\cdot\|_{w,2}, \lambda) \lesssim \left(\frac{2}{\lambda}\right)^{\frac{2p}{2-p}} r.$$

Therefore,

$$\log \mathcal{N}(B_{w,p}, \|\cdot\|_{w,q}, t) \lesssim \left(\frac{2}{\lambda}\right)^{\frac{2p}{2-p}} r + \frac{\lambda^2}{t^2} d^{2/q} q.$$

Optimizing $\lambda$ yields that

$$\log \mathcal{N}(B_{w,p}, \|\cdot\|_{w,q}, t) \lesssim \frac{q^{p/2} r^{1-p/2}}{t^p} \lesssim \left(\frac{1}{\sqrt{p-1}} + \sqrt{\log d}\right) \cdot \frac{q}{t^p}.$$

This completes the proof for $1 < p \leq 2$.

When $p = 1$, Maurey's empirical method gives that (using the fact that, by Lemma 5(c), $\|\cdot\|_{w,2} \leq \|\cdot\|_{w,1}$ in $E$)

$$\log \mathcal{N}(B_{w,1}, \|\cdot\|_{w,2}, \lambda) \lesssim \frac{\log d}{\lambda^2}$$

and thus

$$\log \mathcal{N}(B_{w,1}, \|\cdot\|_{w,q}, t) \lesssim \frac{\log d}{\lambda^2} + \frac{\lambda^2}{t^2} d^{2/q} q.$$

Optimizing $\lambda$ yields that

$$\log \mathcal{N}(B_{w,1}, \|\cdot\|_{w,q}, t) \lesssim \frac{\sqrt{q \log d} \cdot d^{1/q}}{t}.$$

$\square$

**Lemma 33.** *Suppose that $A \in \mathbb{R}^{n \times d}$ has full column rank and $W$ is a diagonal matrix whose diagonal entries are the Lewis weights of A. When $p, q \geq 2$, it holds that*

$$\log \mathcal{N}(B_{w,p}(W^{-1/p}A), \|\cdot\|_{w,q}, t) \lesssim \frac{d^{1 - \frac{2}{p} + \frac{2}{q}} q}{t^2}$$

*Proof.* Suppose that $E$ is the column space of $W^{-1/p}A$ and is endowed with norms $\|\cdot\|_{w,p}$ and an inner product $\langle \cdot, \cdot \rangle_w$. By Lemma 5(b), there exist $u_1, \ldots, u_d \in \mathbb{R}^n$ such that

$$\langle u_i, u_{i'} \rangle_w = 0, \quad \|u_i\|_{w,p} = 1 \quad \text{and} \quad \sum_{j=1}^{d} u_{ij}^2 = 1 \quad \text{for any } i, i' = 1, \ldots, n.$$

Recall that $B_{w,p} = B_{w,p}(W^{-1/p}A)$ and $B_{w,2} = B_{w,2}(W^{-1/p}A)$. First, observe, by Lemma 5(c), that $B_{w,p} \subseteq d^{1/2 - 1/p} \cdot B_{w,2}$, thus

$$\log \mathcal{N}(B_{w,p}, \|\cdot\|_{w,q}, t) \leq \log \mathcal{N}\left(B_{w,2}, \|\cdot\|_{w,q}, \frac{t}{d^{1/2 - 1/p}}\right)$$

and it suffices to show that

$$\log \mathcal{N}\left(B_{w,2}, \|\cdot\|_{w,q}, t\right) \lesssim \frac{q d^{2/q}}{t^2}.$$

Let $q'$ be the conjugate index of $q$, i.e. $\frac{1}{q} + \frac{1}{q'} = 1$. By dual Sudakov minorization,

$$\log \mathcal{N}\left(B_{w,2}, \|\cdot\|_{w,q}, t\right) \lesssim \frac{1}{t^2}\left(\mathbb{E}_{g \sim N(0, I_d)} \sup_{x \in B_{w,q'}} \left|\left\langle \sum_{i=1}^{d} g_i u_i, x \right\rangle_w\right|\right)^2,$$

By duality again,

$$\mathbb{E}_{g \sim N(0, I_d)} \sup_{x \in B_{w,q'}} \left|\left\langle \sum_{i=1}^{d} g_i u_i, x \right\rangle_w\right| = \mathbb{E}_{g \sim N(0, I_d)} \left\|\sum_{i=1}^{d} g_i u_i\right\|_{w,q}.$$

Then,

$$\mathbb{E}_{g \sim N(0, I_d)} \left\|\sum_{i=1}^{d} g_i u_i\right\|_{w,q} = \mathbb{E}_{g \sim N(0, I_d)} \left(\sum_{j=1}^{n} w_j \left|\sum_{i=1}^{d} g_i u_{ij}\right|^q\right)^{1/q}$$

$$\leq \left(\mathbb{E}_{g \sim N(0, I_d)} \sum_{j=1}^{n} w_j \left|\sum_{i=1}^{d} g_i u_{ij}\right|^q\right)^{1/q}$$

$$\leq \left(\left(\sum_{j=1}^{n} w_j \left|\left(\sum_{i=1}^{d} u_{ij}^2\right)^{\frac{1}{2}}\right|^q\right) \cdot \mathbb{E}_{g \sim N(0,1)} |g|^q\right)^{1/q}$$

$$\leq \left(\mathbb{E}_{g \sim N(0,1)} |g|^q\right)^{1/q} \left(\sum_{j=1}^{n} w_j\right)^{1/q} \qquad \text{since } \sum_{i=1}^{d} u_{ij}^2 = 1$$

$$\lesssim \sqrt{q} d^{1/q}.$$

$\square$

