# OpenReview forum: "Near-optimal Active Regression of Single-Index Models"
_ICLR.cc/2025/Conference — ICLR 2025 Poster_

### Official Review · Reviewer_kECa · 2024-10-31

**Soundness:** 3
**Presentation:** 2
**Contribution:** 3
**Rating:** 6
**Confidence:** 2

**Summary:**

This paper studies the problem of learning single-index models in the active learning setting, where given a dataset $(A,b)$ and an activation function $f$, the goal is to output some $\hat{x}$ by querying the entries of $b$ such that $||f(A\hat{x})-b||_p$ is close to $opt$, the error of the best $x \in R^d$. Previous works that study this problem give several upper bounds for the query complexity for different $p$, but can only approximate $opt$ within a constant factor of $C$. In this work, the authors develop active learning algorithms for this problem with different $p$. For $p \in [1,2]$, the query complexity of the algorithms is $\tilde{O}(d/\epsilon^2)$, which matches the lower bound provided by this paper. For $p>2$, the query complexity is $\tilde{O}(d^{p/2}/\epsilon^p)$. In this regime, the paper also gives a lower bound of $\tilde{\Omega}(d/\epsilon^p)$.

**Strengths:**

Learning single-index models and active learning are both central problems in learning theory. Compared to active learning for classification problems, active learning for regression problems is much less understood. This paper gives a nearly optimal query complexity for the corresponding $l_p$ regression problem for a wide range of $p$. The algorithm builds on a similar sample and learn frameworks used by previous works but uses several solid and new ideas to analyze and improve the error guarantee and the query complexity. The results and new techniques would be of interest to ML and theory communities.

**Weaknesses:**

I think one main weakness of the paper is its presentation. The paper itself is very technical and involves deep mathematical tools such as  Lewis weights that are frequently used for other problems such as subspace embedding. As a person who does not work on these problems and is not familiar with these tools, I found sometimes it hard to go over the paper as the related background is not clearly provided in the main body. Furthermore, I feel that to fully understand the techniques, the paper requires the reader to be familiar with the line of the previous work. For example, in line 219-line 226
>we obtain the following concentration bound .... which has been the central tool in the work on subspace embeddings... In short, when the number of queries is m...

These statements are confusing for a reader who is not familiar with the technique and related literature. It is hard to get an intuition of how to relate the query complexity with the previous error analysis.

2. Another weakness I found is the computational complexity of the problem. I think to implement the algorithm in the paper, one has to solve a sub-optimization problem, which I believe has a super-polynomial time complexity with respect to the error parameter $\epsilon$. In fact, there have been works that show that agnostic learning single index model is usually computationally hard. (see my questions below) Given these hardness results, I am not sure whether it is still motivated to pursue $(1+\epsilon)$-approximation, such a strong guarantee. I think adding some numerical simulations for the algorithm would be helpful to show the algorithm is useful and it is still reasonable goal to achieve $(1+\epsilon)$-approximation for some reasonable choice of $\epsilon$.

**Questions:**

1. I would like to suggest the authors improve the presentation and move some related background from the appendix to the main body of the paper.

2. There have been papers such as [DKR23] that show that even if the data is sampled from a Gaussian and the activation function $f$ is a ReLU, it is computationally hard to achieve $opt+\epsilon$. Can you help explain the relation between these hardness results and the results in this paper? Given the hardness result, why it is still motivated to design algorithms that learns a $(1+\epsilon)$-approximate solution?




>Diakonikolas, Ilias, Daniel Kane, and Lisheng Ren. "Near-optimal cryptographic hardness of agnostically learning halfspaces and relu regression under gaussian marginals." International Conference on Machine Learning. PMLR, 2023.

---

> ### Author Response · Authors · 2024-11-17
>
> We thank the reviewer for the comments. Below we address the reviewer's questions.
>
>
> - **Presentation:** We agree with the reviewer that the paper is very technical and providing  background on the mathematical tools such as Lewis weights helps readers unfamiliar with these concepts better understand our result.
> We will move this background from the appendix to the main text with additional explanations to improve the presentation.
>
>     In Line 219 - Line 226, the techniques used to upper bound $\sup_{x \in T} \operatorname{Err}(x)$ have been employed in several earlier works related to the active regression problem (such as (Yasuda, 2024)) or subspace embeddings.
>     Therefore, we intentionally phrased them as black boxes as they are not our main contribution. We will make this clearer in the revised version.
>     The most novel contribution lies in the section on ``further improvement'', which enables us to achieve a near-optimal dependence on $\epsilon$ in query complexity.
>
> - **Motivation for $(1+\epsilon)$-approximation:** We would like to clarify that our paper does not consider the agnostic case, that is, the function $f$ is given. The statement of Theorem 1 says ``given ... $f$'' and Algorithm 2 lists $f$ as part of the input. Therefore, the hardness result of the agnostic case does not apply here.
>     Even in the agnostic case, where one can only have a PTAS for $(1+\epsilon)$-approximation,
>     a smaller query complexity implies solving a smaller-scale regression problem (dimension $\operatorname{poly}(d/\epsilon)$ versus $n$), which is advantageous, particularly when solving such problems is computationally hard (runtime $(d/\epsilon)^{\operatorname{poly}(1/\epsilon)}$ versus $n^{\operatorname{poly}(1/\epsilon)}$).

---

> > ### Comment · Reviewer_kECa · 2024-11-25
> >
> > I am a little bit confused about the term "agnostic" in this rebuttal. Agnostic learning usually means the label $b$ does not equal $f(Ax)$ even for the optimal $x$. This has no relation to whether $f$ is provided or not. In the hardness paper that I mentioned, $f$ is also given, which is just a ReLU function. So I believe the agnostic ReLU learning problem studied in [DKR23] is a special case of the setting studied in this paper. So, the rebuttal does not fully address my concern.

---

> > > ### Author Response · Authors · 2024-11-26
> > >
> > > We thank the reviewer for the explanation and acknowledge our inaccuracy in the earlier response. Admittedly we do not have a good answer to the reviewer's question on computational hardness.
> > > In our formulation, each row of $A$ would be a data point and thus our regression formulation corresponds to a uniform distribution over a finite point set for $x$ in the context of half-space learning. The hardness result may not hold for finite support, if the support size is not large enough. For a continuous marginal distribution of $x$, one can approximate this distribution using a finite support with sufficiently large size and then apply our framework. However, approximating the distribution may require a support size of $\exp(d)$, which does not contradict existing hardness results for time complexity.

---

### Official Review · Reviewer_rYLj · 2024-11-03

**Soundness:** 4
**Presentation:** 3
**Contribution:** 3
**Rating:** 8
**Confidence:** 3

**Summary:**

The paper considers the problem of active regression where the objective is to solve the problems of the $\min \|f(Ax) - b\|_p^p \|$ where $f$ is an unknown Lipschitz function, $A$ is a known matrix. The learner aims to solve the minimization problem by querying only a fraction of elements of $b$. The problem has been well studied for $f(x) = x$. For the case of general $f$, existing results only obtain a constant factor approximation. In this work, the authors obtain a $(1 + \varepsilon)$ approximate solution with general $f$ with a new algorithm and analysis.

**Strengths:**

I think this is an interesting paper overall. While the problem appears to be simple, there is quite of bit of machinery involved in establishing the results and designing the algorithm. I believe there are several novel aspects in the algorithm design and analysis that might be useful in general for future studies in this direction.

**Weaknesses:**

I don't see any glaring weakness in the paper.

**Questions:**

Can the authors elaborate why the techniques in Yasuda (2024) do not carry forward to the case of $p > 2$. As far as I understand, the hard instance needs to be updated where $1/\varepsilon$ is replaced by $d/\varepsilon$ and the analysis should work out. I am trying to understand what does not work out with this choice.

---

> ### Author Response · Authors · 2024-11-17
>
> We thank the reviewer for the comments. Below we explain the difficulty in extending the lower bound from $d=1$ to $d>1$ when $p>2$:
>
> We first briefly summarize the constructions in (Yasuda, 2024) and our paper for $d=1$.
> In (Yasuda, 2024), the first hard instance gives $x^\ast=0$ and $\textsf{OPT}=0$ while the second hard instance gives $x^\ast\neq 0$ and $\textsf{OPT}\leq 1-\epsilon$.
> In our construction, the first hard instance gives $x^\ast=1$ and $\textsf{OPT}=2/\epsilon^p$ while the second hard instance gives $x^\ast=-1$ and $\textsf{OPT}=2/\epsilon^p$.
> The key difference is that Yasuda's construction is asymmetric (with one of the $\textsf{OPT}$ values being $0$) whereas our construction is symmetric.
>
> When extending the result to the case of $d > 1$, Yasuda (2024) constructs a set $S$ of unit vectors of size $d^{p/2}$, where any two vectors in the set are almost orthogonal.
> The existence of such a set is shown in previous work.
> The asymmetry can then be exploited as follows: duplicate the first hard instance in each of these $d^{p/2}$ directions, then choose one of these instances and flip it to the second hard instance with probability $1/2$.
> If the chosen copy is not flipped, we have $x^\ast=0$ and $\textsf{OPT}=0$.
> If the chosen copy is flipped, we have $x^\ast\neq 0$ and $\textsf{OPT}\leq  1-\epsilon  +\text{small error}$. This small error is negligible because the other directions are almost orthogonal to the chosen direction and the $\textsf{OPT}$ value of the first hard instance is $0$.
>
> Unfortunately, our construction is symmetric and neither $\textsf{OPT}$ value is $0$.
> To extend the lower bound to $d^{p/2}$ using the set $S$ defined above, we cannot follow the same approach because of the absence of a "base" instance like the first hard instance in Yasuda's construction.
> This prevents us from setting all copies to be the "base" instance with a random flip on one of them, as the $\textsf{OPT}$ value is not $0$, which makes the small error mentioned above is no longer negligible.
>
> Regarding the possibility of an asymmetric construction in our case, we found that the symmetry is crucial for improving the lower bound to $1/\epsilon^p$ when $d=1$.
> Therefore, we encounter this dilemma when extending the lower bound to the case of $d>1$.

---

> > ### Comment · Reviewer_rYLj · 2024-11-22
> > **Response to the authors**
> >
> > I see, the issue is bit more subtle than I previously thought. You might have thought about this but does taking a scaled version of the problem help here? In particular, after flipping you set the hard instance to be scaled version of the current one. The optimal value will change without affecting $x^*$.

---

> > > ### Author Response · Authors · 2024-11-23
> > >
> > > Thanks for the question.
> > >
> > > If we scale the flipped copy only, it seems that we only need $m = d^{p/2}$ queries to identify the chosen copy where $m$ is the number of copies.
> > > It is because, in our construction, we plant the large value in an **all-one** vector and hence scaling the entire copy makes it immediately distinguishable without considering the planted entry.
> > > In this case, it may not be very helpful.
> > >
> > > To complement our response, we would like to further point out that the reason why the small error mentioned previously is negligible also comes from the fact that $x^*=0$ in the first instance.
> > > It is because if we pick $x$ to be one of the vectors in $S$ mentioned before then part of the error also comes from the inner product of $x$ and other vectors in $S$.
> > > By the almost orthogonality property, this contribution to the error is small.
> > > Hence, in Taisuke's construction, both $x^*=0$ and $\textsf{OPT}=0$ are crucial properties in the argument.

---

### Official Review · Reviewer_82LY · 2024-11-04

**Soundness:** 2
**Presentation:** 2
**Contribution:** 2
**Rating:** 5
**Confidence:** 2

**Summary:**

In summary, this paper studies the single-index (label efficient) active regression problem of $\min_{x} \|f(Ax)-b\|_p$, where $b$ is sampled as efficiently as possible. It contributes a $(1+\epsilon)$-approximation, replacing the constant-factor approximations in the literature, with near optimal query complexities.

**Strengths:**

**Originality:**
It presents novel findings from theoretical and algorithmic perspectives.

**Quality:**
The approach is well motivated, including being well-placed in the literature.

**Clarity:**
The general approach is observable.

**Significance:**
Regarding potential value and impact, the paper has the goal of better addressing a known application/problem, namely single-index active regression, with new theoretical findings of improved query complexities.

**Weaknesses:**

The submission is not sufficiently clear, with many occasions of ambiguous wording.
It seems technically correct but hard to gauge at various times.
The experimental analysis is missing, hence not rigorous or reproducible.

The paper does not sufficiently support the claims. A key issue is that the analysis accompanying the theoretical results of query complexities includes many approximations, the significance of which are mostly omitted without necessary explanations, resulting in a somewhat lacking scientific rigor.

The paper could bring new knowledge of sufficient value to the community. However, it needs to be revised to better present its soundness.

**Questions:**

**Questions:**
- Line 148: why is this approximate?
- Line 163: what is the relation between $\overline{x}$ and $x^*$?
- Line 172: what is this approximate inequality defined as?
- Line 197: is this inequality exact?
- Line 215: again, what is this inexact inequality?
- Line 220: how do you conclude $\hat{x} \in T$?
- Line 299: how do you apply (13) repeatedly, AM-GM inequality? Don't the regularization terms also accumulate?
- Line 397: is the second line an inequality?
- Line 423: how is the last inequality obtained?

**Major Suggestions:**
- There are too many approximative parts in the analysis, which hardens the verifiability. The writing should be improved to ease digestion.
- Line 68: improvement from [2023b] to [2024] is unclearly worded.
- Line 191: error terms need to be expanded and explained more clearly.
- Line 278: need more intuition into (10) and (13).
- Line 461: this multi-dimentional extension could be better explained, feels a bit rushed.

**Minor Suggestions:**
- Line 16: V1, V2? better write these (and similar ones) as $\max, \min$ initially to ease reading.

---

> ### Author Response · Authors · 2024-11-17
> **Full, rigorous proof provided in the appendix**
>
> We thank the reviewer for the detailed comments. We would like to clarify that the main body of the paper presents only an outline of our approach, focusing on the core ideas, while **the full rigorous proof is provided in the appendix**. Given that rigorous proofs have been included, we do not think the paper is ''not rigorous or reproducible'' or lacks ''scientific rigor''.
>
> We would also like to clarify that this submission focuses on the theoretical aspects of query complexity, with a particular emphasis on the $\epsilon$-dependence of query complexity. While fewer queries may suffice for certain real-world datasets, we consider this a potential direction for future work.
>
> Below, we address the reviewer's major suggestions:
>
> - **Line 68:** There is a typo. Gajjar et al. (2023b) achieved $\tilde{O}(d/\epsilon^4)$ queries, which was later improved to $\tilde{O}(d/\epsilon^2)$ by Gajjar et al. (2024).
>
> - **Line 191:** The main purpose of this part (Lines 187--200) is to explain why we set the regularization parameter to be $\epsilon$.
> Indeed, in this equation, we would like to explain $\tau$ should be smaller than $\epsilon$.
> As explained in Line 193, the final guarantee has the term $\epsilon \|Ax^\ast\|_p^p$ and hence we should set $\tau$ smaller than $\epsilon$ *regardless* of the error term.
> Hence, the error term plays a minor role in this argument and we intentionally did not fully expand it.
> The use of regularized regression was previously employed in the work of Gajjar et al. (2024). Our submission uses the same regularized regression problem. While we could have omitted the explanation and used the regularized version as our starting point, we chose to include it for completeness.
>
> - **Equations (10) and (13):** These error bounds are similar to previous work such as that of Yasuda (2024). The most novel part of this submission is the section on ``further improvement'', which enables us to obtain a near-optimal dependence on $\epsilon$ in query complexity.
> Therefore, we intentionally phrased them as black boxes as they are not our main contribution.
> We will make this clearer in the revised version.
>
> - **Extension to $d>1$ is not as difficult as $d=1$.** Once the argument for $d=1$ works out, it is more or less a standard technique to extend it to $d > 1$.
> Hence, we presented it in a way that focuses on our main contribution, which is the case of $d=1$.
> The full proof for $d>1$ can be found in the appendix.
> We will polish the paper to improve the presentation.

---

> ### Author Response · Authors · 2024-11-17
>
> The following are our responses to the reviewer's questions.
>
> - **Line 148:** Here we are looking at the $\ell_p$ norm and it can be proved that exact equality (i.e. isometric embedding) is generally impossible. Normally the approximation in subspace embeddings is multiplicative, i.e., one has a distortion parameter $\epsilon$ and wants $(1-\epsilon)\|Ax\|_p \leq \|SAx\|_p \leq (1+\epsilon)\|Ax\|_p$.
>
> - **Line 163:** In the definition of $\operatorname{Err}(x)$ in Equation (5), we use $x^\ast$, which is the optimal solution to the regression problem. Here, we are saying that this $x^\ast$ can be actually replaced with an arbitrary point $\bar{x}$ in the definition of $\operatorname{Err}(x)$ and the argument for upper bounding $\operatorname{Err}(x)$ still works.
>     Indeed, in our detailed analysis, we will pick $\bar{x}$ to be different points including $x^\ast$ and hence we would like to highlight this flexibility.
>
> - **Line 172:** The symbol ``$\lesssim$'' indicates an inequality up to a constant factor, similar to the big-$O$ notation. Specifically, we write $f(x) \lesssim g(x)$ if there exists a constant $C$ such that $f(x) \leq C g(x)$.
> This notation is explained in the Preliminaries section (Section A) of the appendix and we will move it to the main text to improve the presentation.
>
> - **Line 197:** This inequality is exact.
>
> - **Line 215:** This inequality uses ``$\lesssim$'',  indicating that a constant factor is lost.
>
> - **Line 220:** Equation (9) shows that $\|A\hat{x}\|_p^p \lesssim R$ for $R = \frac{1}{\epsilon}\textsf{OPT} + \|Ax^\ast\|_p^p$. This fits the definition of $T$ (with appropriate constants hidden in $\lesssim$) and so $\hat{x}\in T$.
>
> - **Line 299:** To repeat the process, observe that Line 296 gives a better bound on $\|A\hat{x}\|_p^p \lesssim \frac{1}{\sqrt{\epsilon}}\textsf{OPT} + \|Ax^\ast\|_p^p$ and so we can define a new, smaller $R$ to be $R = \frac{1}{\sqrt{\epsilon}}\textsf{OPT} + \|Ax^\ast\|_p^p$. Using this new $R$ and repeat the argument, we can then obtain a better bound $\|A\hat{x}\|_p^p \lesssim \frac{1}{\epsilon^{1/4}}\textsf{OPT} + \|Ax^\ast\|_p^p$ and so on.
>     The regularization term does accumulate, but the accumulated error is still bounded by $\log$ factors and will be absorbed in the $\log$ factor in the final bound.
>     We will make this clearer in the revised version.
>
> - **Line 397:** We thank the reviewer for pointing out this confusion.
>     This line should read "$\leq (1 + c  \epsilon)2^p(n-k) + c \epsilon  \|a x^\ast\|_p \approx (1 + c \epsilon)2^p n(1-2\epsilon) + c \epsilon  \|a x^\ast\|_p$" by Line 377.
>     We can rigorously express this approximation as an inequality by applying the Chernoff bound, as detailed in the appendix. We use $\approx$ here to substitute in the expectation, providing intuition of our idea.
>
> - **Line 423:** There is a typo. The "$(1/\epsilon-1)^p$'' should be ``$(1/\epsilon+1)^p$''.
>     Recall that $n = 1/\epsilon^p$. Thus, we have $n-1+(1/\epsilon+1)^p = n-1 + (1+\epsilon)^p/\epsilon^p = n-1+n(1+\epsilon)^p > 2n(1+\epsilon)$.

---

> ### Comment · Reviewer_82LY · 2024-11-24
>
> Can we say that you achieve $(1+\epsilon)$-approximation with the complexity of Gajjar et al. (2024)?
>
> Additionally, regarding (13) and "Further Improvement", is your main contribution showing how the iterative refinement of (13) results in tighter guarantees?
>
> Your work is not experimentally rigorous, or reproducible in general, since no experiments are provided.
>
> Similarly, it seems many approximations are given without sufficient justification, e.g., how some terms can easily be omitted without issues in the subsequent calculations. This makes the analysis (at least in the main text) lacking in rigor. You could have potentially frame your work in a more digestible manner.
>
> Moreover, the appendix feels convoluted. There are many assumptions, for which the corresponding justifications seem to be missing. Some examples:
> - Line 978: How did the error term in Corollary 16 turn into this, where is the $\epsilon$-term on the left side? Consequently, it is not clear how (27) is reached.
> - Line 1058: Where is this assumption coming from?
> - Line 1080: Following here, it is not clear to me how you compute $Y_i$ or how such a given $K$ exists.
>
> Regarding Line 299, you need to carefully explain how the omitted additive and multiplicative terms accumulate with successive iterations. It is hard to see and judge in the current approximative form.

---

> > ### Author Response · Authors · 2024-11-25
> >
> > We thank the reviewer for further questions and comments, which we address below.
> >
> > - The reviewer is correct that we achieve a $(1+\epsilon)$-approximation with the same query complexity (up to log factors) as in (Gajjar et al., 2024). We would also like to note that (Gajjar et al., 2024) considers only $p=2$ while we handle the full range $p \geq 1$. Our analysis for controlling $\sup_{x\in T} \operatorname{Err}(x)$ is also simpler and more generalizable than theirs,
> > though we acknowledge that similar arguments have appeared in prior work and do not claim this to be our contribution. We also prove the near-optimality of our query complexity for $p\in [1,2]$ and the near-optimality of the $\epsilon$-dependence for $p > 2$.
> >
> > - The section in ``Further Improvement'' improves the $\epsilon$-dependence in query complexity from a more straightforward $\tilde{O}(1/\epsilon^4)$ to the near-optimal $\tilde{O}(1/\epsilon^2)$.
> >
> > - Given the length of the full analysis, only the key ideas are presented in the main body. To better emphasize proof ideas, it is common practice to omit minor terms or use approximations, avoiding a complex presentation. Readers with doubts or questions may refer to the detailed proofs for clarification.
> >
> > - The algorithm is explicitly provided and is straightforward to implement; we do not see issues in its implementation. The challenge lies in the analysis rather than the algorithm itself. Since (1) the query complexity is rigorously established through proof and shown to be nearly optimal, and (2) the claims on query complexity are derived from theoretical analysis rather than empirical experiments, we do not see concerns regarding reproducibility or the necessity of numerical experiments to validate our claims.
> >
> > - We do not introduce additional assumptions in the full proof. It is common for lemmas or intermediate results to be stated with specific assumptions, which are then verified when these lemmas are invoked in proofs elsewhere.
> >
> > - **Line 978**: We shorthand $\log d\sqrt{\log(n/(\epsilon d))}$ as $\operatorname{polylog} n$ here, since one can assume that $n\gtrsim d/\epsilon^2$ (mentioned in Line 958), otherwise one can just sample all entries of $b$ for such a small $n$. Under this assumption, we have $\log(n/(\epsilon d)) \lesssim \log((n/d)\cdot\sqrt{n/d}) \lesssim \log n$.
> >
> > - **Line 1058**: The assumption comes from Line 865, which is part of the assumptions stated in the statement of Lemma 13.
> >
> > - **Line 1080**: There is a typo. It should read "$C_6 \leq X_i \leq C_5^{1/\theta}\cdot 100/\epsilon$". By the definition that $Y_{i+1} = 1 + \cdots$ (Line 1071) and $Y_0 = 1$, we have $Y_i\geq 1$. We also have $X_i \leq 100C_5^{1/\theta}/\epsilon$ as mentioned above, so $Y_i/X_i \geq \epsilon/(100 C_5^{1/\theta})$ and one can choose $K = 100 C_5^{1/\theta}$ as claimed.
> >
> > - **Line 299**: We agree with the reviewer that the current presentation may cause a misunderstanding about the constant hidden in $\lesssim$ potentially becoming large (although the detailed proof shows this is not the case).  In the revised version, we will include a better description of the proof idea.

---

> ### Comment · Reviewer_82LY · 2024-12-03
>
> Thank you for the reply. Although, I am not fully convinced by the state of the paper, I am raising my score after considering your rebuttals.

---

### Official Review · Reviewer_qzE3 · 2024-11-04

**Soundness:** 3
**Presentation:** 2
**Contribution:** 2
**Rating:** 5
**Confidence:** 2

**Summary:**

This paper presents and algorithm to provide an epsilon approximate solution to the active regression problem. They show that their number of query points is minimax optimal.

**Strengths:**

S1 - The paper is generally well written.

S2 - The technical claims in the paper are generally well explained.

S3 - The idea is interesting.

**Weaknesses:**

W1 - Literature review is a bit lacking.

W2 - No numerical experiments.

W3 - The presentation can be improved by first introducing the algorithm. In the current version, the algorithm is somewhat built up during the upper bound proofs.

**Questions:**

Q1 - Following the impossibility result of Gajjar et al., how does your result compare?

Q2 - Why are the probabilities different in Theorem 1,2,3.

Q3 - In addition to the constant probability ones, do you have a high probability results, where the number of queries is a function of that probability?

---

> ### Author Response · Authors · 2024-11-17
>
> We thank the reviewer for the comments. Below we address the reviewer's questions.
>
> - **Q1:** As explained in the introduction section, the impossibility result of Gajjar et al. (2024) states that one cannot hope for
>     $$
>         \|f(A\hat{x})-b\|_2^2 \leq C\cdot \textsf{OPT}
>     $$
>     with an algorithm using a polynomial number of queries.
>     Therefore, their error guarantee contains an additive term as
>     $$
>         \|f(A\hat{x})-b\|_2^2 \leq C(\textsf{OPT} + \epsilon L^2\|Ax^\ast\|_2^2).
>     $$
>     Note that this is for $p=2$ with a constant-factor approximation.
>     For us, this hardness result remains and our result has a similar error guarantee to that of Gajjar et al. (2024). We obtain a $(1+\epsilon)$-approximation for general $p$ with an analogous error (see Theorem 1)
>     $$
>         \|f(A\hat{x})-b\|_p^p \leq (1+\epsilon)(\textsf{OPT} + \epsilon L^p\|Ax^\ast\|_p^p).
>     $$
>     By rescaling $\epsilon$ by at most a constant factor, we can rewrite the error guarantee above as
>     $$
>         \|f(A\hat{x})-b\|_p^p \leq (1+\epsilon)\textsf{OPT} + \epsilon L^p\|Ax^\ast\|_p^p.
>     $$
>
> - **Q2:**  Theorem 1 is an upper bound while Theorems 2 and 3 are lower bounds. The success probabilities in these theorems can be made an arbitrary number greater than $1/2$ by adjusting the constants (the hidden constants in big-$O$ and big-$\Omega$ notations usually depend on the failure probability).
> We chose these constants specifically to simplify the analyses and will make appropriate adjustments to improve the readability.
>
> - **Q3:** At present, we do not have a high-probability result, as the primary focus is on achieving near-optimal query complexity.
> For failure probability $\delta$, the current argument introduces a $1/\delta$ factor in the query complexity, owing to the application of Markov's inequality for the initial value of $R$. It may be possible to improve the dependence on $\delta$ using the techniques similar to those in (Musco et al., 2022), and we leave this as a direction for future work.
>
> Regarding the weaknesses,
>
> - **W1:** There are relatively few papers with provable query complexities for this problem, and, to the best of our knowledge, we have cited all known theoretical works in this line of research.
> On the other hand, we believe that there are other papers related to our problem and including them may help readers better understand our problem and active regression in general.
> We will provide a more comprehensive literature review of the relevant areas in the revised version.
>
> - **W2:** This submission focuses on the theoretical aspects of query complexity, with a particular emphasis on the $\epsilon$-dependence of query complexity.
> While fewer queries may suffice for certain real-world datasets, we consider this a potential direction for future work.
>
> - **W3:** The algorithm is presented in Section 3. We agree with the reviewer that moving it earlier in the paper may enhance readability.

---

> > ### Comment · Reviewer_qzE3 · 2024-11-27
> >
> > Does the main difference to (Gajjar et al., 2024) reside in the generation of the sampling matrix?

---

> > > ### Author Response · Authors · 2024-11-27
> > >
> > > The main difference from (Gajjar et al. 2024) is that we achieve a $(1+\epsilon)$-approximation, which requires a significantly harder analysis, in particular, the development of new techniques (described in the section titled 'Further Improvement') to optimize the $\epsilon$-dependence. The sampling matrix is different though similar, (Gajjar et al. 2024) samples the rows of $A$ and $b$ using leverage scores as they consider only $p=2$, while we use Lewis weights for general $p$.

---

### Meta-Review · Area_Chair_rqfQ · 2024-12-26

**Metareview:**

The paper considers the following active learning problem, find an (approximate) solution to minimise $\Vert f(Ax) - b \Vert_p$ for $p \geq 1$ by making queries to the entries of $b$. The paper obtains an $(1 + \epsilon)$ approximation for this problem by making a certain number of queries, for which some lower bounds are also provided. The $(1 + \epsilon)$ approximation is somewhat unusual in that the additive term is not just $\epsilon \cdot \mathrm{OPT}$, but is $\epsilon ( \mathrm{OPT} + L^p \Vert A x^* \Vert_p^p)$, where $x^*$ is the min-norm optimizer. The inverse link function $f$ is assumed to be $L$ Lipschitz. The problem is interesting, though the results for single-index models are not as clean as in the case of linear regression. The techniques appear to be relatively standard.

**Additional Comments On Reviewer Discussion:**

The two most negative reviewers didn't provide very useful comments as part of their reviews or in subsequent discussion. They also declared low confidence in their own reviews. As a result, I've discounted them substantially, pushing this paper into accept range. However, I wouldn't object if it were rejected.

---

### Decision · Program_Chairs · 2025-01-22

Accept (Poster)